# Synaptic targets of photoreceptors specialized to detect color and skylight polarization in *Drosophila*

**Emil Kind[1†], Kit D Longden[2†], Aljoscha Nern[2†], Arthur Zhao[2†], Gizem Sancer[1‡], Miriam A Flynn[2], Connor W Laughland[2], Bruck Gezahegn[2], Henrique DF Ludwig[2], Alex G Thomson[2], Tessa Obrusnik[1], Paula G Alarcón[1], Heather Dionne[2], Davi D Bock[2§], Gerald M Rubin[2], Michael B Reiser[2]\*, Mathias F Wernet[1]\***

[1]Institut für Biologie – Abteilung Neurobiologie, Fachbereich Biologie, Chemie & Pharmazie, Freie Universität Berlin, Berlin, Germany; [2]Janela Research Campus, Howard Hughes Medical Institute, Ashburn, United States

**\*For correspondence:**
reiserm@janelia.hhmi.org (MBR);
mathias.wernet@fu-berlin.de (MFW)

†These authors contributed equally to this work

**Present address:** ‡Department of Neuroscience, Yale University School of Medicine, Sterling Hall of Medicine, New Haven, United States; §Department of Neurological Sciences, University of Vermont, Burlington, United States

**Competing interest:** The authors declare that no competing interests exist.

**Abstract** Color and polarization provide complementary information about the world and are detected by specialized photoreceptors. However, the downstream neural circuits that process these distinct modalities are incompletely understood in any animal. Using electron microscopy, we have systematically reconstructed the synaptic targets of the photoreceptors specialized to detect color and skylight polarization in *Drosophila*, and we have used light microscopy to confirm many of our findings. We identified known and novel downstream targets that are selective for different wavelengths or polarized light, and followed their projections to other areas in the optic lobes and the central brain. Our results revealed many synapses along the photoreceptor axons between brain regions, new pathways in the optic lobes, and spatially segregated projections to central brain regions. Strikingly, photoreceptors in the polarization-sensitive dorsal rim area target fewer cell types, and lack strong connections to the lobula, a neuropil involved in color processing. Our reconstruction identifies shared wiring and modality-specific specializations for color and polarization vision, and provides a comprehensive view of the first steps of the pathways processing color and polarized light inputs.

## Editor's evaluation

This paper will be of interest to the large class of neuroscientists who perform network analyses and are interested in the processing of visual information. It sets a new standard in connectomic analysis because it combines EM data of a whole fly brain with fluorescent labeling of specific neurons. The key claims of the manuscript are well supported by the data, and the approaches used are thoughtful and rigorous.

## Introduction

Both the wavelength and the polarization angle of light contain valuable information that can be exploited by many visual animals. For instance, color gradients across the sky can serve as navigational cues, and skylight's characteristic pattern of linear polarization can also inform navigation by indicating the orientation relative to the sun (*Heinze, 2017*; *Figure 1A*). The spectral content of light is detected by groups of photoreceptor cells containing rhodopsin molecules with different sensitivities, often organized in stochastic retinal mosaics (*Rister and Desplan, 2011*), and specialized, polarization-sensitive photoreceptors have been characterized in many species, both vertebrates and

invertebrates (*Nilsson and Warrant, 1999*). These two visual modalities, color and polarization vision, require the processing of signals over a wide range of spatial and temporal scales, and many questions remain about how the signals from functionally specialized photoreceptors are integrated in downstream neurons. Are color and polarization signals mixed at an early stage, or are they processed by different, modality-specific cell types? Do separate pathways exist that selectively process and convey information from photoreceptor types in the retinal mosaic to targets in the central brain? The full scope of the early synaptic stages of color and polarization circuitry is unknown in any animal, and the analysis of electron microscopy (EM) connectomes is ideally suited to exhaustively answer these questions, especially when corroborated with genetic labeling of cell types and circuit elements imaged with light microscopy. We make significant progress on the questions by mapping the neuronal connections of specialized, identified photoreceptors within the *Drosophila* full adult fly brain data set (*Zheng et al., 2018*) and validate many of these results by using the powerful genetic tools available in *Drosophila*.

While studies in many insects have contributed to the understanding of polarized light and color vision (*Homberg, 2015*; *Dacke and El Jundi, 2018*; *Heinze, 2017*; *Hempel de Ibarra et al., 2014*), the visual system of *Drosophila* offers many advantages for the exploration of neural circuits (*Wernet et al., 2014*). Anatomical studies are facilitated by the stereotyped, repetitive structure of the optic lobes, with many cell types, the so-called columnar neurons, found in repeated circuit units, called visual columns, that are retinotopically arranged and each correspond to one of the ~800 unit eyes (ommatidia) of the compound eye. Over 100 optic lobe cell types have been described in classical Golgi work (*Fischbach and Dittrich, 1989*), by more recent studies combining genetic labeling with light microscopy (*Morante and Desplan, 2008*; *Otsuna and Ito, 2006*; *Wu et al., 2016*; *Nern et al., 2015*) and, for some cell types, through EM reconstructions that have revealed not only cell morphologies but also detailed synaptic connectivity (*Takemura et al., 2008*; *Takemura et al., 2013*; *Takemura et al., 2015*; *Meinertzhagen and O'Neil, 1991*; *Rivera-Alba et al., 2011*; *Shinomiya et al., 2014*). Furthermore, genetic tools (*Jenett et al., 2012*; *Pfeiffer et al., 2008*; *Kvon et al., 2014*; *Dionne et al., 2018*; *Tirian and Dickson, 2017*), and, most recently, gene expression data (*Davis et al., 2020*; *Konstantinides et al., 2018*; *Özel et al., 2021*; *Kurmangaliyev et al., 2020*) are available for many optic lobe cell types. An enabling feature of this emerging body of work is that in nearly all cases, distinct cell types can be reliably identified across data sets, such that new studies often directly enrich prior ones.

Each *Drosophila* ommatidium contains eight photoreceptors whose output is processed in a series of neuropils called the lamina, medulla, lobula, and lobula plate that together form the optic lobes of the fly (*Fischbach and Dittrich, 1989*). Outer photoreceptors R1-6 project to the lamina neuropil, and serve as the main input to the motion vision circuitry (*Mauss et al., 2017*); inner photoreceptors R7 and R8 pass through the lamina without connections and project directly to the deeper medulla neuropil, which also receives lamina projections (*Fischbach and Dittrich, 1989*). The organization of the inner photoreceptors along the dorsal rim area (DRA) of the eye characteristically differs from that of the rest of the retina. In the non-DRA part of the retina, R7 and R8 differ in their axonal target layers, with R7 projecting to layer M6, and R8 to layer M3 (*Fischbach and Dittrich, 1989*). R7 and R8 also differ in their rhodopsin expression, being sensitive to short wavelength ultraviolet (UV, R7) and blue (R8), respectively, in so-called 'pale' ommatidia (*Chou et al., 1996*; *Papatsenko et al., 1997*), and to long wavelength UV (R7) and green (R8) in 'yellow' ommatidia (*Salcedo et al., 1999*; *Huber et al., 1997*; *Figure 1B*). Pale and yellow ommatidia are distributed randomly (*Feiler et al., 1992*; *Fortini and Rubin, 1990*), at an uneven ratio that is conserved across insects (*Wernet et al., 2015*; *Kind et al., 2020*). Meanwhile, DRA ommatidia are morphologically and molecularly specialized for detecting skylight polarization (*Wernet et al., 2003*; *Wernet et al., 2012*; *Wada, 1974a*), that *Drosophila* can use to set a heading (*Mathejczyk and Wernet, 2019*; *Mathejczyk and Wernet, 2020*; *Warren et al., 2018*; *Weir and Dickinson, 2012*). In the DRA, the inner photoreceptors express the same UV rhodopsin (Rh3; *Fortini and Rubin, 1991*, *Fortini and Rubin, 1990*) and detect perpendicular angles of polarized UV light (*Weir et al., 2016*). In contrast to the rest of the medulla, R7-DRA and R8-DRA project to the same medulla layer (M6; *Chin et al., 2014*; *Pollack and Hofbauer, 1991*; *Fischbach and Dittrich, 1989*), where their targets include polarization-specific cell types (*Sancer et al., 2019*; *Sancer et al., 2020*; *Hardcastle et al., 2021*). Across insects, a 'compass pathway' connects the DRA to the central brain via the anterior optic tubercle (AOTU; *Homberg, 2015*; *Hardcastle et al.,*

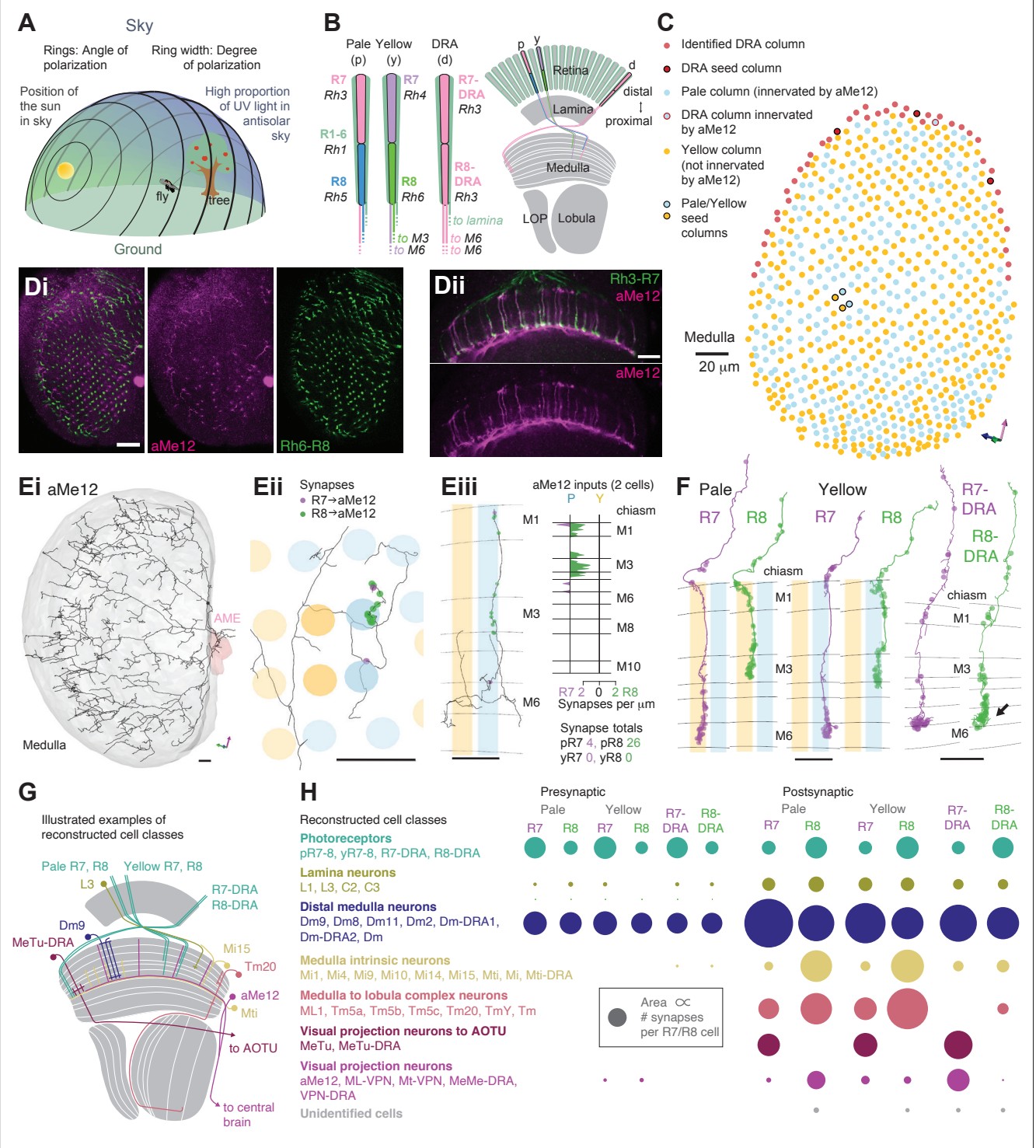

**Figure 1.** Systematic reconstruction of all synaptic targets of inner photoreceptor subtypes. (**A**) Simplified schematic summarizing some of the most salient visual stimuli of a fly (center): celestial cues (sun), color gradients (distribution of green versus UV wavelengths) and skylight polarization (as defined by degree of polarization and angle of polarization) can be used for navigation, as well as more or less colorful landmarks (tree). (**B**) Schematic representation of the fly visual system. Left: In the retina, inner photoreceptor R7 (distal) and R8 (proximal) rhodopsin expression differs across three functionally specialized subtypes pale (**p**), yellow (**y**), and DRA (**d**). Rh3 and Rh4 opsins are both UV-sensitive, whereas Rh5 and Rh6 are more sensitive to blue and green wavelengths, respectively. R1-6 express Rh1 and are broadly sensitive to green and UV wavelengths. Only in the DRA, both R7 and R8 axons terminate in the same layer of the medulla neuropil (**M6**), which is the target layer of R7 cells outside the DRA, and non-DRA R8 cells terminate in layer M3. Right: Overview of the main four optic lobe neuropils. Only the lamina and medulla receive direct photoreceptor input. (**C**) Distribution

*Figure 1 continued on next page*

*Figure 1 continued*

of medulla columns downstream of either p (light blue), y (yellow), and DRA (red) photoreceptors reconstructed from the *Zheng et al., 2018* data set. Pale columns were assigned via presence of aMe12 long vertical projections (see below). Seven seed columns used for systematic reconstruction are highlighted with black circles. The colored axes indicate the orientation of the displayed view: green for lateral, magenta for dorsal, and blue for posterior axes. Scale bar: 20 µm. (D) Double labeling of aMe12 vertical processes and yellow R8 cells. See Materials and methods and *Supplementary file 4* for fly genotypes and other details of this and other light microscopy panels. Left (Di) Confocal image showing the array of medulla columns with labeling of aMe12 neurons (purple) and yellow R8 axons (green). Note that the two patterns appear near-mutually exclusive. Right (Dii) Side view of aMe12 vertical projections and pale R7 axons (green). (Di–ii) Scale bars: 20 µm. (E) Left (Ei) Skeleton of the optic lobe part of a fully reconstructed aMe12 neuron (gray), with processes leaving the medulla through the accessory medulla (pink). (Eii) Across the four central seed columns (darker shading), synaptic input from R7 (purple dots) and R8 photoreceptors (green dots) to aMe12 is specific to the designated pale columns. (Eiii) Left: Sideview depicting the distribution of R7 and R8 inputs into aMe12 across medulla layers (color code of pale and yellow columns as before). Right: Same distribution plotted as synapses/µm, for both pale and yellow columns (color code as before). (Ei–iii) Scale bars: 10 µm. (F) Reconstructed pale R7 and R8, yellow R7 and R8, and R7-DRA and R8-DRA terminals with R7 presynapses in purple and R8 presynapses in green. Note the termination of R8-DRA in the R7 target layer M6 (arrow). Scale bars: 10 µm. (G) Illustrations of examples of cell types from reconstructed neuron classes, including lamina monopolar (L3), distal medulla (Dm9), medulla intrinsic (Mi15), transmedulla (Tm20), medulla tangential intrinsic (Mti), visual projection neurons targeting the central brain (e.g. aMe12), and medulla-to-tubercle (MeTu) cells projecting to the anterior optic tubercle (AOTU). (H) For the synapses from seed column inner photoreceptors to cells with ≥3 synapses, we identified the cell type or cell class of 99.5% (3803/3821 synapses). Overview of the relative strength of these connections with different neuron classes (color code as before), including with the unidentified cells (*Table 2*, *Table 3*, *Table 4*, *Table 5*), across pale, yellow, and DRA columns (pre- and postsynaptic). Area of circles corresponds to the number of synapses per R7 and R8 cell.

The online version of this article includes the following figure supplement(s) for figure 1:

**Figure supplement 1.** Example electron microscopy (EM) images.

**Figure supplement 2.** Comparison with medulla-7-column connectome.

*2021*; *Pfeiffer and Kinoshita, 2012*). Anatomical and functional data from *Drosophila* suggests that the non-DRA medulla is also connected to the compass pathway (*Omoto et al., 2017*; *Hardcastle et al., 2021*; *Otsuna et al., 2014*), potentially forming parallel pathways for processing different celestial cues (*Timaeus et al., 2020*; *Tai et al., 2021*).

EM studies have already revealed some of the circuitry downstream of R7 and R8 (*Takemura et al., 2013*; *Takemura et al., 2015*; *Gao et al., 2008*). For example, axons of R7 and R8 from the same ommatidium are reciprocally connected with inhibitory synapses (*Takemura et al., 2013*), leading to color-opponent signals in their presynaptic terminals (*Schnaitmann et al., 2018*). Interestingly, R7-DRA and R8-DRA also inhibit each other (*Weir et al., 2016*). Other known R7 and R8 targets in the main medulla include local interneurons (e.g. Dm8; *Gao et al., 2008*; *Karuppudurai et al., 2014*; *Pagni et al., 2021*; *Menon et al., 2019*) and projection neurons that provide connections to deeper optic lobe regions (e.g. Tm5 and Tm20 neurons; *Karuppudurai et al., 2014*; *Meinertzhagen et al., 2009*; *Gao et al., 2008*). A previous light microscopy study (*Karuppudurai et al., 2014*) identified a single cell type, Tm5a, that is specific for yellow medulla columns; this neuron has been used to identify pale and yellow columns in an EM volume (*Menon et al., 2019*; *Takemura et al., 2015*; *Karuppudurai et al., 2014*). Using genetic labeling techniques, four classes of TmY cells have also been reported as specific targets of pale versus yellow photoreceptors (*Jagadish et al., 2014*), yet previous connectomic studies did not reveal similar cells. The currently most comprehensive EM study of the medulla reconstructed the connections between neurons in seven neighboring medulla columns (*Takemura et al., 2015*), revealing a detailed, yet incomplete, inventory of cell types connected to R7–8. This data set, now publicly available (*Clements et al., 2020*), is remarkable for its dense reconstruction of columnar circuits, but could not be used to identify many multicolumnar neurons, that were cut-off at the edge of the data volume, leaving ~40% of R7–8 synapses onto unidentified cell types. In addition, no EM-based reconstructions of DRA neurons and their connections are currently available.

Here, we present a comprehensive reconstruction of all inner photoreceptor synaptic outputs and inputs, from pairs of pale and yellow columns and from three DRA columns in the *Zheng et al., 2018*, data set. These reconstructions were carried out within a full-brain volume, such that, for the first time, nearly all neurons connected to these photoreceptors were identified. We discovered, through light microscopy, a large visual projection neuron (VPN) with distinctive morphology, named accessory medulla cell type 12 (aMe12), that selectively innervates pale ommatidia across the medulla. We then reconstructed and subsequently used this cell to identify pale and yellow columns, from which we enumerated the connectivity of R7 and R8 with known and novel cell types within the optic lobes, and projecting to the central brain, including more cells with pale-yellow specificity, and synapses on axons

between neuropils. In the DRA, we show that cellular diversity is reduced, with local interneurons and projection neurons to the AOTU dominating, and connections to the lobula virtually missing. We identify circuit motifs shared between DRA and central columns, and describe modality-specific cell types, including cells with interhemispheric connections and projections to the central brain. Together, we identify the connected neurons that account for 96% of these inner photoreceptor synapses, a comprehensive set of the neurons that comprise the first step of the pathways through which color and polarization signals are transduced to the rest of the brain.

## Results

### Systematic reconstruction of all synaptic targets of inner photoreceptor subtypes

We reconstructed R7 and R8 photoreceptor axons in the central region of the right optic lobe, as well as inner photoreceptors from 27 DRA columns from the *Zheng et al., 2018* data set (*Figure 1C*). From these, we selected seven 'seed columns' (four central, three DRA), for which all pre- and postsynaptic sites of the photoreceptor axons were identified and manually reconstructed, which is annotated and assembled across EM images (*Figure 1F*; Material and methods). There were 422 skeletons, corresponding to individual cells, contacted by the seed column photoreceptors, and we focused our analysis on the 245 individual cells with >2 synapses to or from the photoreceptors. We reconstructed the morphology of these cells with a sufficient degree of completeness to ascribe a unique cell type to 92% (225/245) or to identify a cell class (Dm, Mi, Tm, or TmY cell class) for an additional 6.5% (16/245), so that 98% (241/245) of these cells were identified. We also traced and uniquely identified the cell type of all 30 cells that provided >2 synaptic inputs to the photoreceptors. In total, we identified 4043 synaptic connections from the seed column R7 and R8 photoreceptors, and 970 synaptic inputs to them, summarized in *Figure 1H*.

To distinguish between pale and yellow columns, we took advantage of our discovery of one specific cell type from the accessory medulla that widely arborized in the medulla with ascending branches that align with the photoreceptors (*Figure 1Ei–iii*). The cell type was selective for R8 input and showed a strong preference for innervating pale columns (*Figure 1D and Eiii*). We identified three of these cells per hemisphere (hereinafter referred to as aMe12), which had heavily overlapping arbors, but only rarely shared ascending branches in the same columns (3/779 columns). Using the presence of aMe12 branches, we were able to assign as pale 38% (297/779) of the medulla columns, in good agreement with studies on the retinal mosaic of *Drosophila* (*Feiler et al., 1992*; *Fortini and Rubin, 1990*; *Bell et al., 2007*; *Hilbrant et al., 2014*). Since further analysis identified a total of 42 DRA columns (see below), we assigned the remaining columns as nominally yellow (*Figure 1C*). Our reconstructions also revealed another new pale-selective VPN, which we named ML-VPN1, and confirmed the previously reported yellow selectivity of Tm5a cells (*Menon et al., 2019*; *Karuppudurai et al., 2014*), findings that further supported the designations of pale and yellow seed columns (see below).

After mapping the pale and yellow columns, we selected four adjacent central seed columns, two nominally pale and two nominally yellow (*Figure 1C*). Across the four central seed columns, the absolute number of both synaptic inputs and outputs for R7 and R8 photoreceptors were in good agreement with previous reports analyzing a medulla-7-column connectome (*Takemura et al., 2015*; *Figure 1—figure supplement 2A-D*). As we had access to the complete brain volume, we additionally found substantial numbers of synapses in the axon bundles projecting between the lamina and the medulla, which accounted for 14% and 12% of the output synapses for R7 and R8, respectively, and 16% and 27% of their input synapses (*Figure 1F*, *Figure 1—figure supplement 2G*).

We readily identified columns in the DRA region of the medulla since it is only there that both inner photoreceptors terminate in medulla layer M6 (*Figure 1F*, arrow; *Pollack and Hofbauer, 1991*; *Fortini and Rubin, 1991*; *Sancer et al., 2019*; *Chin et al., 2014*; *Fischbach and Dittrich, 1989*). The morphology of DRA-specific Dm-DRA1 cells differs significantly between central and polar positions of the DRA (*Sancer et al., 2019*), and we therefore chose one polar DRA seed column and two more equatorial columns (*Figure 1C*). Beyond the 27 DRA columns with manually traced R7-DRA and R8-DRA photoreceptors, we mapped the full extent of the DRA region of the medulla by systematically probing for the presence of two inner photoreceptor profiles in layer M6 without reconstruction

(see Materials and methods), which resulted in a total of 42 identified DRA columns (*Figure 1C*). Direct comparison of the absolute number of pre- versus postsynaptic sites between central and DRA photoreceptors revealed drastic differences: while the numbers of R7 and R7-DRA presynapses were comparable (mean 279, range 262–284 for R7-DRA versus mean 297, range 274–322 for R7), the number of R8-DRA presynapses was reduced by ~60% compared to those of R8 (mean 166, range 157–172 for R8-DRA versus mean 380, range 356–417 for R8). In particular, there was a striking reduction in the number of synapses from R8-DRA to cells intrinsic to the medulla and to cells that projected to the lobula compared to R8 (see below and *Figure 1G and H*).

This report is a comprehensive reconstruction of all inner photoreceptor synaptic outputs and inputs in our sample seed columns. The data set comprises cells that were identified and reconstructed because they are connected, by identifiable synapses, to our seed column inner photoreceptors. To be clear what this data set is not, it is not a reconstruction of all the cells, and the connections between them, in a volume around the seed columns.

We have endeavored to make this comprehensive data set as accessible and navigable as possible. In the following sections we first describe the connections of the central column R7 and R8 cells (Figures 2–5), following the sequence of cell classes in *Figure 1G–H*, before presenting the connections of the DRA (Figures 6–9). For brevity, we refer to the region of the medulla innervated by DRA inner photoreceptors as the DRA region, and for clarity we refer to the central column inner photoreceptors as R7 and R8, and the DRA inner photoreceptors as R7-DRA and R8-DRA. The synaptic outputs from and inputs to the central pale and yellow R7–8 cells are summarized in Table 2, Table 3, for R7-DRA and R8-DRA in Table 4, Table 5, and presentations of individual cell types include summaries of total photoreceptor synapse counts (e.g. *Figure 1Eiii* for aMe12). The text highlights downstream cell types that are selective for inputs from either R7 or R8 or are preferentially found in pale or yellow columns. We used 65% as the threshold to consider an input selective and tables of connectivities of the individual cells are arranged by cell type and collected together in *Supplementary file 1*. The main figures show the morphology of reconstructed exemplars, with the many full reconstructions noted (and listed in Materials and methods), while the morphologies of individual cells are collected together in *Supplementary file 2*. The numbers of synapses outside the medulla are summarized in *Supplementary file 3*. For the cell types for which we are unaware of published, detailed descriptions, we endeavored to produce light microscopic images of single cell clones which were matched largely based on the cell body location, neuropil-layer-specific branching patterns, and comparisons to any other known cell types (see Materials and methods). In a few cases, we also use light microscopy to illustrate specific features of known neurons or to explore additional properties (such as direct co-labeling with markers for pale and yellow columns) that are not readily accessible by EM. A key prior EM data set for comparison with our results is the medulla connectome of *Takemura et al., 2017*; *Takemura et al., 2015*, fully released in *Clements et al., 2020*. We refer to this connectome as the 'medulla-7-column connectome'. For ease of reference, all abbreviations used, including anatomical abbreviations, are listed in *Table 1*.

## Synapses between R7 and R8, and with lamina cell types

The inner photoreceptors inhibit each other via histaminergic synapses, a key mechanism for converting their sensitivity to specific wavelengths into presynaptic color opponency (*Schnaitmann et al., 2018*). We annotated a large proportion of the synapses between R7 and R8 along the axons between the lamina and the medulla (*Figure 2A*, *Figure 1—figure supplement 2G*). For R8, ~80% of the synaptic input from R7 was found outside the medulla (mean 11.5 synapses per cell, range 9–17), and the R8 inputs to R7 in this region were as frequent (mean 10.3 synapses per cell, range 5–13). These results were not anticipated by prior EM reconstructions of individual neuropils, and our counts are substantial increases in the numbers of inter-photoreceptor synapses (*Figure 1—figure supplement 2D*; *Takemura et al., 2015*; *Meinertzhagen and O'Neil, 1991*).

R7–8 pass through the lamina without forming synapses (*Meinertzhagen and O'Neil, 1991*), and instead form synapses with lamina cell types in the medulla (*Takemura et al., 2008*; *Takemura et al., 2013*). Our reconstructions revealed that 58% of R7–8 inputs to the lamina cells L1 and L3 were also located within the optic chiasm (*Figure 1—figure supplement 1B*, *Figure 1—figure supplement 2G*, *Figure 2—figure supplement 1A, B*).

**Table 1.** Abbreviations used.

**Abbreviations used:**

|  |  |  |
|---|---|---|
|  | EM | Electron microscopy |
|  | FAFB | *Drosophila* Full Adult Fly Brain data set (*Zheng et al., 2018*) |
|  | FIB-SEM | Focused ion-bean serial electron microscopy |
|  | Medulla-7-column connectome | FIB-SEM data set of *Takemura et al., 2017*; *Takemura et al., 2015*. Connectivity accessible through *Clements et al., 2020*. |
|  | UV | Ultraviolet |
|  | P | Pale, for example, pR7 are R7 photoreceptors of pale ommatidia |
|  | Y | Yellow, for example, yR8 are R8 photoreceptors of yellow ommatidia |

Brain regions:

|  |  |  |
|---|---|---|
|  | AME | Accessory medulla |
|  | AOTU | Anterior optic tubercle |
|  | DRA | Dorsal rim area |
|  | FLA | Flange |
|  | LO | Lobula |
|  | IPS | Inferior posterior slope |
|  | LOP | Lobula plate |
|  | SPS | Superior posterior slope |
|  | PLP | Posterior lateral protocerebrum |
|  | L1-6 | Lobula layers 1–6 |
|  | M1-10 | Medulla layers 1–10 |
|  | ME | Medulla |
|  | CA | Mushroom body calyx |

Cell type names or part of cell type names:

|  |  |  |
|---|---|---|
|  | aMe12 | Accessory medulla 12 |
|  | C2-3 | Lamina centrifugal cell types |
|  | Dm | Distal medulla cell class |
|  | -DRA | Suffix indicating the cell type is located in the DRA |
|  | L1-5 | Lamina monopolar cell types |
|  | MeMe-DRA | Cell type connecting ipsilateral medulla and contralateral medulla |
|  | MeTu | Medulla to anterior optic tubercle cell class |
|  | Mi | Medulla intrinsic cell class |
|  | ML | Medulla to lobula cell class |
|  | ML-VPN | Medulla to lobula visual projection neuron (named for similarity to ML cells) |
|  | Mti | Medulla tangential intrinsic cell class |
|  | Mt-VPN | Medulla tangential visual projection neuron |

*Table 1 continued on next page*

*Table 1 continued*

**Abbreviations used:**

| | |
|---|---|
| OA-AL2i3 | Octopaminergic neuron named by Busch et al. (2007) |
| R1-8 | Photoreceptor cell types |
| Tm | Transmedullary cell class (neurons connecting medulla and lobula) |
| TmY | Transmedullary Y cell class (neurons connecting medulla, lobula, lobula plate) |
| VPN | Visual projection neuron (cell connecting optic lobe and central brain) |

## R7–8 synapses with distal medulla cell types

The distal medulla (Dm) neurons included the strongest targets of R7–8 (*Table 2*). Dm9 is glutamatergic and provides excitatory feedback to R7 and R8, regulating their gain and augmenting their color opponency (*Davis et al., 2020*; *Uhlhorn and Wernet, 2020*; *Gao et al., 2008*; *Heath et al., 2020*). It accounted for more than 90% o all the synaptic inputs to R7–8 that were not from photoreceptors (333/369 synapses, *Table 3*). Dm9 cells span multiple columns and tile the medulla (*Nern et al., 2015*; *Takemura et al., 2015*; *Figure 2B*). We reconstructed six Dm9 cells that were contacted by our seed column R7–8s, which received 16% of all R7–8 synapses, and provided 57% of the total seed column photoreceptor synaptic input (*Table 2*, *Table 3*). Of the six Dm9 cells, one cell occupied our seed columns and formed the bulk of the synapses between R7–8 and Dm9 in these columns. The other five cells received far fewer connections and only in medulla layers M1 and M6. The well-connected Dm9 cell that we fully reconstructed received 92 synapses per seed column (mean, range 78–102), and provided 70 synapses per seed column (mean, range 64–78) offering the most detailed picture of Dm9 connectivity yet (*Supplementary file 1* – Dm9 outgoing). While its R7–8 inputs were biased toward R8 (31% R7 versus 69% R8), the output synapses were balanced (55% R7 versus 45% R8), and pale and yellow columns were equally contacted (outputs from R7 and R8: 54% pale versus 46% yellow; inputs from R7 and R8: 53% pale versus 47% yellow). In total, 8% of the synapses from R7–8 to Dm9 were outside the medulla (34/445 of all R7–8 synapses onto Dm9), on processes following the photoreceptor axons (*Figure 2Bii*, arrow).

The most strongly connected R7 target cell type was Dm8, a central component of *Drosophila*'s circuitry mediating color vision and wavelength-specific phototaxis (*Gao et al., 2008*; *Melnattur et al., 2014*; *Pagni et al., 2021*). Individual Dm8 cells cover multiple columns in medulla layer M6 (*Gao et al., 2008*; *Nern et al., 2015*; *Luo et al., 2020*), and extend processes vertically, with one characteristic process usually reaching higher than the others, up to layer M3, defining a so-called 'home' column (*Figure 2Ci*, arrow). In our reconstructions, we analyzed the connectivity of each Dm8 cell with the surrounding pale and yellow R7 photoreceptors in relation to its home column.

Dm8 cells were exclusively R7 targets, with no synapses from our seed column R8s. Individual R7 cells synapsed onto eight Dm8 cells (mean, range 6–10), contacting 15 Dm8s altogether, and there were five Dm8 cells with processes spanning our seed column R7s and receiving >40 synapses from them. Two of these cells had a pale home column and no input from yellow seed column R7s, and two had a yellow home column and received 13% and 20% of their seed column input from pale R7s (*Supplementary file 1* – Dm8 outgoing). The fifth cell had a pale home column outside the seed column area, and its seed column inputs were divided evenly, 55% from pale and 45% from yellow. Collectively, these data revealed that Dm8 cells selectively innervated columns around the home column, but not simply the closest columns.

In view of the importance of Dm8 cells for processing R7 output, we extended our reconstructions to include all the photoreceptor inputs to three Dm8s, one with a pale home seed column, one with a yellow home seed column, and one with a yellow home column outside the seed columns (*Figure 2Cii*). All three cells were most densely innervated by R7 in their home column: the yellow home column cells received 26% and 23% of their R7 input in the home column, and the pale home column cell received 52% (*Figure 2Ciii*). The innervated columns outside the home column were not the nearest neighbors of the home column, but formed a more irregular, idiosyncratic spatial pattern.

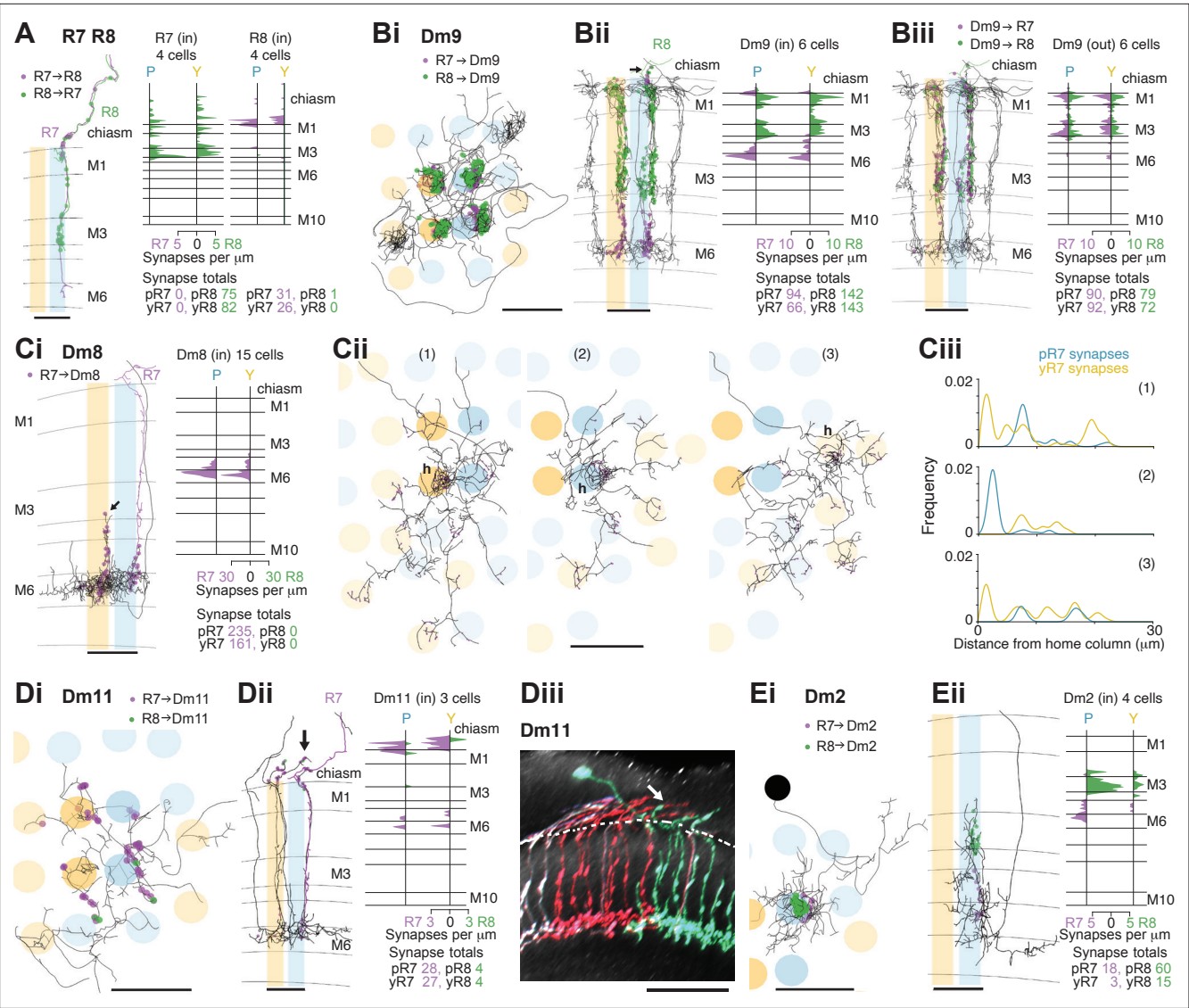

**Figure 2.** Synapses between R7 and R8, and with Dm neurons in the non-dorsal rim area (DRA) medulla. (**A**) Synapses between central R7 and R8 cells. Left: Side view of R7 (purple) and R8 (green), with R7→R8 synapses (purple points) and R8→R7 synapses (green points). Note synapses outside the medulla neuropil (distal to layer M1). Scale bar: 10 μm. Right: Synapse distribution (in synapses/μm) in the yellow and pale seed columns. (**B**) Synapses between R7 and R8 and Dm9 cells. (Bi) Top view of a fully reconstructed Dm9 skeleton (gray) covering all four central seed columns (darker shading) with all R7 (purple) and R8 (green) inputs. (Bii) Left: Side view of R7 (purple) and R8 synapses (green) to the same Dm9 cell across medulla layers. Right: Layer distribution of photoreceptor inputs to six Dm9 cells. (Biii) Left: Side view of Dm9→ R7 (purple) and Dm9→R8 (green) feedback synapses from the same Dm9 cell. Right: Layer distribution of feedback synapses from six Dm9 occupying the seed columns. Most of these synapses are from the illustrated Dm9 cell with other Dm9 cells contributing a small number of connections in M1 and M6. (**C**) Synapses between R7 and Dm8 cells. (Ci): Left: Side view of R7 synapses (purple) to a fully reconstructed Dm8 cell, with one pale R7 cell is shown in purple. Note the characteristic vertical projections of Dm8 in its 'home column' (arrow; yellow R7 cell not shown). Right: Layer distribution of R7 inputs from 14 Dm8 cells innervating the four seed columns. (Cii) Full reconstructions of three Dm8 cells (labeled 1, 2, 3) innervating the four seed columns, with all R7 synapses (purple), including inputs beyond the seed columns. Individual Dm8 home columns (**h**) are marked. (Ci–ii) Scale bars: 10 μm. (Ciii) R7→Dm8 synapses/μm as a function of their distance from the home column, for the three Dm8 cells (1, 2, 3 above). The cells receive dense innervation in their home column and weaker inputs from both pale and yellow R7 cells in their periphery. (**D**) Synapses between R7 and Dm11 cells with R7 (purple) and R8 (green) synapses from seed columns. (Di) Top view of a reconstructed Dm11 skeleton. (Dii) Left: Side view of the layer distribution of photoreceptor synapses, including synapses outside the medulla (arrow). (Di–ii) Scale bars: 10 μm. Right: Layer distribution of R7 and R8 inputs to three Dm11 cells. (Diii) Rendering of a confocal image of MultiColor FlpOut (MCFO) labeled Dm11 cells. Note the characteristic vertical projections leaving the medulla neuropil (arrow). The dashed line marks the approximate boundary of the medulla neuropil. Scale bar: 20 μm. (**E**) Synapses between R7 and R8 and Dm2 cells. (Ei) Top view of a Dm2 skeleton, with R7 (purple) and R8 (green) synapses. (Eii) Left: Side view of layer distribution of photoreceptor synapses onto the same Dm2 cell. (Ei–ii) Scale bars: 10 μm. Right: Layer distribution of R7 and R8 inputs to four Dm2 cells.

*Figure 2 continued on next page*

*Figure 2 continued*

The online version of this article includes the following figure supplement(s) for figure 2:

**Figure supplement 1.** Lamina cell types targeted by central column R7–8.

Together, our data are consistent with Dm8 cells having a central, strong R7 input from either a pale or a yellow cell in its home column and a surround that integrates spatially varied pale and yellow R7 inputs.

Two additional Dm cell types, Dm11 and Dm2, were prominent inner photoreceptor targets. Dm11 cells tile the medulla in layer M6, with each cell covering ~9 columns (*Nern et al., 2015*; *Figure 2Di*), and sending vertical processes that reach far up into the optic chiasm, tracking the R7 and R8 photoreceptor axons (*Courgeon and Desplan, 2019*; *Figure 2Dii*, Diii, arrows). Dm11 was R7-selective, with 87% of its seed column photoreceptor inputs originating from R7s (*Figure 2Dii*). Two-thirds of the Dm11 seed column R7–8 inputs were located outside of the medulla (44/63 synapses), and this was another cell type innervated by photoreceptors much more than previously reported (*Figure 1—figure supplement 2D*; *Clements et al., 2020*; *Takemura et al., 2015*). One Dm11 cell squarely occupied the four seed columns, receiving 52 synapses from the seed column photoreceptors, and this cell showed no clear bias for pale or yellow inputs (40% pale, 60% yellow; *Supplementary file 1* – Dm11 outgoing).

Dm2 is a columnar cell type that spans one to two columns and has processes that reach from layer M6 up to layer M3 to receive synaptic input from R8 cells (*Figure 2Ei*, Eii; *Takemura et al., 2013*; *Nern et al., 2015*). We reconstructed four Dm2s that received 77% of their seed column photoreceptor input from R8 (*Figure 2Eii*) and 81% from pale cells. Three of the Dm2 cells were centered on seed columns, with two in the pale seed columns receiving 36 and 35 synapses, and the cell in a yellow seed column receiving 18; the fourth cell was centered on a neighboring yellow column and innervated one of the pale seed columns (*Supplementary file 2* – Dm2). In the columns of the medulla-7-column connectome that have been identified as pale and yellow (*Menon et al., 2019*; see Materials and methods), Dm2 was also R8-selective (70% of photoreceptor input from R8; *Takemura et al., 2015*; *Clements et al., 2020*). The cell was missing from one of the pale columns in that data set (*Takemura et al., 2015*), but in the columns where it was present, the cells received ~50% more synapses per column in the pale columns (18 mean, range 16–20 in pale columns, versus 12, mean, range 10–15 in yellow columns).

## R7–8 synapses with medulla intrinsic and medulla tangential intrinsic cell types

The medulla intrinsic (Mi) cells connect the distal with the proximal medulla (*Fischbach and Dittrich, 1989*). These two medulla zones are separated by the serpentine layer M7, containing the processes of large medulla tangential cells, including those intrinsic to the medulla which we refer to as medulla tangential intrinsic (Mti) cells. There were three Mti cells contacted by our seed column R7–8, and we matched these cells to light microscopy images of two separate, previously undescribed cell types using the locations of cell bodies, sizes of arbors and layer expression (*Figure 3*; *Figure 3A and B*). Both cell types received a modest number of synapses from our seed column R7–8 (Mti1: 7 synapses; Mti2: 10 and 2 synapses, respectively; *Supplementary file 1* – Mti outgoing), yet were likely to be substantial targets overall since their processes covered large sections of the medulla (*Figure 3A and B*). The first cell type, which we refer to as Mti1, had a cell body located distally to the medulla, in the cell body layer, similar to most medulla neurons. Its processes were asymmetrically oriented mainly along the dorsoventral axis, with dendrites that occupied layers M3–M6, and axonal processes oriented dorsoventrally that spread laterally in layers M7–M8 (*Figure 3A*). The second cell type, here named Mti2 (*Figure 3B*), had cell bodies located at the anterior ventral edge of the medulla, adjacent to where its processes entered and coursed through layer M7 going on to make elaborations in layer 6 and small, further vertical processes that reach vertically up to layers M3 and M4 (*Figure 3Biii* and 3Bvi, arrows). In our EM reconstructions, two cells shared these properties and overlapped around our seed columns (*Figure 3Bv*). From light microscopy, we estimate there are ~50 Mti2 cells. However, we note that the driver line used for this estimate may include related but distinct cell types with a similar cell body location or might not label all Mti2 cells.

**Table 2.** Synaptic targets of seed columns R7 and R8.

| Type | No. | pR7 | yR7 | pR8 | yR8 | Sum | %R7 | %R8 | %p | %y | %Total | %Total_R7 | %Total_R8 |
|---|---|---|---|---|---|---|---|---|---|---|---|---|---|
| Dm9 | 6 | 94 | 66 | 142 | 143 | 445 | 36.0 | 64.0 | 53.0 | 47.0 | 16.4 | 13.5 | 18.8 |
| Dm8 | 15 | 235 | 161 | 0 | 0 | 396 | 100.0 | 0.0 | 59.3 | 40.7 | 14.6 | 33.4 | 0.0 |
| MeTu | 7 | 82 | 93 | 0 | 0 | 175 | 100.0 | 0.0 | 46.9 | 53.1 | 6.5 | 14.7 | 0.0 |
| R7 | 5 | 0 | 3 | 75 | 82 | 160 | 1.9 | 98.1 | 46.9 | 53.1 | 5.9 | 0.3 | 10.3 |
| Tm5c | 6 | 8 | 9 | 20 | 103 | 140 | 12.1 | 87.9 | 20.0 | 80.0 | 5.2 | 1.4 | 8.1 |
| Tm20 | 4 | 5 | 3 | 67 | 65 | 140 | 5.7 | 94.3 | 51.4 | 48.6 | 5.2 | 0.7 | 8.7 |
| Mi15 | 4 | 6 | 7 | 38 | 81 | 132 | 9.8 | 90.2 | 33.3 | 66.7 | 4.9 | 1.1 | 7.8 |
| Mi4 | 4 | 0 | 0 | 58 | 58 | 116 | 0.0 | 100.0 | 50.0 | 50.0 | 4.3 | 0.0 | 7.6 |
| ML1 | 4 | 0 | 0 | 58 | 41 | 99 | 0.0 | 100.0 | 58.6 | 41.4 | 3.7 | 0.0 | 6.5 |
| Dm2 | 4 | 18 | 3 | 60 | 15 | 96 | 21.9 | 78.1 | 81.2 | 18.8 | 3.5 | 1.8 | 4.9 |
| Dm11 | 2 | 28 | 27 | 4 | 4 | 63 | 87.3 | 12.7 | 50.8 | 49.2 | 2.3 | 4.6 | 0.5 |
| L3 | 4 | 19 | 17 | 12 | 13 | 61 | 59.0 | 41.0 | 50.8 | 49.2 | 2.3 | 3.0 | 1.6 |
| Mi1 | 4 | 0 | 0 | 31 | 28 | 59 | 0.0 | 100.0 | 52.5 | 47.5 | 2.2 | 0.0 | 3.9 |
| R8 | 4 | 31 | 26 | 1 | 0 | 58 | 98.3 | 1.7 | 55.2 | 44.8 | 2.1 | 4.8 | 0.1 |
| Tm5a | 2 | 0 | 53 | 0 | 0 | 53 | 100.0 | 0.0 | 0.0 | 100.0 | 2.0 | 4.5 | 0.0 |
| Tm5b | 2 | 39 | 0 | 3 | 6 | 48 | 81.2 | 18.8 | 87.5 | 12.5 | 1.8 | 3.3 | 0.6 |
| Tm | 9 | 17 | 4 | 7 | 16 | 44 | 47.7 | 52.3 | 54.5 | 45.5 | 1.6 | 1.8 | 1.5 |
| Tm5b-like | 3 | 0 | 14 | 0 | 27 | 41 | 34.1 | 65.9 | 0.0 | 100.0 | 1.5 | 1.2 | 1.8 |
| Mi9 | 4 | 3 | 9 | 26 | 1 | 39 | 30.8 | 69.2 | 74.4 | 25.6 | 1.4 | 1.0 | 1.8 |
| L1 | 4 | 2 | 1 | 21 | 14 | 38 | 7.9 | 92.1 | 60.5 | 39.5 | 1.4 | 0.3 | 2.3 |
| aMe12 | 2 | 4 | 0 | 26 | 0 | 30 | 13.3 | 86.7 | 100.0 | 0.0 | 1.1 | 0.3 | 1.7 |
| Dm | 5 | 11 | 7 | 6 | 5 | 29 | 62.1 | 37.9 | 58.6 | 41.4 | 1.1 | 1.5 | 0.7 |

*Table 2 continued on next page*

*Table 2 continued*

| Type | No. | pR7 | yR7 | pR8 | yR8 | Sum | %R7 | %R8 | %p | %y | %Total | %Total_R7 | %Total_R8 |
|---|---|---|---|---|---|---|---|---|---|---|---|---|---|
| ML_VPN1 | 3 | 0 | 0 | 26 | 1 | 27 | 0.0 | 100.0 | 96.3 | 3.7 | 1.0 | 0.0 | 1.8 |
| C2 | 2 | 7 | 13 | 6 | 0 | 26 | 76.9 | 23.1 | 50.0 | 50.0 | 1.0 | 1.7 | 0.4 |
| Mt_VPN | 4 | 0 | 12 | 3 | 8 | 23 | 52.2 | 47.8 | 13.0 | 87.0 | 0.8 | 1.0 | 0.7 |
| Mti | 3 | 2 | 0 | 10 | 7 | 19 | 10.5 | 89.5 | 63.2 | 36.8 | 0.7 | 0.2 | 1.1 |
| Tm5a-like | 1 | 0 | 1 | 0 | 16 | 17 | 5.9 | 94.1 | 0.0 | 100.0 | 0.6 | 0.1 | 1.1 |
| TmY10 | 1 | 1 | 0 | 6 | 0 | 7 | 14.3 | 85.7 | 100.0 | 0.0 | 0.3 | 0.1 | 0.4 |
| Mi10 | 1 | 0 | 0 | 0 | 5 | 5 | 0.0 | 100.0 | 0.0 | 100.0 | 0.2 | 0.0 | 0.3 |
| Mi | 1 | 2 | 1 | 0 | 0 | 3 | 100.0 | 0.0 | 66.7 | 33.3 | 0.1 | 0.3 | 0.0 |
| C3 | 1 | 0 | 0 | 0 | 3 | 3 | 0.0 | 100.0 | 0.0 | 100.0 | 0.1 | 0.0 | 0.2 |
| Identified_ < 3 | 26 | 5 | 14 | 14 | 4 | 37 | 51.4 | 48.6 | 51.4 | 48.6 | 1.4 | 1.6 | 1.2 |
| Unidentified_ ≥ 3 | 2 | 0 | 0 | 5 | 3 | 8 | 0.0 | 100.0 | 62.5 | 37.5 | 0.3 | 0.0 | 0.5 |
| Unidentified_ < 3 | 62 | 9 | 15 | 22 | 24 | 70 | 34.3 | 65.7 | 44.3 | 55.7 | 2.6 | 2.0 | 3.0 |
| Total | 211 | 628 | 559 | 747 | 773 | 2,707 | 43.8 | 56.2 | 50.8 | 49.2 | 100.0 | 100.0 | 100.0 |

**Table 3.** Cells that synapse onto seed columns R7 and R8.

| Type | No. | pR7 | yR7 | pR8 | yR8 | Sum | %R7 | %R8 | %p | %y | %Total | %Total_R7 | %Total_R8 |
|---|---|---|---|---|---|---|---|---|---|---|---|---|---|
| Dm9 | 6 | 90 | 92 | 79 | 72 | 333 | 54.7 | 45.3 | 50.8 | 49.2 | 57.0 | 51.3 | 65.9 |
| R8 | 4 | 75 | 82 | 1 | 0 | 158 | 99.4 | 0.6 | 48.1 | 51.9 | 27.1 | 44.2 | 0.4 |
| R7 | 4 | 0 | 0 | 31 | 26 | 57 | 0.0 | 100.0 | 54.4 | 45.6 | 9.8 | 0.0 | 24.9 |
| Mt_VPN | 1 | 0 | 2 | 0 | 3 | 5 | 40.0 | 60.0 | 0.0 | 100.0 | 0.9 | 0.6 | 1.3 |
| C2 | 1 | 2 | 3 | 0 | 0 | 5 | 100.0 | 0.0 | 40.0 | 60.0 | 0.9 | 1.4 | 0.0 |
| L3 | 1 | 0 | 0 | 4 | 0 | 4 | 0.0 | 100.0 | 100.0 | 0.0 | 0.7 | 0.0 | 1.7 |
| Identified_ < 3 | 11 | 4 | 2 | 4 | 8 | 18 | 33.3 | 66.7 | 44.4 | 55.6 | 3.1 | 1.7 | 5.2 |
| Unidentified_ ≥ 3 | 0 | 0 | 0 | 0 | 0 | 0 | 0.0 | 0.0 | 0.0 | 0.0 | 0.0 | 0.0 | 0.0 |
| Unidentified_ < 3 | 4 | 1 | 2 | 1 | 0 | 4 | 75.0 | 25.0 | 50.0 | 50.0 | 0.7 | 0.8 | 0.4 |
| Total | 32 | 172 | 183 | 120 | 109 | 584 | 60.8 | 39.2 | 50.0 | 50.0 | 100.0 | 100.0 | 100.0 |

Mi15 is the only local dopaminergic medulla cell type identified in *Drosophila* so far (*Meissner et al., 2019*; *Davis et al., 2020*). Its morphology and connectivity in the medulla have been described previously (*Takemura et al., 2013*; *Takemura et al., 2015*), and it has a long process that extends into the optic chiasm and tracks photoreceptor axons. Our reconstructions revealed that this process was a locus of photoreceptor input (*Figure 3Cii*, black arrow), and that Mi15 cells were R8-selective, with 91% of their seed column photoreceptor input drawn from R8. The Mi15 cells received 47% of their seed column photoreceptor input outside the medulla (mean of 15/32 synapses per column; *Figure 1—figure supplement 2G*), and R8s characteristically formed synapses with Mi15 cells at locations where they also made synapses with R7 (*Figure 3—figure supplement 1A*).

The columnar cell types Mi1, Mi4, and Mi9 are integral components of the ON-motion pathway (*Strother et al., 2017*); prior EM reconstructions have found they receive significant input from R7 and R8, a result confirmed by our data (*Takemura et al., 2013*; *Takemura et al., 2015*; *Figure 3—figure supplement 1B-D*). In the identified pale and yellow columns of the medulla-7-column connectome, Mi9 was selective for R8 (65% R8 versus 35% R7 input), and 53% of its photoreceptor input originated from pale R8 (*Takemura et al., 2015*). Our data reinforced the observation that Mi9 show a bias for R8 input: 68% of our reconstructed Mi9 input originated from pale seed column R8; the cells in pale columns received 15 and 14 R7–8 synapses, while those in yellow columns received 6 and 4 (*Figure 3—figure supplement 1D*).

## R7–8 synapses with cell types projecting to the lobula

The lobula receives diverse inputs from the medulla, including from cell types required for wavelength-specific phototaxis and learned color discrimination (*Gao et al., 2008*; *Karuppudurai et al., 2014*; *Melnattur et al., 2014*; *Otsuna et al., 2014*). Many of these cell types express the histamine receptor *ort*, notably the transmedulla (Tm) cell types Tm5a, Tm5b, Tm5c, and Tm20 (*Gao et al., 2008*), indicating they receive direct input from the histaminergic R7 or R8 (*Pantazis et al., 2008*). We found all of these Tm cell types and they were highly connected with the seed column R7–8, with a mean of 35 seed column photoreceptor inputs per cell for Tm20, 27 for Tm5a, 23 for Tm5c, and 24 for Tm5b (*Figure 4*, *Figure 4—figure supplement 1*).

The cholinergic Tm5a cell type was the first cell type described to show yellow versus pale selectivity, being preferentially innervated by yellow R7, as revealed by genetic methods and light microscopy (*Karuppudurai et al., 2014*). It was identified in the medulla-7-column connectome (*Menon et al., 2019*) and used for identifying putative yellow columns therein (see Materials and methods). Its medulla processes are centered on one yellow column, with a single main dendrite from which branches spread laterally reaching some of the neighboring columns in M3, M6, and M8, and an axon terminating in lobula layer 5B manifesting a characteristically hooked final reversal of direction (*Gao et al., 2008*; *Figure 4A*). In both our yellow seed columns there was a Tm5a cell, whose seed column photoreceptor input was exclusively from the R7 in the home column (*Figure 4Aiv*). Reassuringly, there were also no Tm5a cells centered on pale columns or innervated by pale photoreceptors. Thus,

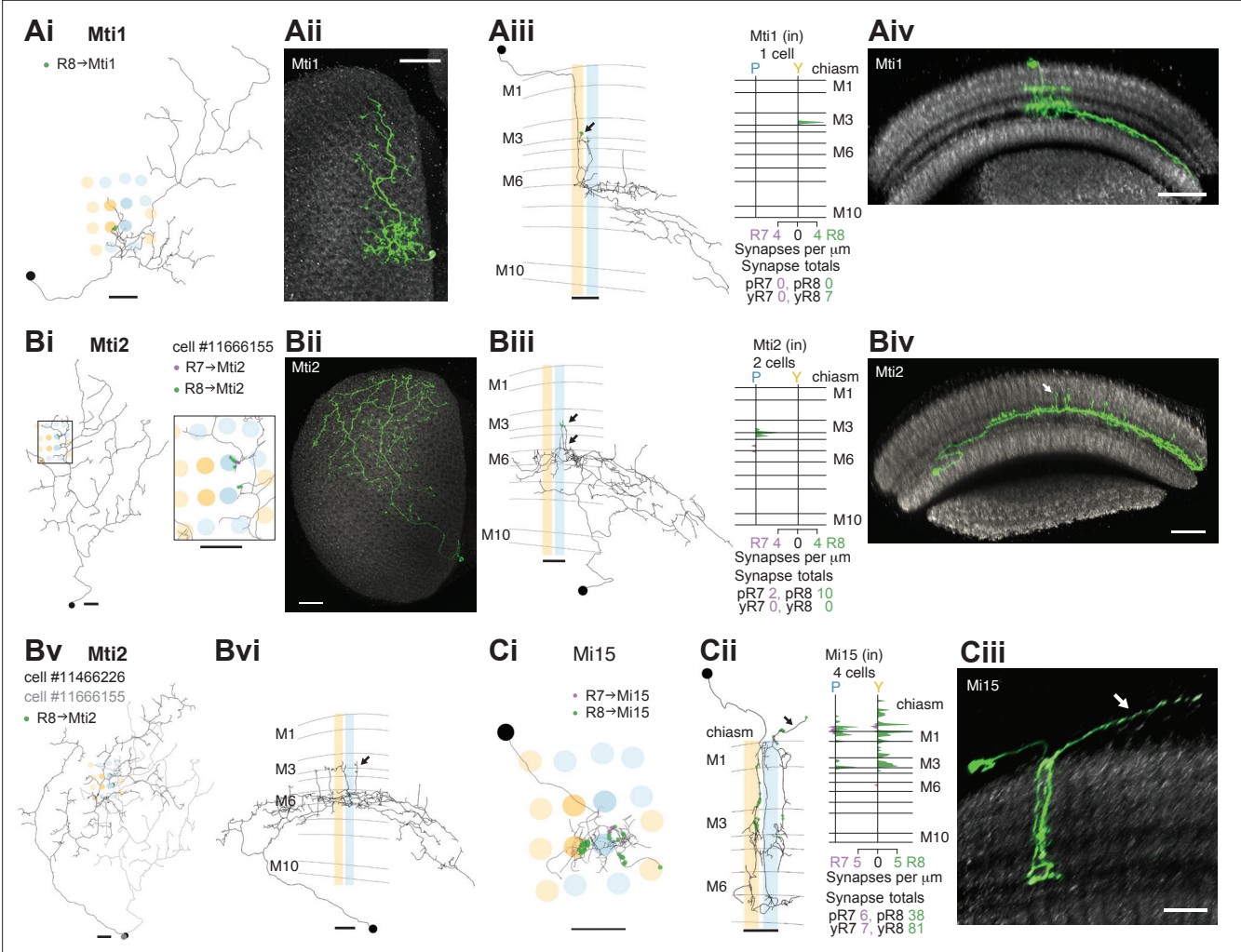

**Figure 3.** R7–8 synapses with medulla intrinsic (Mi) and medulla tangential intrinsic (Mti) cell types. (**A**) Synapses of central R7 and R8 with Mti1 cells. (Ai) Top view of an Mti1 skeleton with synapses from seed column R8 cells (green). Scale bar: 10 μm. (Aii) Top view of an MCFO-labeled cell matching overall Mti1 morphology. Light microscopy images (Aii, Aiv, Bii, Biv, Ciii) were manually segmented to focus on the cells of interest. Scale bar: 20 μm. (Aiii) Side view of the same Mti1 cell as in Ai with R8 (green) synapses. Scale bar: 10 μm. (Aiv) Side view rendering of the same light microscopy Mti1 image (green). Scale bar: 20 μm. (**B**) Synapses from R7 and R8 to Mti2. (Bi) Top view of a reconstructed Mti2 cell, with R7 (purple) and R8 (green) synapses. Scale bar: 10 μm. (Bii) Top view of an MCFO-labeled cell matching overall Mti2 morphology. Scale bar: 20 μm. (Biii) Side view of the same Mti2 skeleton as in Bi with R7 (purple) and R8 (green) synapses on characteristic vertical projections (arrow). Scale bar: 10 μm. (Biv) Side view of the cell shown in Bii. Arrow indicates vertical projections. Scale bar: 20 μm. (Bv) Top view of a second reconstructed Mti2 cell (dark gray) overlaid on the first (light gray), with R8 (green) synapses. Bvi: Side view of the second Mti2 skeleton, with R8 (green) synapses again located on vertical projections (arrow). (Bv–vi) Scale bars: 10 μm. (**C**) Synapses of R7 and R8 with Mi15 cells. (Ci) Top view of a fully reconstructed Mi15 skeleton, with R7 (purple) and R8 (green) synapses. (Cii) Left: Side view depicting the layer distribution of photoreceptor synapses (same color code) onto the same Mi15 skeleton, with synapses outside the medulla (arrow). (Ci–ii) Scale bars: 10 μm. Right: Layer distribution of R7 and R8 inputs to four Mi15 cells. (Ciii) Side view rendering of a light microscopy image (MCFO) of an Mi15 cell. Note the long process leaving the medulla (arrow). Scale bar: 10 μm.

The online version of this article includes the following figure supplement(s) for figure 3:

**Figure supplement 1.** Mi cell types targeted by R7–8.

the Tm5a cells supported the designation of our seed columns as pale or yellow, which was based on the presence or absence of aMe12 vertical processes. One more cell that we refer to as a Tm5a-like had the morphological features of a Tm5a cell, and was selective for yellow seed column input, but from R8 (*Figure 4—figure supplement 1A*). There was no cell with a similar morphology or connectivity in the medulla-7-column connectome, as explored using NeuPrint (*Clements et al., 2020*). The cell received its R8 input on three different dendritic branches, indicating that they were unlikely to have resulted from a reconstruction error (*Figure 4—figure supplement 1A* iv). Overall, our results

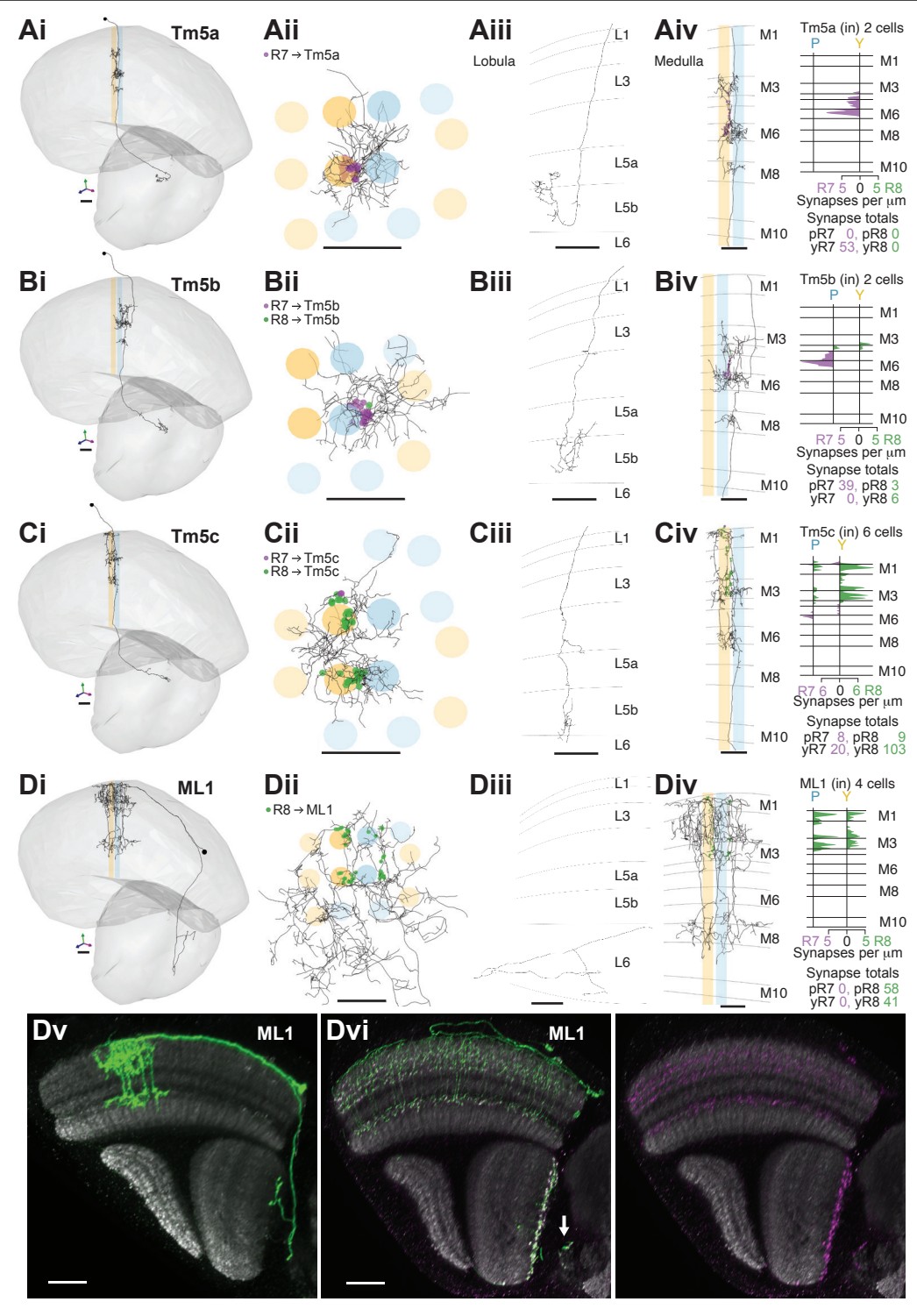

**Figure 4.** R7–8 synapses with cell types projecting to the lobula. (**A–D**) Synapses between R7 and R8 and Tm5a (**A**), Tm5b (**B**), Tm5c, (**C**) and ML1 (**D**) cell types. R7 synapses are indicated in purple, R8 synapses in green. Scale bars: 10 μm. Panels i–iii: Anatomy of fully reconstructed cells. (**i**) Side view. (**ii**) Top view. (**iii**) Side view of axon terminals in the lobula. (**iv**) Left: Side view of medulla branches. Right: Layer distribution of R7 and R8 inputs. (**A**) Tm5a. Two Tm5a cells were exclusively targeted by yellow R7 cells.( **B**) Tm5b. The two Tm5b cells were highly biased to pale R7 inputs. (**C**) Tm5c. The five Tm5c cells were highly biased to yellow R8 inputs. (**D**) ML1. Scale bars: 20 μm. (**Dv**) Side view of a single MCFO-labeled ML1 cell (light microscopy). Note this cell has terminals

*Figure 4 continued*

both at the base of the deepest lobula layer and in the central brain; not all examined ML1 cells have terminals in the central brain. (**Dvi**) Side view showing the distribution of a membrane marker (green) and presynaptic marker (synaptotagmin-HA, purple) in ML1 cells imaged as a population. The combined patterns (left) and the presynaptic marker alone (right) are shown. Arrow indicates central brain terminals.

The online version of this article includes the following figure supplement(s) for figure 4:

**Figure supplement 1.** Additional connections of R7–8 to cell types projecting to the lobula.

**Figure supplement 2.** ML1 arbor distribution in the medulla and lobula.

---

confirmed the selectivity of Tm5a cells for yellow columns and the reliability of finding R7-selective Tm5a cells in yellow columns.

The cholinergic Tm5b cell type is similar to Tm5a, but with ~2–3 vertical main dendrites spanning ~5 columns, from which branches spread laterally in M3, M6, and M8 (*Gao et al., 2008*; *Karuppudurai et al., 2014*; *Meinertzhagen et al., 2009*; *Figure 4B*). A recent analysis of the medulla-7-column connectome has proposed that Tm5b cells are pale-specific (*Menon et al., 2019*). Our reconstructions revealed two Tm5b cells that were innervated by pale R7, and not yellow R7 (*Figure 4B*), with only a minor input from seed column R8 (19 and 20 R7 inputs, versus 7 and 2 R8 inputs, respectively). We also identified three more cells with a morphology matching the Tm5b cell type, but innervated by predominantly yellow inputs, which we refer to as Tm5b-like (*Figure 4—figure supplement 1B*). In these yellow-specific cells, R8 input dominated, with 9.0 R8 synapses (mean, range 4–16), versus 4.7 R7 synapses (mean, range 3–7; *Figure 4—figure supplement 1B* iv). In particular, one Tm5b-like cell had >20 synapses, and innervated all four seed columns, and so was well covered in our reconstructions (*Figure 4—figure supplement 1B*).

The Tm5c cells are glutamatergic neurons, spanning ~8 columns with single vertical dendrite in the medulla, from which lateral branches spread out in M1, as well as M3 and M6, while the axons terminate near the boundary between lobula layers 5 and 6, often with a branch in layer 4 (*Gao et al., 2008*; *Karuppudurai et al., 2014*; *Meinertzhagen et al., 2009*; *Figure 4C*); one cell axon terminated deep in layer 6 (*Supplementary file 2* – Tm5c). Our reconstructions confirmed they were selective for R8, as indicated previously (*Takemura et al., 2013*; *Takemura et al., 2015*; *Karuppudurai et al., 2014*; *Figure 1—figure supplement 2D*). Six Tm5c cells were innervated by seed column photoreceptors, and they all received many synapses from multiple columns, ranging from 11 to 44, consistent with R7–8 input over a large spatial receptive field. One Tm5c was centered on our seed columns, with processes in all of them (*Figure 4Cii*), and only targeted by yellow seed column photoreceptors, predominantly R8. This bias toward yellow R8 inputs was maintained over the population (*Figure 4Civ*), yet there were two Tm5c cells with pale selectivity, and it remains possible that different Tm5c subtypes exist (*Supplementary file 1* – outgoing Tm5c). In the identified pale and yellow columns of the medulla-7-column connectome, there were two Tm5c cells reliably contacted by R7–8, with >10 synapses (*Takemura et al., 2015*; *Clements et al., 2020*). These two cells also showed a bias for yellow R8 input, with respectively 18 and 11 synapses from yellow R8 cells, versus 5 and 0 synapses from pale R8 and R7 combined.

The Tm20 cell type is a known columnar target of R8 (*Takemura et al., 2013*; *Takemura et al., 2015*; *Gao et al., 2008*), and the four Tm20 cells located in our seed columns were indeed selectively connected to R8 (*Figure 4—figure supplement 1C*, *Figure 1—figure supplement 2D*). The Tm20 and Tm5c cell types together received 140 synapses from seed column photoreceptors, making them two of the most targeted cell types (*Table 2*). We also found one TmY10 cell that received seven synapses from our seed column photoreceptors (*Supplementary file 1* – TmY10). In addition, we partially reconstructed 10 other Tm cells with a mean of 4.4 photoreceptor synapses per cell. We did not fully reconstruct these lightly innervated Tm cells; Tm subtypes may manifest subtle differences in morphology and connectivity that require multiple examples to distinguish (*Jagadish et al., 2014*; *Gao et al., 2008*; *Fischbach and Dittrich, 1989*), which was beyond the scope of our targeted reconstruction.

We found one more prominent cell type projecting to the lobula, which we refer to as the medulla-to-lobula cell type ML1 (*Figure 4D*). We completely reconstructed one cell and used light microscopy data to explore the anatomy of the population (*Figure 4Dv-–vi*), and we estimate that there are ~45 cells per optic lobe (counts of 44, 44, 45, and 47 from four optic lobes). These cells projected

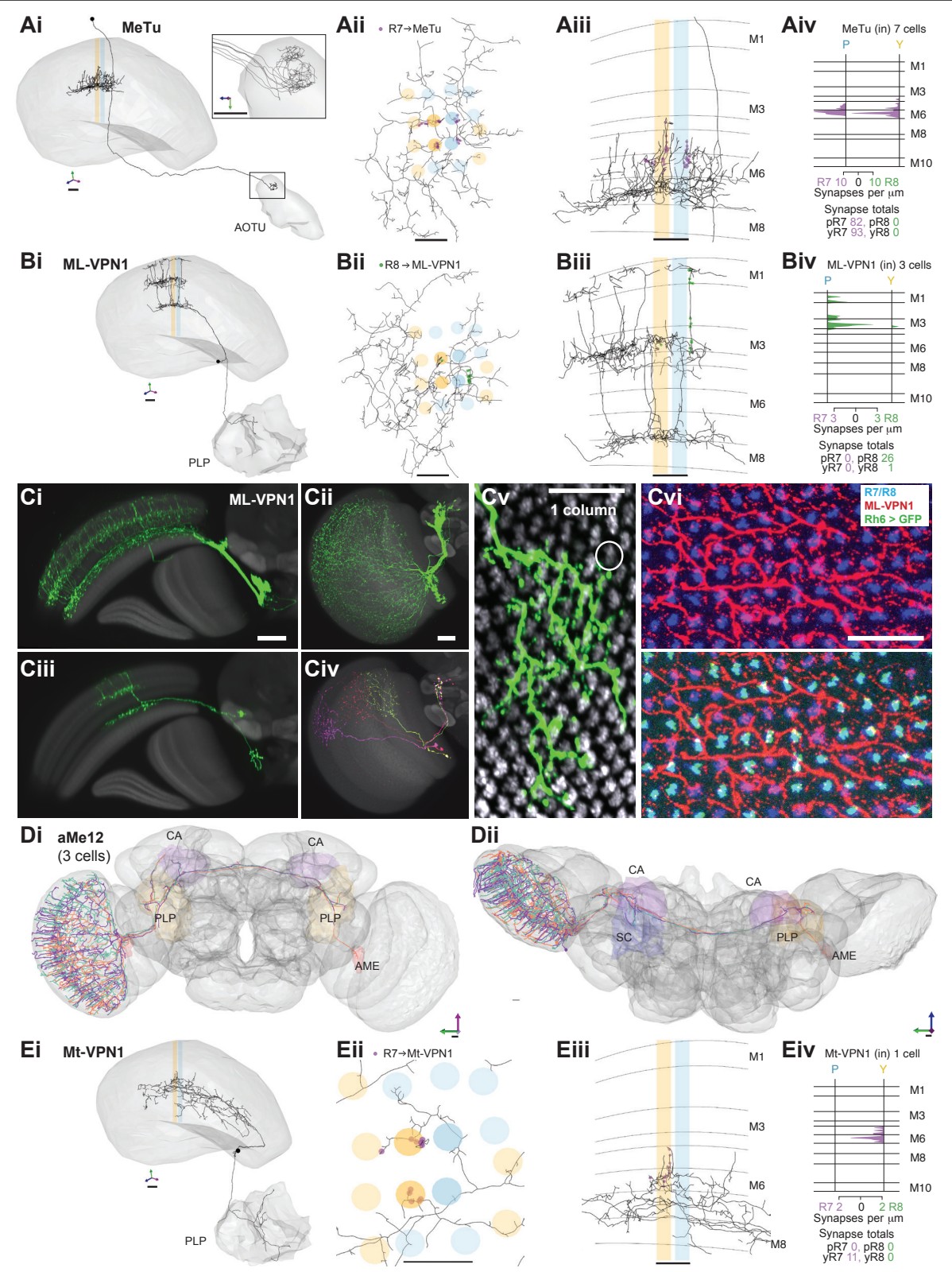

**Figure 5.** Visual projection neurons (VPN) connecting R7–8 with the central brain. (**A,B,E**) Synapses between R7 and R8 and MeTu (**A**), ML-VPN1 (**B**), and Mt-VPN1 (**E**) cells. Panels i–iii: Anatomy of reconstructed cells. (**i**) Side view. (**ii**) Top view. (**iii**) Side view of medulla branches. R7 synapses are indicated in purple, R8 synapses in green in ii, iii. (**iv**) Layer distribution of R7 and R8 inputs. Scale bars: 10 μm. (**A**) The MeTu cell shown was fully reconstructed. (**Ai**) Inset shows a magnified view of axon projections to the anterior optic tubercle (AOTU). (**Aiv**) All seven MeTu cells were exclusively R7 targets. (**B**)

*Figure 5 continued on next page*

_Figure 5 continued_

The ML-VPN1 cell shown was fully reconstructed. (Bi) ML-VPN1 cells project to the PLP. (Biv) Both ML-VPN1 cells were pale R8 targets. (Ci–vi) Light microscopy of ML-VPN1 anatomy. (Ci,Cii) Side view (Ci) and top view (Cii) of the population of ML-VPN1 cells. (Ciii) Side view of a single MCFO-labeled ML-VPN1 cells. (Civ) Top view of multiple MCFO-labeled ML-VPN1 cells. (Cv) Overlay of arbors of a single cell ML-VPN1 cell with L2 terminals (gray) indicating medulla columns. Images in Ci–Cv show overlays of aligned confocal images with the standard brain used for registration (Ci–iv) or a second registered image showing L2 terminals (Cv). (Cvi) Confocal substack projection showing medulla columns at approximately the level of R8 terminals. ML-VPN1 (red) and photoreceptor axons (blue) are shown without (top) and with (bottom) labeling of yellow R8 axons (Rh6> green). Overlap between ML1 and photoreceptors is largely limited to pale columns (i.e. columns without the Rh6 marker). (Ci–vi) Scale bars: 20 μm. (D) Reconstructions of three aMe12 cells (orange, turquoise, purple) covering the entire medulla, with axons leaving via the accessory medulla and innervating the mushroom body calyces (CA) and the PLP and accessory medulla (AME) of both hemispheres. (Di) Frontal view. (Dii) Dorsal view. (E) Mt-VPN1 cells. (Eiv) Tracing of Mt-VPN1 photoreceptor synapses in 16 additional columns did not confirm yellow specificity found in the seed columns.

The online version of this article includes the following figure supplement(s) for figure 5:

**Figure supplement 1.** Pale-specificity of ML-VPN1.

**Figure supplement 2.** All additional neurons connecting R7–8 to the central brain with ≥3 synapses.

to the deepest layer of the lobula and also to the adjacent central brain, with cell bodies in the anterior medulla cell body rind (_Figure 4Dv_). They were morphologically similar to a putative _ort_-expressing cell identified by _Gao et al., 2008_; (see their Figure S6). ML1 dendrites covered ~20 medulla columns with overlap, and the population covered the medulla (_Figure 4Dii_,vi), ramifying from vertical processes in layers M1-4, and also in M8 (_Figure 4Diii_,v). Unlike the Tm cells, the ML1 axons exited the distal surface of the medulla and traveled anteriorly to enter and terminate in or near lobula layer 6 (_Figure 4Diii_, v, vi), providing an alternate pathway connecting the medulla to the lobula. Light microscopy suggests that a fraction of the cells also formed synapses in the central brain, in the posterior lateral protocerebrum (PLP) (_Figure 4Dvi_, arrow). The input to the lobula was non-columnar, and although it was not strictly retinotopically organized, it was spatially organized, with two axon bundles originating in the dorsal and ventral medulla terminated in two general locations, incompletely covering the lobula (_Figure 4—figure supplement 2_). Our seed columns contacted four ML1 cells, making an average of 25 synapses per ML1 cell. The cell was exclusively targeted by R8s from both pale and yellow columns (_Figure 4Div_).

## Visual projection neurons connecting R7–8 with the central brain

Nearly 10% of the synapses made by the central seed column R7–8 were with cells projecting directly to the central brain (_Table 2_). Of these, the strongest targets were the MeTu cells, which resemble Tm cells, but instead of sending axons to the lobula, projected to the AOTU via the anterior optic tract (_Fischbach and Lyly-Hünerberg, 1983_; _Otsuna et al., 2014_; _Otsuna and Ito, 2006_; _Omoto et al., 2017_; _Timaeus et al., 2020_; _Tai et al., 2021_; _Figure 5A_). Our reconstructions confirmed that MeTu cells were targets of R7 (_Timaeus et al., 2020_), and that their photoreceptor input was exclusively from R7 (_Figure 5Aiv_). In total MeTu cells received 15% of all the seed column R7 synapses (_Table 2_). We found seven such neurons and reconstructed the complete morphology of one. Their dendrites overlapped and covered ~20–30 columns (_Figure 5Aii_), while the axons all targeted the lateral tip of the AOTU (_Timaeus et al., 2020_; _Omoto et al., 2017_; _Tai et al., 2021_; _Figure 5Ai_).

The presence of MeTu cells was the largest discrepancy by synapse count between our list of R7 targets and those of the medulla-7-column connectome (_Clements et al., 2020_; _Takemura et al., 2015_; _Figure 1—figure supplement 2E_), which was focused on columnar cell types and did not identify MeTu cells. In the medulla-7-column connectome, we identified a partially reconstructed Tm cell (_Figure 1—figure supplement 2F_) and several cells annotated as Dm7 as potential matches to MeTu cells. We did not identify any Dm7 cells postsynaptic to R7 in our data. We confirmed details of the connectivity of our MeTu neurons that were found for the medulla-7-column connectome putative Dm7 neurons (including strong inputs from Dm2 and Mi15 neurons), and we propose that the partially reconstructed Tm cells and also Dm7 cells in the 7-column data set were most likely MeTu cells, resolving the largest (numerical) discrepancy between the data sets, and underscoring the benefit of reconstructing small circuits in whole-brain EM volumes.

Among the photoreceptor target cells projecting to the central brain, we identified a second cell type whose axonal projections follow the anterior axonal tracts of the ML1 neuron, which we refer to here as ML-VPN1 (_Figure 5B_). The cell bodies were located near the anterior proximal medulla.

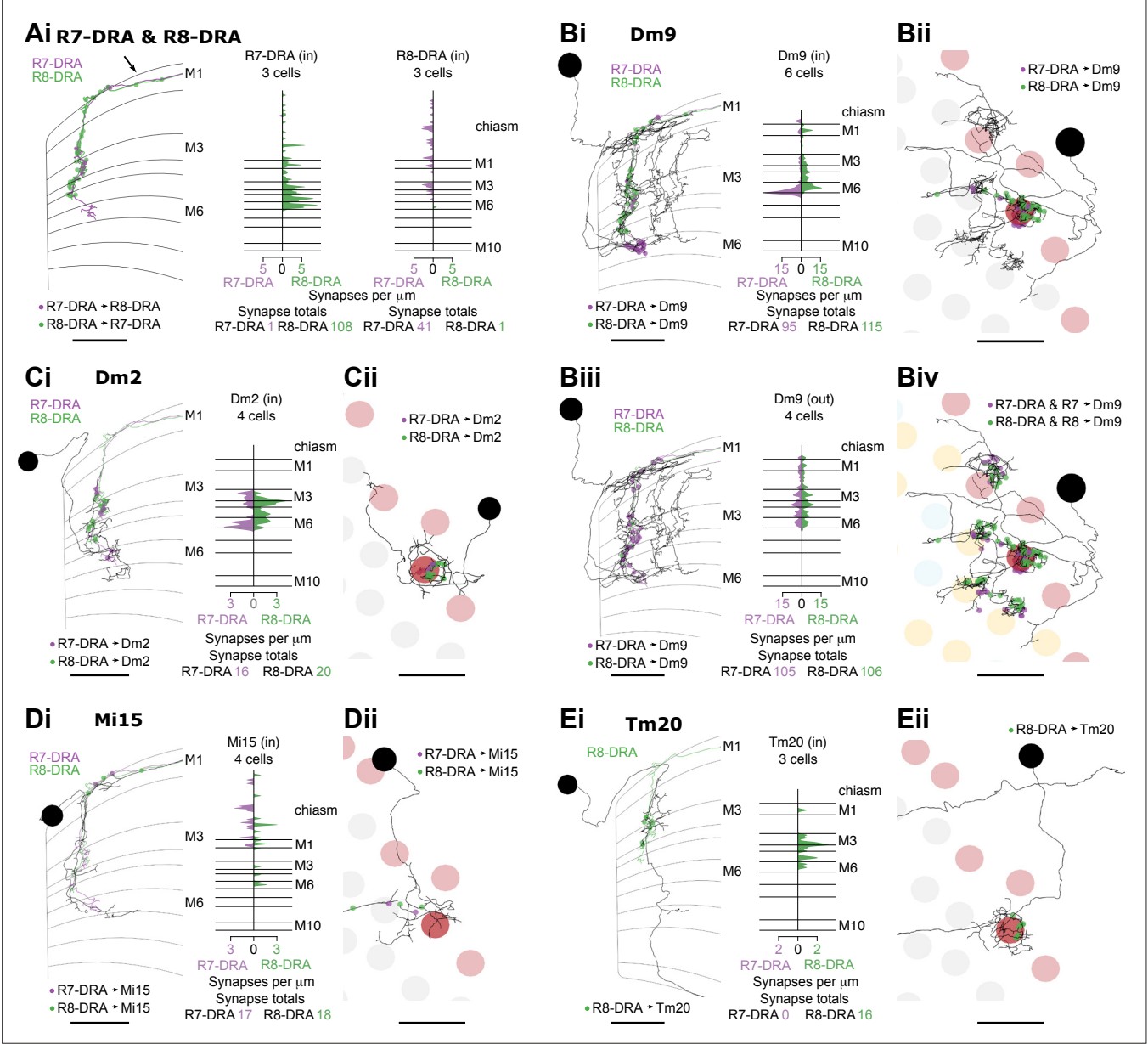

**Figure 6.** Synapses between R7-dorsal rim area (DRA) and R8-DRA, and with medulla cell types. (**A**) Reciprocal synapses between R7-DRA and R8-DRA. Left: Side view of an R7- and R8-DRA cell with R7-DRA→R8-DRA (purple points) and R8-DRA→R7-DRA (green points). Right: Layer distribution of R7-DRA and R8-DRA reciprocal synapses in three columns. (**B**) Reciprocal synapses between Dm9 in the DRA region and R7-DRA and R8-DRA. (Bi) Left: Side view of a reconstructed Dm9 skeleton (gray) in the DRA region, with R7-DRA→Dm9 synapses (purple) and R8-DRA→Dm9 synapses (green). Right: Layer distribution of DRA photoreceptor inputs into Dm9. (Bii) Top view of Dm9 cell that connects to photoreceptors in both DRA (light red) and non-DRA columns (gray). (Biii) Left: Feedback synapses from Dm9 to R7-DRA (purple) and R8-DRA (green). Right: Layer distribution of Dm9 inputs into R7-DRA and R8-DRA. (Biv) DRA and non-DRA R7 (purple) and R8 (green) synapses to the same Dm9 cell. (**C**) Synapses between R7-DRA, R8-DRA, and Dm2. (Ci) Left: Side view of a Dm2 cell with synapses from R7-DRA (purple) and R8-DRA (green). Right: Layer distribution of R7-DRA and R8-DRA synapses onto four Dm2 cells. (Cii) Top view of Dm2 skeleton in the DRA region. (**D**) Synapses between R7-DRA, R8-DRA, and Mi15. (Di) Left: Side view of a reconstructed Mi15 cell with R7-DRA→Mi15 synapses (purple) and R8-DRA→Mi15 synapses (green). Right: Layer distribution of R7-DRA and R8-DRA synapses to four Mi15 cells. (Dii) Top view of Mi15 skeleton in the DRA region. (**E**) Synapses between R7-DRA, R8-DRA, and Tm20. (Ei) Left: Side view of a reconstructed Tm20 cell with R8-DRA→Tm20 synapses in green. Right: Layer distribution of R8-DRA input to three Tm20 cells. (Eii) Top view of Tm20 skeleton in the DRA region (all scale bars: 10 μm).

The online version of this article includes the following figure supplement(s) for figure 6:

**Figure supplement 1.** Synapses between inner photoreceptors in both the central and dorsal rim area (DRA) columns.

**Figure supplement 2.** Additional cell types connected to R7-dorsal rim area (DRA) and R8-DRA.

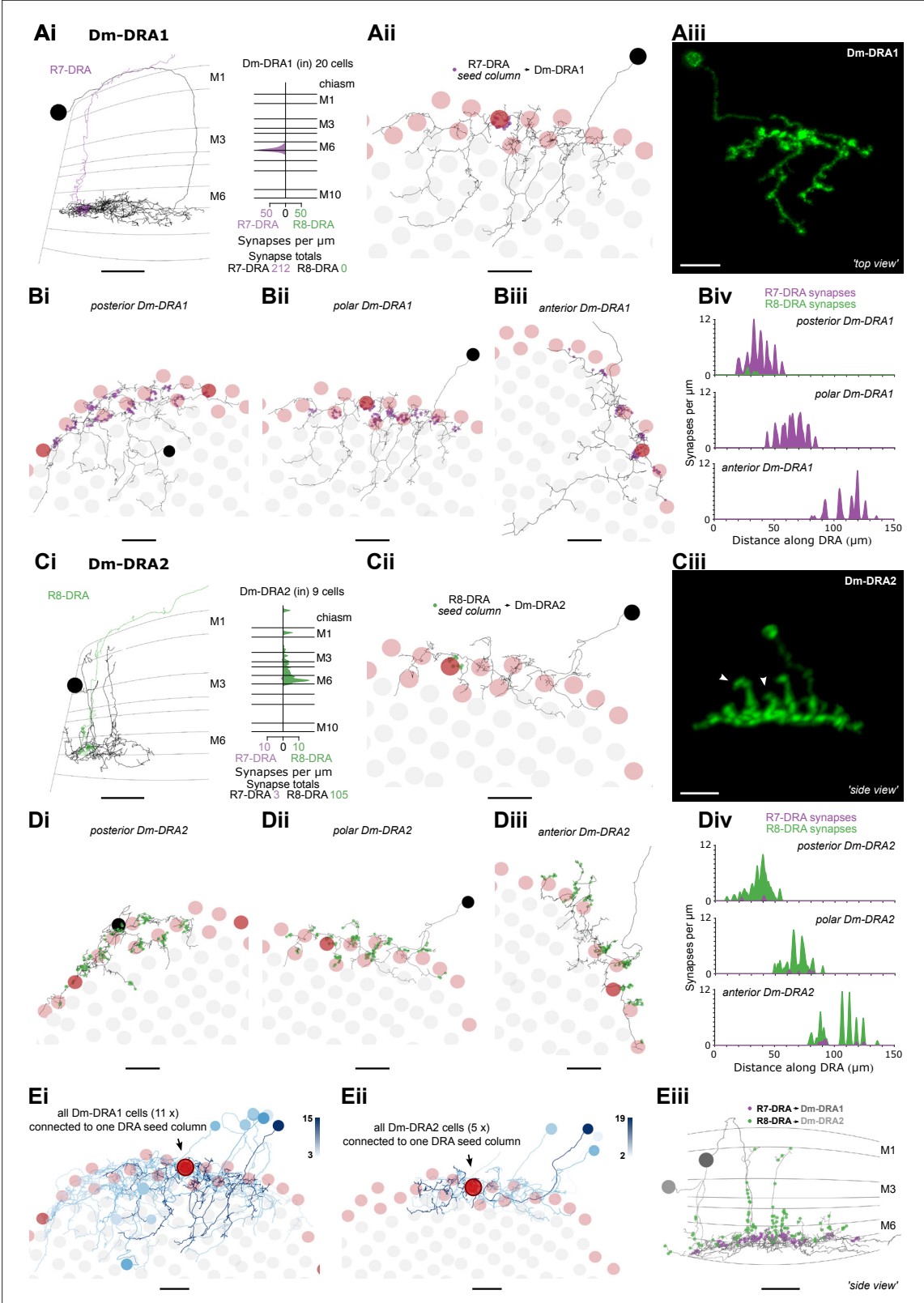

**Figure 7.** Dm8-like photoreceptor targets in the dorsal rim area (DRA) region. (**A**) Synapses between photoreceptors and Dm-DRA1 cells. (Ai) Left: Side view of a reconstructed Dm-DRA1 (gray) innervated by R7-DRA (purple). Right: Distribution of R7-DRA synapses (purple) onto 20 Dm-DRA1 cells, plotted across medulla layers. (Aii) Top view of the same fully reconstructed Dm-DRA1 skeleton and R7-DRA inputs from the (polar) seed column. (Aiii) Top view of a light microscopic Dm-DRA1 single cell clone with processes leaving the DRA region. (**B**) R7-DRA inputs into Dm-DRA1 cells. (Bi–iii) Three

*Figure 7 continued on next page*

*Figure 7 continued*

skeletons of fully reconstructed Dm-DRA1 cells (gray) at different positions along the DRA (posterior, polar, and anterior) with all R7-DRA synapses (purple) originating from an average of 11 columns. (Biv) Distribution of R7-DRA synapses onto the three Dm-DRA1 cells from Bi–iii along the DRA region. (**C**) Synapses between photoreceptors and Dm-DRA2 cells. (Ci) Left: Side view of one reconstructed Dm-DRA2 skeleton (gray) innervated by R8-DRA (green). Right: Layer distribution of DRA photoreceptor synapses onto 9 Dm-DRA2 cells. (Cii) Top view of the same fully reconstructed Dm-DRA2 skeleton and photoreceptor inputs from the (polar) seed column. (Ciii) Side view of a light microscopic Dm-DRA2 single cell clone with vertical processes (arrows in Ci and Ciii indicate vertical projections). (**D**) Photoreceptor inputs to Dm-DRA2 cells. (Di–iii) Three skeletons of fully reconstructed Dm-DRA2 cells (gray) at different positions along the DRA (posterior, polar, and anterior) with all R8-DRA (green) synapses originating from an average of 11 columns. (Div) Distribution of R8-DRA (and few R7-DRA) synapses onto the three Dm-DRA2 cells from Di–iii along the DRA region. (**E**) Comparison of Dm-DRA1 and Dm-DRA2 connectivity. All reconstructed Dm-DRA1 skeletons connected to R7-DRA in the polar seed column (circled column, arrow). The saturation of blue color indicates strength of connectivity (from 3 to 15 synapses). (Eii) All Dm-DRA2 skeletons connected to R8-DRA in the polar seed column (circled column, arrow), blue color indicates the strength of connectivity (from 2 to 19 synapses). (Eiii) Side views of overlapping Dm-DRA1 and Dm-DRA2 cell skeletons with all R7-DRA and R8-DRA synapses (same cells as in Bii and Dii) (all scale bars: 10 µm).

The online version of this article includes the following figure supplement(s) for figure 7:

**Figure supplement 1.** Non-dorsal rim area (DRA) photoreceptor inputs to Dm-DRA cells.

**Figure supplement 2.** Deep projections of Dm-DRA1 cells.

**Figure supplement 3.** Fitting a linear distance of synapses along the dorsal rim area (DRA) region.

In our match using light microscopic data, the dendrites of individual neurons overlap, with each cell spanning tens of columns, collectively covering the medulla (***Figure 5Ci–vi***), and we estimate ~65 cell bodies per optic lobe (counts of 64, 64, 65, and 68 from four optic lobes). The dendrites ramified along the border of M7 and M8, with vertical processes reaching up and spreading laterally in M3 and again vertically up to M1 (***Figure 5Bii–iii and Ciii***). Although the ML-VPN1 axons followed the ML1 axonal tract, they innervated the PLP of the ipsilateral hemisphere, in a region just posterior to the optic glomeruli (***Figure 5Bi***, Ci). The ML-VPN1 cells were almost exclusively targeted by the pale R8, with 8.7 seed pale R8 inputs per cell (mean, range 5–14), and just one yellow R8 input across the three cells connected to our seed column photoreceptors (***Figure 5Biv***). Light microscopy of co-labeled ML-VPN1 neurons and either yellow R8 axons (labeled for Rh6 in ***Figure 5Cvi***) or pale R8 axons (labeled for Rh5 in ***Figure 5—figure supplement 1***) supported the pale preference seen in the EM data: ~90% of columns in which ML1 arbors and photoreceptors appeared to overlap were pale. Together, our data indicate that ML-VPN1 neurons have a strong bias for pale R8 input.

The aMe12 cells were also R8 targets, and the two contacted cells received 15 and 11 R8 synapses, and 2 and 2 R7 synapses, respectively, in the seed columns (***Figure 1D and Eiii***, ***Supplementary file 1*** – aMe12 outgoing). Their axons also targeted the PLP, although in different locations to ML-VPN1, but they did so in both hemispheres (***Figure 5D***). In addition, aMe12 cells also projected to the mushroom body calyces of both hemispheres, as well as to the contralateral and ipsilateral accessory medulla and superior clamp (***Figure 5Di***, Dii). All three aMe12 cells shared a characteristic branching pattern, but only one of them additionally innervated the contralateral accessory medulla (***Figure 5Dii***). The aMe12 cell type is present in the hemibrain connectome, where it is also a strong input to γd Kenyon cells (***Li et al., 2020***; ***Scheffer et al., 2020***).

There were five additional tangential cells projecting to the central brain (medulla tangential [Mt]-VPNs) targeted by seed column R7–8, collectively receiving 25 synapses (one neuron in ***Figure 5E***, others in ***Figure 5—figure supplement 2***). The most targeted cell, which we refer to as Mt-VPN1, had processes in layer M7, with branches reaching up into layers M5–M6, where it received inputs from both seed column yellow R7s, and projected an axon to the PLP (***Figure 5E***). Despite the exclusive seed connections with yellow R7, Mt-VPN1 was not specific for yellow R7: we traced Mt-VPN1 in 20 columns, in which it received input from 13 yellow R7, and 7 pale R7 cells. The remaining tangential cells included a match to an octopaminergic neuron (OA-AL2i3; ***Busch et al., 2009***; ***Figure 5—figure supplement 2A***). This cell had processes in the distal medulla in layers M1–M2 as well as processes reaching out into the chiasm which received photoreceptor input (***Figure 5—figure supplement 2A*** ii-iii), and an axon that ipsilaterally innervated the inferior and superior posterior slope, and the flange (***Figure 5—figure supplement 2A*** i). This cell was one of the rare cell types that formed synapses onto photoreceptors, making 5 synapses onto the R7-R8, but only in one yellow column (***Supplementary file 1*** – Mt-VPN). The remaining Mt-VPN cells were very lightly innervated by the seed column photoreceptors in multiple (***Figure 5—figure supplement 2C*** ii) and single columns (***Figure 5—figure***

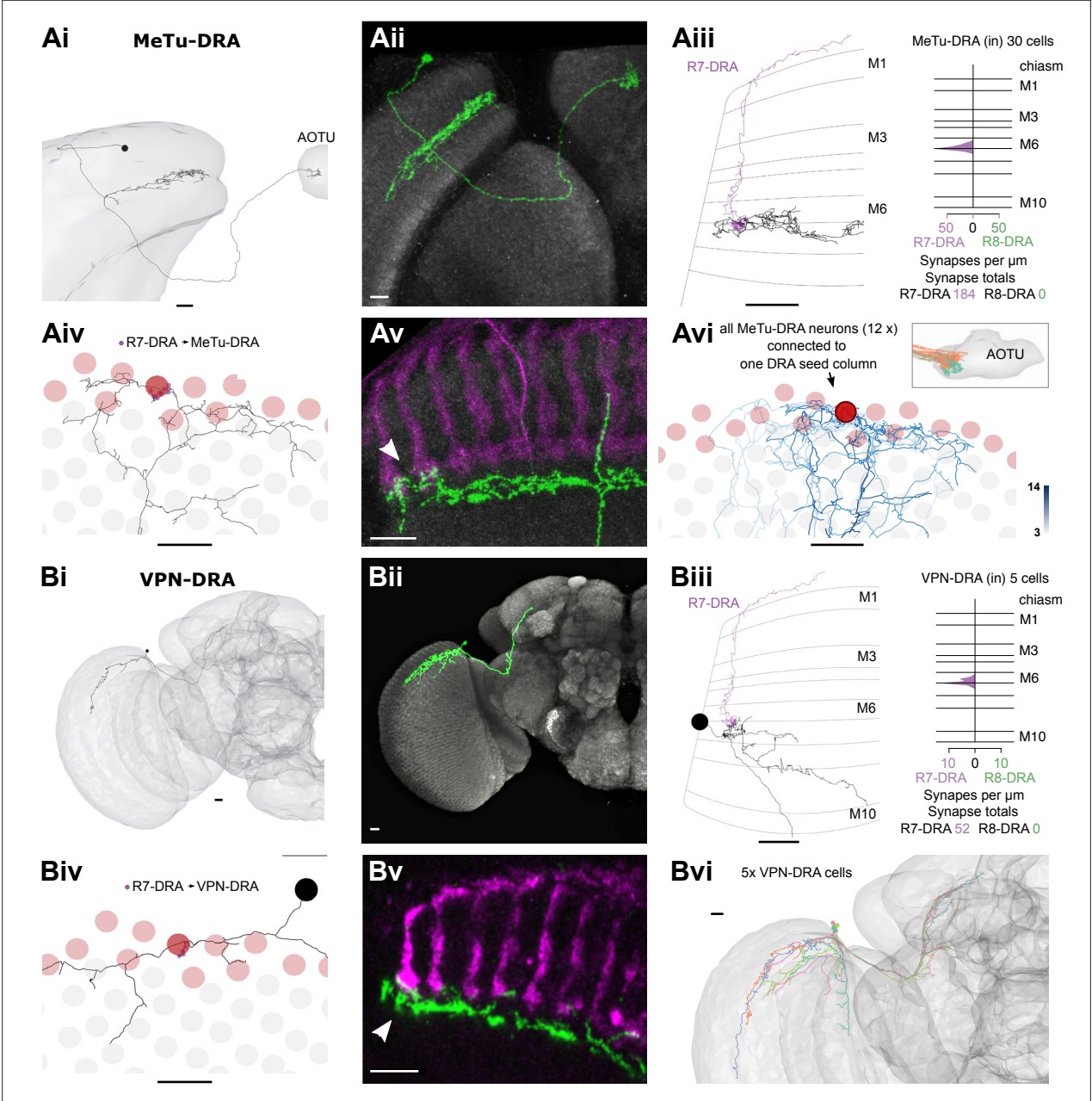

**Figure 8.** Visual projection neurons (VPNs) connecting R7-dorsal rim area (DRA) with the central brain. (**A**) Synapses of R7-DRA onto MeTu-DRA cells. (Ai) The complete skeleton of a fully reconstructed MeTu-DRA cell with an axon projecting to the anterior optic tubercle (AOTU). (Aii) Light microscopic single cell clone of an MeTu-DRA cell. (Aiii) Left: Side view of the same MeTu-DRA skeleton (gray) as in Ai innervated by R7-DRA (purple). Right: Layer distribution of R7-DRA synapses (purple) onto 30 MeTu-DRA cells. (Aiv) Top view of the MeTu-DRA skeleton depicting its medulla processes innervating both DRA (red circles) and non-DRA columns. (Avi) Light microscopic side view of an MeTu-DRA single cell clone (green) with exclusive contacts to DRA photoreceptor terminals (white arrowhead). (Avi) All MeTu-DRA skeletons connected to R7-DRA in the polar seed column (circled column, arrow). The saturation of blue color indicates the strength of connectivity (from 3 to 14 synapses). Inset: MeTu-DRA axon terminations in the AOTU in orange and reconstructed MeTu cells from the central columns in cyan. (**B**) Synapses between R7-DRA and VPN-DRA. (Bi) The entire skeleton of a reconstructed VPN-DRA cell with its axon projecting to the PLP. (Bii) Light microscopic image of a VPN-DRA single cell clone shown in green and neuropil reference in gray (Nc82). (Biii) Left: Side view of one VPN-DRA skeleton (gray) innervated by R7-DRA (purple). Right: Layer distribution of R7-DRA synapses onto fiveVPN-DRA cells. (Biv) Top view of medulla processes formed by one reconstructed VPN-DRA skeleton (gray) with all R7-DRA synapses from the seed column (purple). (Bv) Double labeling of several VPN-DRA cells (green) with R7 and R8 photoreceptors (purple, anti-Chaoptin). VPN-DRA processes overlap with DRA photoreceptors (white arrowhead) but also appear to show some contacts to non-DRA photoreceptors. (Bvi) Skeletons of all reconstructed VPN-DRA cells covering the dorsal medulla (all scale bars: 10 µm).

The online version of this article includes the following figure supplement(s) for figure 8:

**Figure supplement 1.** Visual projection neuron (VPN) morphology in the dorsal rim area (DRA) region.

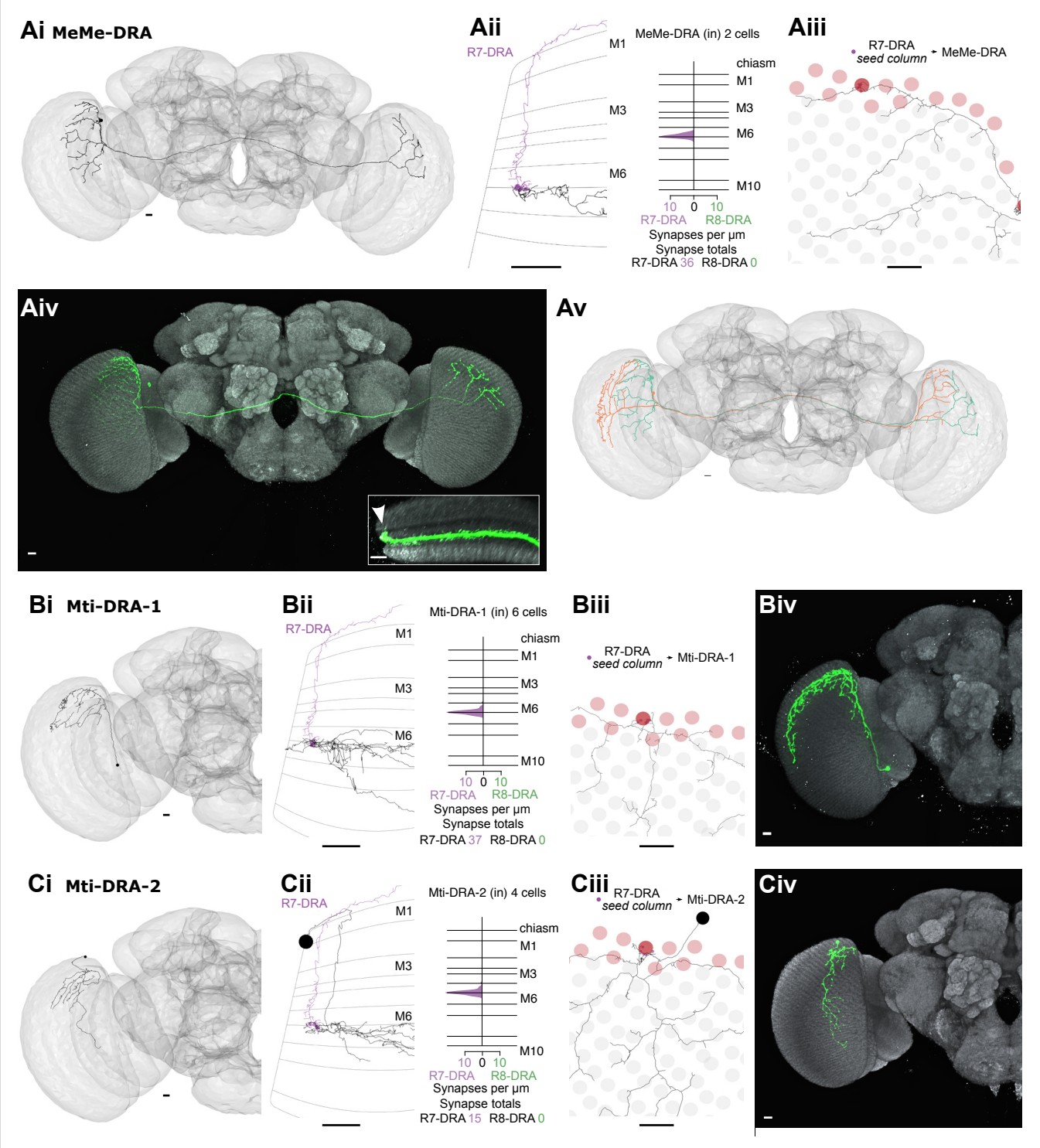

**Figure 9.** Other newly identified photoreceptor targets in the dorsal rim area (DRA) region. (**A**) Synapses from R7-DRA to bilaterally projecting MeMe-DRA neurons. (Ai) A reconstructed MeMe-DRA cell with projections to the dorsal periphery of the contralateral medulla. (Aii) Left: Side view of one MeMe-DRA skeleton (gray) innervated by R7-DRA (purple). Right: Layer distribution of R7-DRA synapses onto two MeMe-DRA cells. (Aiii) Top view of medulla processes of the MeMe-DRA skeleton (gray) with all R7-DRA synapses from the seed columns (purple). (Aiv) Light microscopic image of a MeMe-DRA single cell clone shown (in green) with processes to the contralateral medulla (neuropil reference in gray, Nc82). Inset: Medulla cross section showing branches of a MeMe-DRA cell. Note that the most dorsal arbors innervate the distal medulla (arrowhead). (Av) Two reconstructed MeMe-DRA skeletons, both with cell bodies on the left side (cyan and orange), connecting both medullas in a reciprocal manner, innervating opposite DRA regions along the anterior-posterior axis. (**B**) Synapses from R7-DRA onto Mti-DRA-1. (Bi) A reconstructed Mti-DRA-1 cell with processes covering the dorsal

*Figure 9 continued on next page*

*Figure 9 continued*

periphery of the medulla. (Bii) Left: Side view of one Mti-DRA-1 skeleton (gray) innervated by R7-DRA (purple). Right: Layer distribution of R7-DRA synapses onto six Mti-DRA-1 cells. (Biii) Top view of medulla processes of one reconstructed Mti-DRA-1 cell (gray) with all R7-DRA synapses from the (polar) seed column (purple). (Biv) Light microscopic image of a putative morphological single cell match of Mti-DRA-1 shown in green. (C) Synapses from R7-DRA onto Mti-DRA-2. (Ci) A reconstructed Mti-DRA-2 cell with processes covering dorsal parts of the medulla. (Cii) Left: Side view of one Mti-DRA-2 skeleton (gray) innervated by R7-DRA (purple). Right: Layer distribution of R7-DRA synapses onto four Mti-DRA-2 cells. (Ciii) Top view of medulla processes formed by one reconstructed Mti-DRA-2 cell (gray) with all R7-DRA synapses from the (polar) seed columns (purple). (Cv) Light microscopic image of a putative morphological single cell match of Mti-DRA-2 shown in green (all scale bars: 10 μm).

The online version of this article includes the following figure supplement(s) for figure 9:

**Figure supplement 1.** Mti cells in the dorsal rim area (DRA) region.

**Table 4.** Synaptic targets of seed column R7-dorsal rim area (DRA) and R8-DRA.

| Type | No. | DRAR7 | DRAR8 | Sum | %R7 | %R8 | %Total | %Total_R7 | %Total_R8 |
|---|---|---|---|---|---|---|---|---|---|
| Dm-DRA1 | 20 | 212 | 0 | 212 | 100.0 | 0.0 | 15.9 | 25.3 | 0.0 |
| Dm9 | 6 | 95 | 115 | 210 | 45.2 | 54.8 | 15.7 | 11.3 | 23.0 |
| MeTu_DRA | 29 | 181 | 0 | 181 | 100.0 | 0.0 | 13.5 | 21.6 | 0.0 |
| R7_DRA | 3 | 1 | 108 | 109 | 0.9 | 99.1 | 8.2 | 0.1 | 21.6 |
| Dm-DRA2 | 9 | 3 | 103 | 106 | 2.8 | 97.2 | 7.9 | 0.4 | 20.6 |
| Dm2 | 4 | 21 | 27 | 48 | 43.8 | 56.2 | 3.6 | 2.5 | 5.4 |
| R8_DRA | 3 | 41 | 1 | 42 | 97.6 | 2.4 | 3.1 | 4.9 | 0.2 |
| Mi15 | 4 | 21 | 18 | 39 | 53.8 | 46.2 | 2.9 | 2.5 | 3.6 |
| Mti_DRA_1 | 6 | 37 | 0 | 37 | 100.0 | 0.0 | 2.8 | 4.4 | 0.0 |
| MeMe_DRA | 2 | 36 | 0 | 36 | 100.0 | 0.0 | 2.7 | 4.3 | 0.0 |
| L3 | 3 | 16 | 16 | 32 | 50.0 | 50.0 | 2.4 | 1.9 | 3.2 |
| VPN_DRA | 6 | 30 | 0 | 30 | 100.0 | 0.0 | 2.2 | 3.6 | 0.0 |
| L1 | 3 | 9 | 9 | 18 | 50.0 | 50.0 | 1.3 | 1.1 | 1.8 |
| Tm20 | 3 | 0 | 16 | 16 | 0.0 | 100.0 | 1.2 | 0.0 | 3.2 |
| Mti_DRA_2 | 4 | 15 | 0 | 15 | 100.0 | 0.0 | 1.1 | 1.8 | 0.0 |
| Mi1 | 3 | 6 | 9 | 15 | 40.0 | 60.0 | 1.1 | 0.7 | 1.8 |
| Tm5-like | 1 | 0 | 12 | 12 | 0.0 | 100.0 | 0.9 | 0.0 | 2.4 |
| Mi9 | 2 | 0 | 12 | 12 | 0.0 | 100.0 | 0.9 | 0.0 | 2.4 |
| MeTu | 2 | 12 | 0 | 12 | 100.0 | 0.0 | 0.9 | 1.4 | 0.0 |
| Dm11 | 1 | 4 | 8 | 12 | 33.3 | 66.7 | 0.9 | 0.5 | 1.6 |
| MeTu_unknown | 1 | 4 | 0 | 4 | 100.0 | 0.0 | 0.3 | 0.5 | 0.0 |
| aMe12 | 1 | 3 | 1 | 4 | 75.0 | 25.0 | 0.3 | 0.4 | 0.2 |
| TmY | 1 | 0 | 3 | 3 | 0.0 | 100.0 | 0.2 | 0.0 | 0.6 |
| ML_VPN2 | 1 | 3 | 0 | 3 | 100.0 | 0.0 | 0.2 | 0.4 | 0.0 |
| C2 | 1 | 2 | 1 | 3 | 66.7 | 33.3 | 0.2 | 0.2 | 0.2 |
| Identified_ < 3 | 33 | 34 | 14 | 48 | 70.8 | 29.2 | 3.6 | 4.1 | 2.8 |
| Unidentified_ ≥ 3 | 2 | 5 | 5 | 10 | 50.0 | 50.0 | 0.7 | 0.6 | 1.0 |
| Unidentified_ < 3 | 57 | 46 | 21 | 67 | 68.7 | 31.3 | 5.0 | 5.5 | 4.2 |
| Total | 211 | 837 | 499 | 1,336 | 62.6 | 37.4 | 100.0 | 100.0 | 100.0 |

**Table 5.** Cells that synapse onto seed column R7-dorsal rim area (DRA) and R8-DRA.

| Type | No. | DRAR7 | DRAR8 | Sum | %R7 | %R8 | %Total | %Total_R7 | %Total_R8 |
|------|-----|-------|-------|-----|-----|-----|--------|-----------|-----------|
| Dm9 | 4 | 105 | 106 | 211 | 49.8 | 50.2 | 54.7 | 46.1 | 67.1 |
| R8_DRA | 3 | 108 | 1 | 109 | 99.1 | 0.9 | 28.2 | 47.4 | 0.6 |
| R7_DRA | 3 | 1 | 41 | 42 | 2.4 | 97.6 | 10.9 | 0.4 | 25.9 |
| C2 | 2 | 4 | 4 | 8 | 50.0 | 50.0 | 2.1 | 1.8 | 2.5 |
| Mi15 | 1 | 2 | 2 | 4 | 50.0 | 50.0 | 1.0 | 0.9 | 1.3 |
| Identified_ < 3 | 9 | 6 | 3 | 9 | 66.7 | 33.3 | 2.3 | 2.6 | 1.9 |
| Unidentified_ ≥ 3 | 0 | 0 | 0 | 0 | 0.0 | 0.0 | 0.0 | 0.0 | 0.0 |
| Unidentified_ < 3 | 2 | 2 | 1 | 3 | 66.7 | 33.3 | 0.8 | 0.9 | 0.6 |
| Total | 24 | 228 | 158 | 386 | 59.1 | 40.9 | 100.0 | 100.0 | 100.0 |

supplement 2Bii, Dii), but may integrate large numbers of photoreceptor inputs as they covered many columns. Collectively they innervated the ipsilateral superior posterior slope, accessory medulla, and lobula, and in both hemispheres the inferior posterior slope and lobula plate (*Figure 5—figure supplement 2B-D*), revealing multiple pathways for transmitting direct photoreceptor signals into the central brain.

## Synapses between R7-DRA and R8-DRA, and with medulla cell types

The following four sections present the first EM connectomic data for targets of the polarization-sensitive photoreceptors in the DRA of *Drosophila*. We first describe the cell types targeted by photoreceptors in the DRA region that were also found in the central columns (*Figure 6*), followed by modality-specific Dm cell types connected exclusively to DRA inner photoreceptors (*Figure 7*), VPNs connecting the DRA region and central brain (*Figure 8*), and finally cell types only found in the DRA region (*Figure 9*).

The anatomical arrangement of synapses between R7-DRA and R8-DRA strikingly resembled that of central R7 and R8. The number of R7-DRA synapses onto R8-DRA was similar (13.7 mean, range 10–16), accounting for 4.9% (41/837) of the total R7-DRA output (*Figure 6—figure supplement 1A*, *Table 4*), and the number of R8-DRA synapses onto R7-DRA was also similar (36.0 mean, range 33–38; *Figure 6—figure supplement 1B*, *Table 4*). Many of the synapses between the inner photoreceptors were also located outside the medulla (15% of R8-DRA synapses to R7-DRA, 16/108, mean 5.3 synapses per cell, range 3–8; and 48% of R7-DRA synapses to R8-DRA, 19/41, mean 6.3 synapses per cell, range 4–8; *Figure 6Ai*, arrow, *Supplementary file 3*).

The Dm9 cells in the DRA region were prominent synaptic targets of both inner photoreceptors, and provided strong synaptic feedback onto them (*Figure 6B*, *Table 4*, *Table 5*). They received 31.7 R7-DRA synapses per column (mean, range 26–38), corresponding to 11.3% (95/837) of the total R7-DRA output (*Table 4*), and R8-DRA was more strongly connected with 38.3 synapses (mean, range 36–43), corresponding to 23% (115/499) of the total R8-DRA output. Although the total number of inner photoreceptor inputs to Dm9 per seed column was fewer in the DRA region compared to central seed columns (70 mean, range 62–81), their relative strength was similar to central columns (*Supplementary file 1*). The distribution of R8 and R8-DRA synapses onto Dm9 was also different, with the greatest density of R8-DRA synapses in layer M6, reflecting the characteristic morphology of R8-DRA (*Figure 6Bi*). Our prior light microscopic study suggested that marginal Dm9 cells might receive both DRA and non-DRA inputs (*Sancer et al., 2020*). We therefore reconstructed R7 and R8 in three non-DRA columns innervating the same DRA Dm9, and found that cell indeed received input from both color and polarized light-sensitive photoreceptors (*Figure 6Biv*).

Dm2 was an R8 target in the central columns, but in the DRA region there were drastically fewer R8-DRA inputs, with equal contacts from R7-DRA and R8-DRA (7.0 mean, range 5–10 for R7-DRA, 9.0 mean, range 5–15 for R8-DRA synapses per column; *Figure 6Ci–ii*, *Table 4*; *Sancer et al., 2020*). The Dm11 cell type was an R7 target in central columns, but in the DRA region it received minor, balanced input from both R7-DRA and R8-DRA and was only found in one of the three DRA seed columns

(four from R7-DRA and eight from R8-DRA; *Figure 6—figure supplement 2D*, *Table 4*). Thus, while a reduction in R8-DRA input to cells targeted by R8 in the central columns was a repeated feature of the DRA region, as exemplified by Dm2, it was also seen in cells targeted by R7, such as Dm11.

The connectivity of inner photoreceptors with lamina cells was maintained in the DRA region, including with a substantial fraction of synapses outside the medulla (*Figure 6—figure supplement 2A,B*; *Figure 1—figure supplement 2G*; *Supplementary file 3*), but there were markedly fewer synapses to Mi cells from R8-DRA than from R8, including to Mi1 and Mi9, while Mi4 was not contacted in the seed DRA columns at all (Mi1: 3 mean, range 2–5; Mi9: 4 mean, range 0–9; *Figure 6—figure supplement 2C,E*). Of all the Mi cell types, Mi15 alone was a significant photoreceptor target in the DRA region, receiving synapses from both R7-DRA and R8-DRA (respectively, 7 mean, range 4–9, and 6 mean, range 3–10; *Figure 6Di–ii*, *Table 4*). Compared to the central column R7–8 input, Mi15 cells gained R7-DRA synapses in the DRA region, and they maintained their high fraction of inputs outside the medulla (*Figure 6D*; *Figure 1—figure supplement 2G*; *Supplementary file 3*).

Only one Tm cell type, Tm20, was connected to inner photoreceptors with more than two synapses throughout all three DRA seed columns. This cell type was selective for R8-DRA, but was much more weakly connected (5.3 mean, range 4–8 for R8-DRA; *Table 4*, *Supplementary file 1*) than it was to R8 in the central columns. There was only one other strongly connected Tm cell, which we refer to as Tm5-like, that had 12 synaptic inputs from both R7-DRA and R8-DRA inputs in one of the three DRA columns (*Figure 6—figure supplement 2F*). Together, these four cells highlight the absence of projections to the lobula, as compared to the central columns with 27 identified Tm cells (*Table 2*).

## Dm8-like photoreceptor targets in the DRA region

The strongest target of R7-DRA cells was Dm-DRA1 (*Sancer et al., 2019*). Together, 20 Dm-DRA1 cells received 70.1 (mean, range 66–77) synapses per DRA seed column, corresponding to 25.3% (212/837) of total R7-DRA output (*Table 4*, *Supplementary file 1*). There were no synaptic inputs from R8-DRA, and synapses from R7-DRA to Dm-DRA1 were consistently restricted to the proximal edge of layer M6 where R8-DRA are not present (*Figure 7Ai*, Aii). There were few synapses in neighboring non-DRA columns (*Figure 7—figure supplement 1A*), and only one of the contacting R7 cells made a substantial number (11) of synapses. The inner photoreceptors choose their DRA fates independently, and therefore this photoreceptor could be an R7-DRA, even though the R8 in the same column terminated in M3; low numbers of such mixed ommatidia consistently occur at the DRA boundary (*Wernet et al., 2003*; *Wada, 1974b*). Dm-DRA1 cells have deep projections that leave the DRA region and stratify below layer M6, and these did not receive strong photoreceptor input (*Figure 7Aii*, compare light microscopy of clone in 7Aiii), nor did they preferentially occupy either pale or yellow columns (*Figure 7—figure supplement 2*).

To gain a better understanding of the synaptic connectivity between R7-DRA and Dm-DRA1, we identified all the DRA photoreceptor inputs to three fully reconstructed Dm-DRA1 cells (*Figure 7B*). Each cell was connected to 9 (mean, range 8–10) neighboring R7-DRA, receiving a total of 94 synapses (mean, range 66–112) from R7-DRA, but almost no connections from R8-DRA (two R8-DRA form 3 and 2 synapses with the posterior Dm-DRA1; *Figure 7Bi*). Dm-DRA1 cells have been proposed to be DRA-specific equivalents of Dm8 (*Courgeon and Desplan, 2019*; *Sancer and Wernet, 2021*), but we never detected vertical protrusions resembling Dm8 'home columns'. The smoothed distribution of synapses along a linear projection of the DRA covered ~50 μm with the strongest input in the center, but without a single, dominant home column (*Figure 7Biv*, *Figure 7—figure supplement 3*).

The Dm-DRA2 cell type was the strongest R8-DRA target after Dm9 and R7-DRA (*Table 4*), with 34.3 (mean, range 19–44) seed column synapses to nine cells (*Figure 7C*, *Supplementary file 1*; *Sancer et al., 2019*). Each Dm-DRA2 cell had multiple characteristic vertical projections, whose lengths were variable but markedly longer than the single vertical processes of Dm8 home columns, often reaching layer M1 (*Figure 7Ci*, Ciii; *Sancer et al., 2019*), and these vertical projections received R8-DRA synapses (*Figure 7Cii*, Ciii, Eiii). To quantify the pattern of their photoreceptor inputs, we fully reconstructed three Dm-DRA2 cells. These cells were innervated by 9.3 (mean, range 8–11) R8-DRA and received minor inputs from R7-DRA (3.7 mean, range 3–4 cells per Dm-DRA2 with an average of 5.3, range 3–7 synapses; *Figure 7D*, *Supplementary file 1*). The distribution of R8-DRA synapses to each Dm-DRA2 cell covered ~50 μm with a peak in its center and without a single, obvious home column (*Figure 7Di*, Dii). Thus, Dm-DRA2 directly integrate photoreceptor inputs over

multiple columns, with stronger input at the center of their receptive field, much like we found for Dm-DRA1.

The reconstructed Dm-DRA1 cells highly overlapped along the DRA region, as did the Dm-DRA2 cells (*Figure 7Ei*, Eii; *Sancer et al., 2019*). The population of all Dm-DRA1 cells connected to the same R7-DRA seed photoreceptor (8.3 mean, range 6–11) spanned 17 DRA columns (mean, range 13–19), corresponding to just under half the length of the DRA region. In comparison, the number of Dm-DRA2 targets of one R8-DRA was fewer (3.3 mean, range 2–5 Dm-DRA2 cells per column), and the length of the DRA region covered was also shorter at 10.7 columns (mean, range 10–12). These differences in connectivity may have implications for how the two Dm-DRA cell types differently compute skylight polarization information originating from the same visual field. Finally, comparison of inputs to Dm-DRA1 and Dm-DRA2 from the same location along the DRA revealed a separation of R7-DRA and R8-DRA synapses into different sublayers of M6 (*Figure 7Eiii*), confirming prior observations from light microscopy (*Sancer et al., 2019*).

## Visual projection neurons connecting R7-DRA with the central brain

The second most prominent R7-DRA target was a cell type that we refer to as MeTu-DRA (*Figure 8A*). These cells exclusively received R7-DRA synapses, with no input from R8-DRA, and are modality-specific. Both the reconstructed cells and the matching light microscopy MeTu-DRA clones projected long axons to the small unit of the AOTU (*Figure 8Aii*), identifying them as specialized MeTu cells (*Omoto et al., 2017*; *Panser et al., 2016*; *Otsuna et al., 2014*; *Tai et al., 2021*). They closely resemble cells described in the DRA region of optic lobes from larger insects (*Pfeiffer and Kinoshita, 2012*; *el Jundi et al., 2011*). All 30 reconstructed MeTu-DRA cells received 61.3 (mean, range 49–85) R7-DRA synapses per seed column, corresponding to 22.0% (184/837) of the total R7-DRA output (*Figure 8Aiii*, *Table 4*, *Supplementary file 1*). Synapses from R7-DRA to MeTu-DRA were restricted to the photoreceptor tips at the proximal edge of layer M6, as we found for Dm-DRA1 (*Figure 8Aiv*). MeTu-DRA cells also innervated the dorsal medulla outside the DRA region (*Figure 8Av*), but avoided non-DRA photoreceptors by stratifying below M6, and showed no obvious pale or yellow preference (*Figure 8—figure supplement 1A*). We visualized single MeTu-DRA cell clones in the DRA and found their photoreceptor contacts were always restricted to the DRA region (*Figure 8Av*, arrow). The medulla branches of all MeTu-DRA skeletons connected to R7-DRA from the same DRA seed column (11.3 mean, range 8–16) widely overlapped, covering 15 DRA columns (mean, range 12–17; *Figure 8Avi*). Thus, the MeTu-DRA cells receive skylight polarization information from R7-DRA over similar portions of the sky as Dm-DRA1 cells.

Since the axons of the central region MeTu cells projected to a discrete subdomain of the small unit of the AOTU (*Figure 5Ai*), we assessed whether they intermixed with MeTu-DRA terminals, and found they do not. The MeTu-DRA and the MeTu cells we reconstructed from central columns terminate in separate, adjacent AOTU compartments (*Figure 8vii*, inset).

We identified one additional, previously uncharacterized VPN type connecting the DRA region with the central brain, which we refer to as VPN-DRA. These cells send axonal projections to the PLP (*Figure 8Bi*, Bii). There were six VPN-DRA cells with >2 synapses contacted by the DRA seed columns, and they received 10 (mean, range 6–13) R7-DRA synapses per seed column (corresponding to 3.6%, 30/837, of total R7-DRA output), with no R8-DRA synapses (*Figure 8Biii*, *Table 4*, *Supplementary file 1*). The main branches in the medulla of VPN-DRA cells followed the DRA region, with smaller branches innervating non-DRA columns (*Figure 8Biv*). Double labeling of photoreceptors and a driver line with apparent VPN-DRA expression indicated that photoreceptor input to VPN-DRA cells might be dominated by DRA inner photoreceptors, with only fine processes contacting non-DRA photoreceptors (*Figure 8Bv*).

VPN-DRA cells form an overlapping group of neurons covering the entire DRA region of a given hemisphere (*Figure 8Bvi*). There was morphological heterogeneity within the population, with some cells having additional processes, both in the medulla and in the central brain and a different cell body position (*Figure 8—figure supplement 1B*). Therefore, there exist different subtypes of VPN-DRA.

## Other newly identified photoreceptor targets in the DRA region

Two more previously uncharacterized cell types received photoreceptor input in the DRA seed columns. The first, which we refer to as MeMe-DRA, were large heterolateral medulla cells that

connected the DRA regions of both hemispheres, crossing the central brain without forming synapses there (*Figure 9A*). We reconstructed two MeMe-DRA cells, and could not find additional ones, and we matched the morphology of the cell type using light microscopy (*Figure 9Aiv*). Morphologically, these cells resemble the polarization-sensitive MeMe1 cells characterized in locusts (*el Jundi et al., 2011*; *El Jundi and Homberg, 2010*). They were exclusively innervated by R7-DRA, receiving 12.0 (mean, range 0–21) synapses per seed column, corresponding to 4.9% of total R7-DRA output (*Figure 9Aii*, Aiii, *Table 4*, *Supplementary file 1*). The MeMe-DRA processes in the ipsilateral hemisphere appear to be primarily dendritic, whereas the contralateral processes appear to be axonic (*Figure 9Aiii*, Aiv). The putative dendritic processes appeared to contact DRA photoreceptors, based on light microscopy labeling, with processes reaching layer M6 exclusively in the DRA region (*Figure 9Aiv*). Intriguingly, the area of medulla columns innervated by a given cell is flipped along the anterior-posterior axis between the two hemispheres, so that the anterior half of the ipsilateral medulla is connected, via the MeMe-DRA cell, with the posterior half of the contralateral medulla and vice versa (*Figure 9Av*).

Another newly identified set of DRA photoreceptor targets were morphologically diverse Mti cells that covered the dorsal medulla (*Figure 9B and C*, *Figure 9—figure supplement 1*), for which we also found putative light microscopy matches. All 10 of these Mti-DRA cells were innervated exclusively by R7-DRA, but there were two distinct morphological populations, which we refer to as Mti-DRA1 (*Figure 9Bi-Biv*) and Mti-DRA2 (*Figure 9Ci-Civ*). There were six Mti-DRA1 cells with ventral cell bodies (*Figure 9Bi*) that received together 12.3 (mean, range 10–14) synapses per seed column, corresponding to 4.4% of total R7-DRA output (*Figure 9Bii*, *Supplementary file 1*). Individual reconstructed Mti-DRA1 cells had main processes along the DRA region with short processes innervating non-DRA columns (*Figure 9Biii*, Biv). Mti-DRA2 cells had dorsally located cell bodies (*Figure 9Ci*) and were more weakly connected to R7-DRA, with four Mti-DRA2 neurons making 5 (mean, range 3–8) synapses per seed column, corresponding to 1.8% of R7-DRA total output (*Figure 9—figure supplement 1*, *Table 4*, *Supplementary file 1*). The main processes of Mti-DRA2 skeletons also innervated the DRA region but these cells formed longer processes extending further ventrally into non-DRA columns (*Figure 9Ciii*, Civ).

## Comparison of inner photoreceptor connectivity of central and the DRA columns

Taken together, our reconstructions of the synapses of the inner photoreceptors specialized for color and polarization vision revealed specific stereotypical similarities, as well as striking differences. The pattern of connectivity between photoreceptor terminals from the same ommatidium, and with Dm9, was highly consistent in columns processing color and polarization, indicating that the mechanisms of lateral inhibition between photoreceptors and gain modulation by Dm9 can serve both modalities (*Schnaitmann et al., 2018*; *Weir et al., 2016*; *Figure 10A*).

A second striking similarity was the location of synapses located outside the medulla (*Figure 1—figure supplement 2G*). In both the central and DRA columns, more than half of the synapses between the photoreceptors and the inputs to the lamina cells L1 and L3 were outside the medulla (*Figure 1—figure supplement 2G*, *Figure 2—figure supplement 1*, *Figure 6—figure supplement 1A*, B, *Supplementary file 3*). In both cases, the strongest lamina monopolar target was L3, receiving 80% of its inner photoreceptor inputs outside the medulla (*Figure 2—figure supplement 1C*, *Figure 6—figure supplement 1B*). L3 could contribute to color processing through innervation of Dm9, Tm20, and Tm5c (*Takemura et al., 2015*), in addition to the ON and OFF pathways computing motion (*Silies et al., 2013*). Our reconstructions therefore provide further support for the early convergence of motion and color channels, which have until recently been believed to operate separately (*Heisenberg and Buchner, 1977*; *Yamaguchi et al., 2008*; *Wardill et al., 2012*; *Pagni et al., 2021*; *Li et al., 2021*).

Among Dm cell types, both R7-DRA and R8-DRA connect to their own modality-specific amacrine-like cell type, Dm-DRA1 and Dm-DRA2, respectively (*Sancer et al., 2019*). Such connections never occur in the central seed columns, where only R7 connects to Dm8 (*Figure 10B*). Recent studies point toward Dm-DRA1 (and possibly Dm-DRA2) as developmentally similar duplicates of Dm8 cells (*Courgeon and Desplan, 2019*; *Sancer and Wernet, 2021*). Many of the connections to Mi cells are weaker or missing in the DRA region (*Figure 10C*). Given the role of Mi1, Mi4, and Mi9 in computing motion (*Strother et al., 2017*), this could indicate a stricter separation between computations of motion and

polarization, as compared to motion and color. In contrast, Mi15 cells are targeted by both central and DRA inner photoreceptors. Its role is not known, but its partly dopaminergic neurotransmitter phenotype suggests a modulatory role (*Davis et al., 2020*). We reconstructed morphologically different Mti cells both from central and DRA columns, and while the Mti cells are R8 targets in the central columns (*Figure 3A and B*), they are targets of R7 in the DRA region with Mti-DRA1 cells having a greater input per seed column than the Mti cells (*Figure 9B and C*).

A striking difference between the central and DRA columns is the absence of most lobula-projecting cells targeted by photoreceptors (*Figure 10D*). We confirm Tm5a, Tm5b, Tm5c, and Tm20 as targets of central R7 or R8 (*Meinertzhagen et al., 2009*; *Gao et al., 2008*; *Melnattur et al., 2014*), and we also confirm the lack of synaptic connections with these cell types in the DRA region (*Sancer et al., 2020*). In addition to Tm cells, we identify the new cell type ML1, also connecting the medulla to the lobula but without the precise retinotopy of the Tm cell arrays, and with additional branches in the central brain (*Figure 10Dii*). The lack of Tm and ML1 connectivity in the DRA region is reflected in the overall lower number of R8-DRA synapses (*Table 4*).

R7 and R7-DRA cells are strongly connected to respectively different classes of MeTu cells, whose axon terminals are spatially segregated in the small unit of the AOTU (*Figure 10E*; *Timaeus et al., 2020*). These data support previous proposals of parallel pathways to the central brain via the AOTU (*Otsuna et al., 2014*; *Omoto et al., 2017*; *Timaeus et al., 2020*; *Tai et al., 2021*). Besides the AOTU, photoreceptor targets in the central and DRA columns both project to the PLP, but otherwise they show little overlap in the central brain areas they project to (*Figure 10F*). The additional contacts of DRA region projection neurons are focused in the lateral horn and the contralateral medulla, while the central column projection neurons innervate widely across the central brain, including the mushroom bodies, and the contralateral lobula plate. A key modality-specific feature of the VPNs is the dominance of R8 input in the central columns, and R7 in the DRA region (*Figure 11A*).

## Cell types distinguishing between pale and yellow inputs

The fly's retinal mosaic can support circuits with different chromatic sensitivities by selective sampling of pale and yellow photoreceptors. For example, cells contrasting pale and yellow R7 input would be expected to have high chromatic sensitivity around ~350 nm, where the spectral sensitivity of Rh3 and Rh4 overlap (*Salcedo et al., 1999*). Our reconstructions revealed several cases of connectivity patterns that could support such selective sampling. For R7, these included two cell types previously reported to be pale- or yellow-selective: Tm5a for yellow R7 (*Karuppudurai et al., 2014*), and Tm5b for pale R7 (*Menon et al., 2019*; *Figure 11Bi*). By completely reconstructing several Dm8 cells for the first time (*Figure 2C*), we found that Dm8 neurons were most highly innervated by R7 in their home columns, with approximately equal sampling of pale and yellow R7 input in the surrounding columns. Examining the encoding of chromatic signals in Tm5a/b and Dm8 cells, and their other synaptic partners, is therefore a promising avenue for addressing the open question of how, or if, UV information from yellow ommatidia is processed differently from UV signals downstream of pale R7 cells.

We found more pale-yellow selective cells among the R8 targets. Two VPNs were newly identified as selective for pale R8 input, the aMe12 and ML-VPN1 cells. ML-VPN1 provides input to the PLP, while aMe12 has presynaptic sites in the accessory medulla, the PLP, and the mushroom body, where it provides input to γd Kenyon cells (*Li et al., 2020*; *Scheffer et al., 2020*). These connectivity patterns suggest roles in circadian entrainment and learning and memory, respectively (*Figure 11Bii*). *Drosophila* pale R8 cells are most sensitive to blue wavelengths (*Sharkey et al., 2020*; *Salcedo et al., 1999*; *Schnaitmann et al., 2018*; *Heath et al., 2020*), and the projections of aMe12 and ML-VPN1 demonstrate that blue light is processed and used by the central brain within one synapse. Little is known about the central processing of color in *Drosophila* (*Longden, 2016*), but the detection of blue light plays an important role in the circadian avoidance of bright light (*Lazopulo et al., 2019*). Flies can also learn to discriminate between large areas of blue and green light and a group of VPNs have been reported to provide input to γd Kenyon cells and be required for this ability (*Vogt et al., 2014*).

While near-perfect specificity for pale versus yellow inputs appears to be limited to a few cell types, several strongly connected cell types show strongly biased inputs: For example, Dm2 favors pale over yellow R8 (60:15), and Tm5c favors yellow over pale R8 (103:20), a result that corroborates and extends the findings of the medulla-7-column connectome. Individual cells, in particular Tm20 neurons, that span just one column, are also constrained to be selective for pale and yellow photoreceptor input.

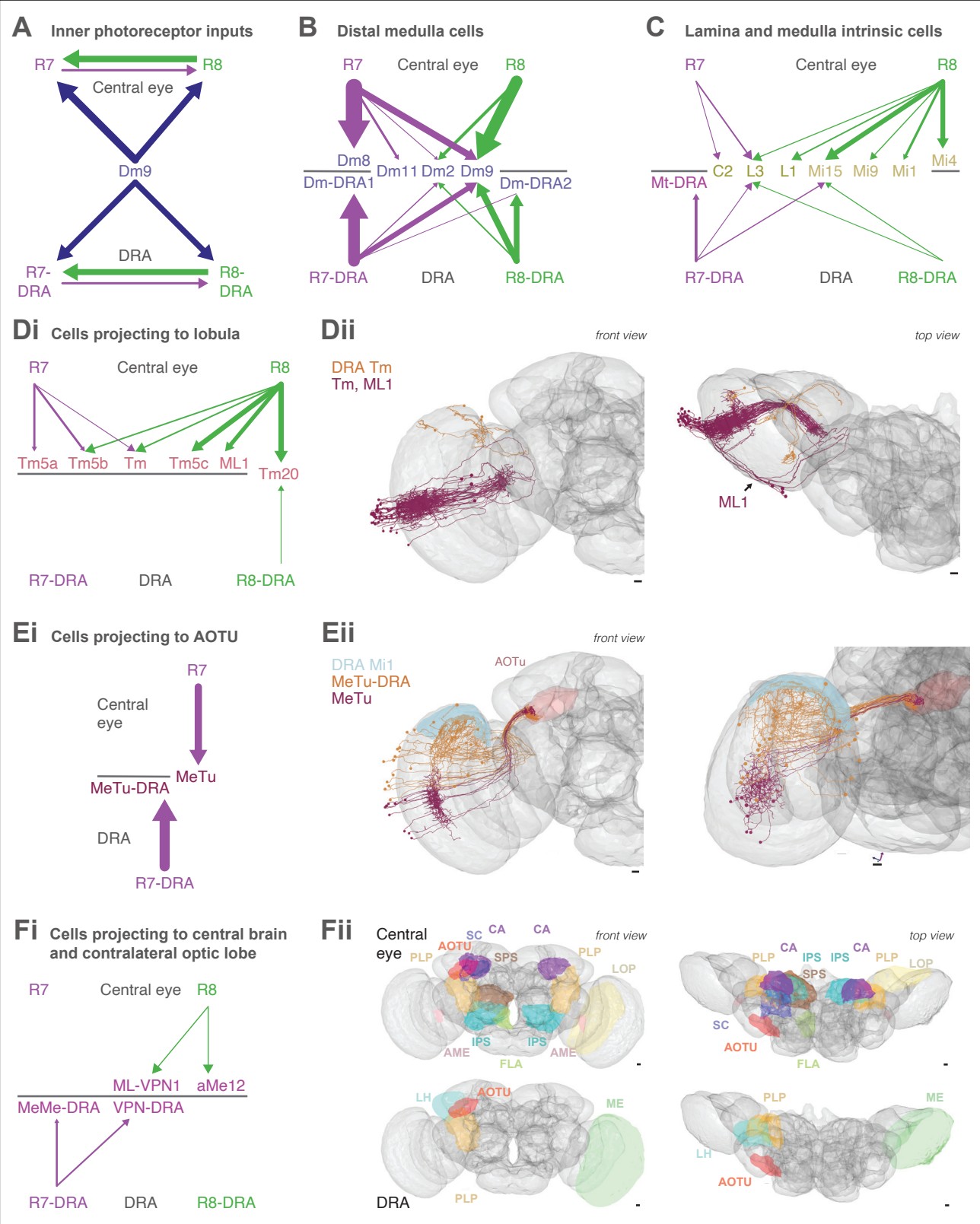

**Figure 10.** Summary of inner photoreceptor connectivity of central and the dorsal rim area (DRA) columns. (**A**) Schematic summary of synaptic connections between central R7–8 or R7-DRA and R8-DRA, as well as other cell types providing synaptic feedback to them (blue). In all panels of this figure, arrow widths are proportional to the numbers of synapses per seed column, and weak connections were excluded for clarity by a threshold of >4 synapses per column per photoreceptor type. Despite the differences in modality, the circuit organization of inputs to R7–8 cells is conserved in the

*Figure 10 continued on next page*

*Figure 10 continued*

inputs to R7-DRA and R8-DRA. (**B**) Synapses between R7–8 and Dm cell types in central and DRA columns. Note the presence of a second Dm8-like cell type downstream of R8 only in the DRA (Dm-DRA2). (**C**) Summary of inner photoreceptor connections with lamina, Mi and Mti cells in central and DRA columns; in the central columns, the Mti connections were below the threshold of 4 synapses per column and are not shown. The connections from R8 to these cells were very reduced in the DRA. (**D**) Summary of lobula connections. (Di) Schematic of R7–8 versus R7-DRA and R8-DRA connections with Tm5a, Tm5b, Tm5c, Tm20, ML1 cells, and cells we identified as belonging to the Tm cell class (Tm). Note the virtual absence of lobula connectivity in the DRA region. (Dii) Front view (left) and top view (right) of all reconstructed central Tm and ML1 skeletons (claret) and Tm skeletons from the DRA (orange). (**E**) Summary of MeTu cell connections. (Ei) Schematic of R7 and R7-DRA cells targeting different MeTu populations. (Eii) Front view (left) and DRA view (right) of all central MeTu skeletons (claret) and MeTu-DRA skeletons (orange), with axons following the same tract but terminating with spatially separated axon target areas in the anterior optic tubercle (AOTU). (**F**) Summary of other visual projection neurons. (Fi) Diagram summarizing VPN connectivity in central and DRA columns. While R8 input dominates in central columns, R7 input dominates in the DRA region. (Fii) Front view (left) and top view (right) of VPN target areas in the central brain. VPNs from both central and DRA columns project to the posterior lateral protocerebrum (PLP, golden yellow) and the AOTU (red). The central column VPNs additionally project ipsilaterally to the mushroom body calyx (mauve), accessory medulla (aMe, pink), superior clamp (SC, dark blue), flange (FLA, green), superior posterior slope (SPS, brown), inferior posterior slope (IPS, cyan), and contralaterally to the PLP, mushroom body calyx (CA), AME, and lobula plate (LOP, cream). The DRA VPNs project ipsilaterally to the AOTU, PLP, and lateral horn (LH, turquoise), and contralaterally to the medulla (ME, lime green).

Such selectivity could in principle be exploited at the level of the synaptic targets of these neurons. Surprisingly, Mi9 cells, another cell type present in every column of the eye, also revealed strong pale/yellow biases in our reconstructions, both with R7 inputs (3:9) and R8 inputs (26:1). Given the known role of Mi9 in shaping direction-selective responses on T4 dendrites (*Strother et al., 2017*; *Takemura et al., 2017*), these data identify yet another level of crosstalk between chromatic and motion-sensitive channels. Overall, it is an unexpected finding that most projections from the medulla to the lobula from photoreceptor target neurons convey biased pale/yellow inputs. These differences could mediate much broader effects of pale/yellow differences in medulla and lobula circuits than currently appreciated, or these specializations could be blunted by combinations at the next synaptic layer.

## Discussion

Our systematic reconstruction of all synaptic inputs and outputs of identified, functionally specialized *Drosophila* photoreceptors (pale and yellow R7–8, R7-DRA, and R8-DRA) provides a comprehensive inventory of the first steps of the color and polarization pathways, from which all the computations of the dependent behaviors stem. These data revealed core connectomic motifs shared across column types (*Figure 10A–C*), multiple new photoreceptor targets, and uncovered additional cell types as being connected to specific photoreceptor subtypes conveying specific color and polarization information to the central brain (*Figure 10E and F*).

### Inner photoreceptor connections outside the medulla neuropil

We confirmed previously reported synaptic partners of the inner photoreceptors in the non-DRA medulla (*Figure 1—figure supplement 2*) and identified new photoreceptor targets. As prior reconstructions were incomplete (*Takemura et al., 2015*), it was unclear whether the unidentified connections were mainly onto new target neurons or represented more connections onto known cell types; our reconstructions revealed both types of omissions. One functionally important set of missed connections are the synapses between inner photoreceptors from the same central and DRA ommatidia, which we found to be stronger than previously reported (*Takemura et al., 2008*; *Takemura et al., 2015*), due to significant numbers of synapses outside the medulla. These synapses likely contribute to color-opponent responses seen in central R7 and R8 terminals (*Schnaitmann et al., 2018*) and the polarization-opponent signals measured from DRA photoreceptors (*Weir et al., 2016*). Our reconstructions also support a larger-scale opponent process mediated by multicolumnar Dm9 cells (*Heath et al., 2020*), which also formed some synapses outside the medulla neuropil.

Other cell types also received inner photoreceptor input outside the medulla, notably the lamina monopolar cells L1 and L3. These lamina connections indicate that chromatic comparisons arising from R7 and R8 may feed into the motion vision pathway, and identifies a new site for interplay between the 'color' and 'motion' pathways (*Wardill et al., 2012*; *Schnaitmann et al., 2013*; *Pagni et al., 2021*; *Li et al., 2021*).

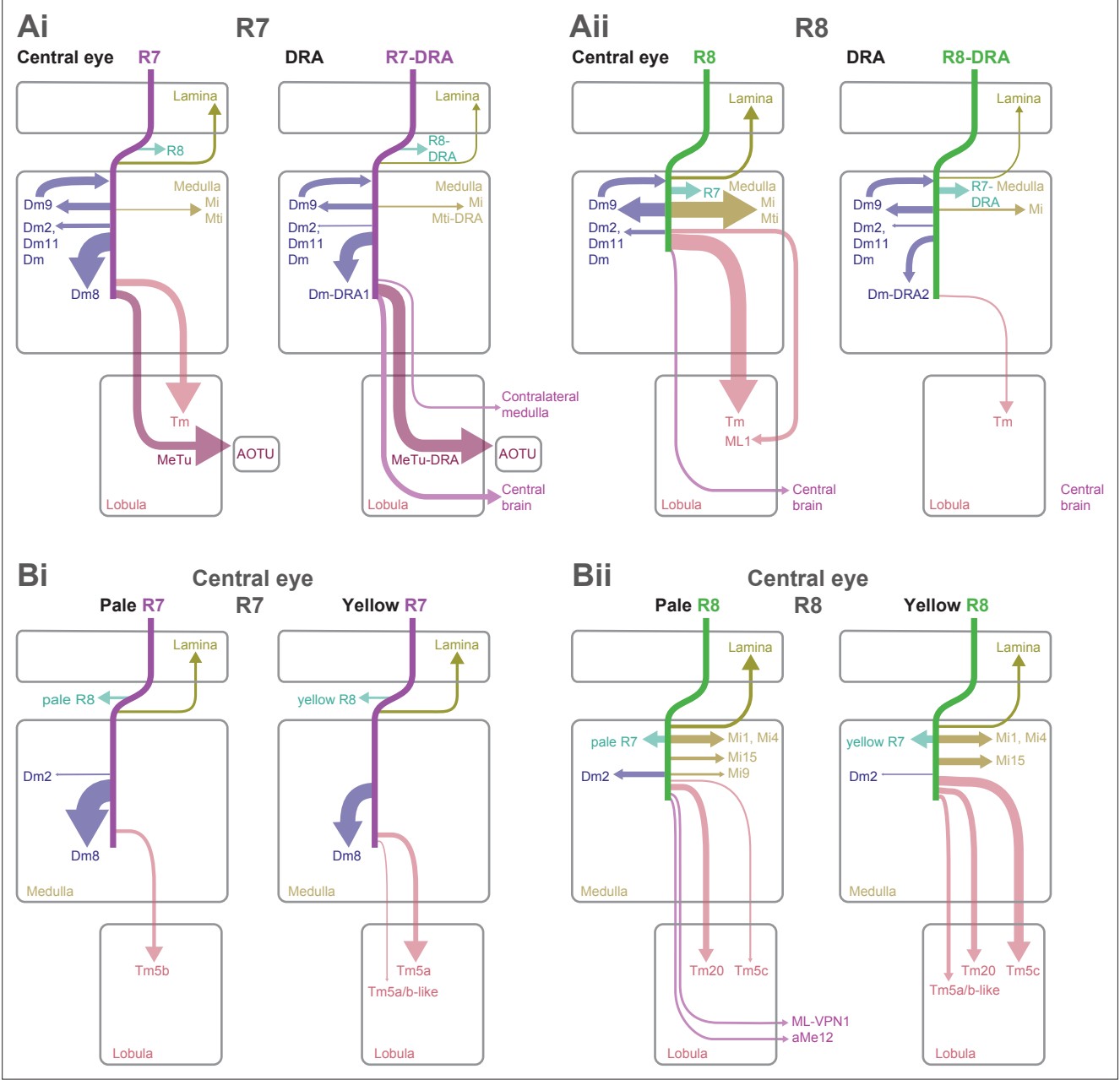

**Figure 11.** Comparison of central versus dorsal rim area (DRA), and pale versus yellow pathways. (**A**) Graphical comparison of central and DRA synaptic pathways. In all panels, the arrow widths are proportional to the numbers of synapses per seed column, and weak connections were excluded for clarity by a threshold of >4 synapses per column. (Ai) Arrows indicating the relative weight of R7 connections (cell types postsynaptic to pale and yellow R7) and R7-DRA. (Aii) Arrows indicating the relative weight of R8 connections (sum of pale and yellow R8) and R8-DRA region. Connections to the lobula neuropil are dominated by R8 targets in the central columns. Lobula connections are virtually absent in the DRA region (synapse numbers below 1% of total synapse count), where connectivity of R8-DRA is dramatically reduced, limited to local computations in the medulla. The projections to the central brain that are not to the anterior optic tubercle (AOTU) are driven by R8 in the central columns, but by R7-DRA in the DRA region. (**B**) Graphical comparison of central pale and yellow-specific synaptic pathways. (Bi) Arrows indicating the relative weight of pale R7 versus yellow R7 connections in central seed columns. Columnar cell types such as lamina cells have the capacity to preserve pale and yellow information, and Dm8 cells are most densely innervated by their home column input. Tm5a cells were selective for yellow R7, Tm5b cells were selective for pale R7, and the Tm5a-like and Tm5b-like cells were selective for yellow R8, but also received yellow R7 synapses. These diagrams omit cells such as the MeTu neurons since they receive nearly balanced input from pale and yellow R7s. (Bii) Arrows indicating the relative weight of pale R8 versus yellow R8 connections in central seed columns. Columnar medulla cells that have the capacity to preserve pale and yellow information were targets of R8 cells. The aMe12 and ML-VPN1 cell types were specific for pale R8 input, while Tm5c had a strong bias for yellow R8, along with the Tm5a-like and Tm5b-like cells. ML1s are omitted from these diagrams since they received nearly balanced input from pale and yellow R8s.

Together, these observations suggest that synapses in an unexpected location, outside the main synaptic layers of the medulla, could play a significant role in early visual processing.

## Newly identified targets of central versus DRA inner photoreceptors

In the non-DRA medulla, we found, for the first time in an EM study, strong synaptic connections between R7s and MeTu cells that project to the AOTU (*Figure 11Ai*), confirming previous claims based on light microscopy (*Timaeus et al., 2020*). This finding may reconcile disparate observations, such as a role in wavelength-specific phototaxis for cells matching MeTu morphology (*Otsuna et al., 2014*), as well as measurements of color-sensitive signals in the AOTU of bees (*Mota et al., 2013*). Previous anatomical studies partitioned MeTu cells into distinct subclasses that terminate in discrete subdomains of the AOTU (*Omoto et al., 2017*; *Timaeus et al., 2020*; *Tai et al., 2021*). Here, we identified modality-specific MeTu-DRA cells that only integrate from the polarization-sensitive R7-DRA photoreceptors, while avoiding synaptic contacts with color-sensitive pale or yellow R7s. Our MeTu and MeTu-DRA cells target adjacent subdomains within the small unit of the AOTU (*Figure 10E*), in agreement with proposals that parallel channels convey different forms of visual information from the eye to the central complex via the AOTU (*Hardcastle et al., 2021*; *Hulse et al., 2021*; *Timaeus et al., 2020*; *Omoto et al., 2017*).

We also identified connections from inner photoreceptors to several cell types either not previously described or not known to be photoreceptor targets, thus setting up a clear expectation that these cells should contribute to color or polarization vision. ML1 is a new, major target of R8, a cell type that connects the medulla to the lobula via a previously unknown, non-columnar pathway and the central brain (*Figure 4D*, *Figure 11Aii*). Previous studies have identified an important role for Tm5a/b/c and Tm20 cell types for chromatic processing in the lobula (*Lin et al., 2016*; *Gao et al., 2008*), and we have confirmed that these cell types are targets of R7–8. Whether these Tm neurons and ML1 cells have common targets in the lobula, feed into shared central pathways or contribute to separate channels remain open questions; the lobula arbors of the Tm and ML1 cells are mostly in different layers arguing against direct synaptic interactions between the cells.

The reconstruction of DRA photoreceptor targets confirmed the modality-specific connectivity of Dm-DRA cell types within layer M6: Dm-DRA1 to R7-DRA, and Dm-DRA2 to R8-DRA (*Figure 11A*). The R7-DRA and R8-DRA cells respond to orthogonal orientations of polarized light at each location of the DRA, so Dm-DRA1 and Dm-DRA2 likely process orthogonal e-vector orientations spatially averaged by pooling over ~10 ommatidia (*Weir et al., 2016*). Our data also revealed additional DRA pathways into the PLP in the central brain via VPN-DRA cells, as well as to the contralateral DRA, via MeMe-DRA cells (*Figure 10F*). Such interhemispheric connections have been demonstrated in larger insects (*el Jundi et al., 2011*; *Labhart, 1988*), but not in *Drosophila,* and their synaptic input was not known. Interocular transfer contributes to navigation by desert ants that can see the celestial polarization but not visual landmarks (*Wehner and Müller, 1985*), but the interactions between the DRA regions remain poorly understood. The identification of MeMe-DRA neurons may enable the mechanisms of such phenomena to be explored.

For color vision, we have confirmed the pale-yellow specificity of Tm5a and Tm5b cells (*Figure 11Bi*). We have additionally identified as pale-specific the aMe12 and ML-VPN1 cell types, and the yellow bias of Tm5c and the pale bias of Mi9 (*Figure 11Bii*). Our detailed analysis of Dm8 inputs confirmed that these neurons receive most of their photoreceptor input in a central home column, consistent with pale and yellow subtypes of Dm8 cells having distinct chromatic properties (*Courgeon and Desplan, 2019*; *Menon et al., 2019*; *Li et al., 2021*; *Pagni et al., 2021*). The selective photoreceptor input to these cell types, combined with input to columnar cells from a single photoreceptor subtype, indicates that wavelength-specific information is maintained in the medulla and lobula (*Figure 11Bi*). The projections of aMe12 and ML-VPN1 to the central brain indicate the possibility that wavelength-specific photoreceptor responses are directly conveyed into the central brain by these cells, although they likely integrate input from other chromatically sensitive cell types (*Figure 11Bii*).

## Limitations of our approach

By focusing on a small number of columns, we have delivered a near-complete picture of local connectivity, but we cannot rule out the possibility that certain cell types may have been overlooked. For instance, we chose an arbitrary threshold of two to three synapses below which we did not endeavor to reconstruct synaptic targets to the extent required to uniquely identify them. As a result we may,

in principle, have missed large cells that receive small, but significant inputs across many columns. Furthermore, we cannot rule out regional specializations in the optic lobes, such that specific cell types might be found outside of our seed columns. For example, some MeTu cells are only found in the dorsal third of the medulla (*Omoto et al., 2017*; *Otsuna et al., 2014*), where incoming R7 cells are known to co-express Rh3 and Rh4 rhodopsins (*Mazzoni et al., 2008*).

## Complementary usage of multiple connectomic data sets and outlook

Taken together, the data presented here provides access to the full complement of R7 and R8 photoreceptor targets from functionally specialized optical units. By reconstructing these local circuits within a full-brain EM volume, we were able to establish the complete morphology of large, multicolumnar cell types, that are strongly connected to photoreceptors, but had eluded previous connectomic reconstruction efforts (*Takemura et al., 2013*; *Takemura et al., 2015*). The sparse tracing approach we have implemented and described here enabled efficient identification of the complete set of upstream and downstream partners of the inner photoreceptors, in a manner that is complementary to the dense connectomes generated from smaller-scale medulla volumes. As an example of this synergistic usage of complementary data sets, we have returned to the 7-column data (*Takemura et al., 2015*; *Clements et al., 2020*) and used our whole-cell morphologies to match the bodies of previously unidentified photoreceptor targets. In that process we have established strong candidates for MeTu, ML1, and perhaps for aMe12 (see Materials and methods) and confirmed several aspects of their connectivity in the *Zheng et al., 2018*, data set. In so doing, we now have access to the additional connectivity data provided by the 7-column dense reconstruction. While full exploration and follow-up analyses of these combined data is beyond the scope of this work, the combined analysis of the *Zheng et al., 2018*, and the medulla-7-column connectome revealed several intriguing connectivity patterns, including new candidate paths for the integration of output from different photoreceptor types. For example, the MeTu cells that are postsynaptic to R7 also receive indirect R8 input via the R8 target Mi15, and ML1 combines direct R8 input with indirect input from outer photoreceptors via lamina neurons. As further connectome data sets are completed, this comparative interplay between data sets with unique advantages and limitations will be an important step in both cross-validating and extending the applicability of all related data sets.

Our reconstruction of the DRA photoreceptor targets provides the first EM-based connectomic data set for modality-specific cell types likely to process skylight information in any insect and will be important for developing refined models of skylight navigation (*Gkanias et al., 2019*). Core motifs shared between DRA and central columns are prime candidates for circuit elements that perform computations, such as establishing opponency, that are key for both polarization and color processing, whereas cell types with preferential connections to either pale or yellow columns are promising candidates for the study of specific aspects of color processing in the insect brain. This comprehensive catalog of the neurons carrying signals from R7 and R8 photoreceptors deeper into the brain establishes a broad foundation for further studies into the mechanistic basis of color vision and its contributions to perception and behavior.

## Materials and methods

### Key resources table

| Reagent type (species) or resource | Designation | Source or reference | Identifiers | Additional information |
|---|---|---|---|---|
| Genetic reagent (*Drosophila melanogaster*) | MCFO-1; "pBPhsFlp2::PEST in attP3;;pJFRC201-10XUAS-FRT> STOP > FRT-myr::smGFP-HA in VK0005, pJFRC240-10X-UAS-FRT> STOP > FRT-myr::smGFP-V5-THS-10XUAS-FRT> STOP > FRT-myr::smGFP-FLAG in su(Hw)attP1" | https://doi.org/10.1073/pnas.1506763112 | RRID:BDSC_64085 | |
| Genetic reagent (*Drosophila melanogaster*) | pJFRC51-3XUAS-IVS-Syt::smHA in su(Hw)attP1, pJFRC225-5XUAS-IVS-myr::smFLAG in VK00005 | https://doi.org/10.1073/pnas.1506763112 | | |

*Continued on next page*

*Continued*

| Reagent type (species) or resource | Designation | Source or reference | Identifiers | Additional information |
|---|---|---|---|---|
| Genetic reagent (*Drosophila melanogaster*) | SS02978 (w; 42E06-p65ADZp in attP40; VT064564-ZpGdbd in attP2/TM6B) | This study | SS02978 | Split-GAL4 lines are available via https://www.janelia.org/split-GAL4 |
| Genetic reagent (*Drosophila melanogaster*) | SS01015 (w; 21B12-p65ADZp in attP40; 20E07-ZpGdbd in attP2) | This study | SS01015 | Split-GAL4 lines are available via https://splitgal4.janelia.org/cgi-bin/splitgal4.cgi |
| Genetic reagent (*Drosophila melanogaster*) | SS28175 (w; 51E06-p65ADZp in attP40/CyO,Tb; 15D05-ZpGdbd in attP2/TM6B) | This study | SS28175 | Split-GAL4 lines are available via https://www.janelia.org/split-GAL4 |
| Genetic reagent (*Drosophila melanogaster*) | R56F07 | Bloomington *Drosophila* Stock Center | RRID:BDSC_39160 | |
| Genetic reagent (*Drosophila melanogaster*) | ortc2b-Gal4 | https://doi.org/10.1016/j.neuron.2013.12.010 | | |
| Genetic reagent (*Drosophila melanogaster*) | 20XUAS-CsChrimson-mVenus in attP18 | https://doi.org/10.1038/nmeth.2836 | RRID:BDSC_55134 | |
| Antibody | Anti-chaoptin mouse monoclonal 24B10 | DSHB | RRID:AB_528161 | (1:20) |
| Antibody | Anti-Brp mouse monoclonal nc82 | DSHB | RRID:AB_2314866 | (1:30) |
| Antibody | Anti-dsRed rabbit polyclonal | Clontech Laboratories, Inc: 632,496 | RRID:AB_10013483 | (1:1000) |
| Antibody | Rat cadherin, DN-Ex #8 (extracellular domain) | Developmental Studies Hybridoma Bank | RRID: AB_528121 | (1:100) |
| Antibody | Anti-HA rabbit monoclonal C29F4 | Cell Signaling Technologies: 3724S | RRID:AB_1549585 | (1:300) |
| Antibody | Anti-FLAG rat monoclonal DYKDDDDK Epitope Tag Antibody [L5], | Novus Biologicals: NBP1-06712 | RRID:AB_1625981 | (1:200) |
| Antibody | DyLight 550 conjugated anti-V5 mouse monoclonal | AbD Serotec: MCA1360D550GA | RRID:AB_2687576 | (1:500) |
| Antibody | DyLight 549 conjugated anti-V5 mouse monoclonal | AbD Serotec: MCA1360D549GA | RRID:AB_10850329 | (1:500) |
| Antibody | Anti-GFP rabbit polyclonal | ThermoFisher: A-11122 | RRID:AB_221569 | (1:1000) |
| Antibody | Anti-GFP mouse monoclonal 3E6 | ThermoFisher: A-11120 | RRID:AB_221568 | (1:100) |
| Software, algorithms | FluoRender | http://www.sci.utah.edu/software/fluorender.html | RRID:SCR_014303 | |
| Software, algorithms | VVDviewer | https://github.com/takashi310/VVD_Viewer (*Kawase, 2021*) | | |
| Software, algorithm | Python 3 | http://www.python.org | RRID:SCR_008394 | |
| Software, algorithm | pymaid | https://github.com/schlegelp/PyMaid (*Neuron Analysis & Visualization, 2021a*) | DOI https://doi.org/10.5281/zenodo.4724558 | Python library |
| Software, algorithm | FAFBseg | https://github.com/flyconnectome/fafbseg-py (*Neuron Analysis & Visualization, 2021b*) | DOI https://doi.org/10.5281/zenodo.496660 | Python library |

*Continued on next page*

*Continued*

| Reagent type (species) or resource | Designation | Source or reference | Identifiers | Additional information |
|---|---|---|---|---|
| Software, algorithm | R | https://www.r-project.org/ | RRID:SCR_001905 | |
| Software, algorithm | R Studio | https://www.rstudio.com/ | RRID:SCR_000432 | |
| Software, algorithm | Natverse | https://natverse.org | DOI:10.7554/eLife.53350 | Collection of R packages for neuroanatomical analysis |
| Software, algorithm | Tidyverse | https://www.tidyverse.org/ | RRID:SCR_019186 | R package |
| Software, algorithm | knitr | https://cran.r-project.org/package=knitr | | R package |
| Software, algorithm | kableExtra | https://cran.r-project.org/package=kableExtra | | R package |
| Software, algorithm | gridExtra | https://cran.r-project.org/package=gridExtra | | R package |
| Software, algorithm | png | https://cran.r-project.org/package=png | | R package |
| Software, algorithm | RColorBrewer | https://cran.r-project.org/package=RColorBrewer | | R package |
| Software, algorithm | alphashape3d | https://CRAN.R-project.org/package=alphashape3d | | R package |
| Software, algorithm | Fiji | http://fiji.sc | RRID:SCR_002285 | |
| Software, algorithm | MATLAB | | RRID:SCR_001622 | |
| Software, algorithm | CATMAID | http://www.catmaid.org/ | | Neuron reconstruction |
| Other | FAFB | *Zheng et al., 2018*; https://v2.virtualflybrain.org/ | | EM data set |
| Other | FAFB-FFN1 | *Li et al., 2020*; http://fafb-ffn1.storage.googleapis.com/data.html | | EM auto-segmentation |
| Other | FlyWire | *Dorkenwald et al., 2020*; https://flywire.ai/ | | EM auto-segmentation |
| Sequence-based reagent | Rh3 F | This paper | PCR primer | CACC GGT ACG CTT ATG ATA ACC TTC G |
| Sequence-based reagent | Rh3 R | This paper | PCR primer | GCT CCG GTC TGC GGG CCA AGA |
| Sequence-based reagent | Rh5 F | This paper | PCR primer | CACC GGT GAT TAA TGC GAA TGT AGC TGC |
| Sequence-based reagent | Rh5 R | This paper | PCR primer | ACT GCT TGA TCC GCT CCA AAA TC |
| Sequence-based reagent | Rh6 F | This paper | PCR primer | CA CCA ACA TGA TGG CGG ACA TCA C |
| Sequence-based reagent | Rh6 R | This paper | PCR primer | TTC GAA TGG CTG GTA CTG GTG |

## EM reconstruction

The *Zheng et al., 2018*, data set is a serial section transmission EM volume of a female *Drosophila melanogaster* brain. We manually reconstructed neurons in this volume in the CATMAID environment, which includes a browser-based annotation interface and a server hosting a copy of the data (*Saalfeld et al., 2009*). We also searched for neurons using two recent auto-segmentations FAFB-FFN1 (*Li et al., 2020*) and FlyWire (*Dorkenwald et al., 2020*), importing some of the FAFB-FFN1 auto-segmentation

fragments afterward into the CATMAID environment, a process that facilitated the identification of a small number of neurons. We followed established guidelines for manually tracing neuron skeletons, annotating synaptic connections, and reviewing reconstructed cells (*Schneider-Mizell et al., 2016*). We completely reconstructed the morphology of pairs of R7 and R8 cells in seven 'seed' columns, four adjacent columns chosen to be near the center of the retinal mosaic, and three columns in the DRA (*Figure 1C*). On the R7 and R8 cells, we annotated as a 'pre-synapse' all the locations where there was a T-bar (*Figure 1—figure supplement 1A*). Associated with each pre-synapse we annotated a 'post-synapse' connection with every synaptic partner. These seed neurons and their pre-synapses were completely proofread by independent team members, a process that amounted in small (<10%) amendments to the connectivity data presented here.

In total, we identified 4043 connections from the seed column R7 and R8 cells to 422 postsynaptic skeletons (the vast majority of which correspond to individual neurons), including the 14 R7–8 cells. We reconstructed their morphology to be able to identify the cell type (e.g. Dm8, Mi1, Tm5a) or cell class (e.g. Dm, Mi, Tm) of 299 cells, whose synapses accounted for 96.2% (3888/4043) of the total synaptic output. For all but one cell, we applied a threshold of >2 synapses to focus our reconstructions on reliable connections – the exception was the Mt-VPN2 cell, which had a large dendritic field and so may integrate many sparse R7 and R8 inputs (*Figure 5—figure supplement 1*). There were 244 cells with >2 synapses, of which 224 were identified to cell type, 16 were identified to cell class, giving a total of 240 identified cells (224 + 16 cells), and the remaining 4 were unidentified (*Table 2* and *Table 4*). The 240 identified cells with >2 synapses accounted for 94% (3803/4043) of the total synapses. The 16 cells reconstructed to cell class were sparsely connected, with 4.9 synapses per cell (mean, 79 synapses total). The R7 and R8 cells make infrequent synaptic contacts with glia as well as neurons (*Figure 1—figure supplement 1C*), but we did not systematically trace the glia connections.

The 244 cells with >2 synapses were the focus of our analysis, and their connections are summarized in (*Table 2* and *Table 4*).; the connections of individual cells are also organized by cell type for incoming and outgoing synapses in *Supplementary file 1*. In addition, the morphologies and connections of individual cells with >1 synapses are shown in the gallery figures (*Supplementary file 2*). Most neurons could be uniquely identified with incomplete reconstructions. Nevertheless, it is useful to establish a reference morphology data and so we completely reconstructed individual examples of specific cell types, in addition to the seed R7 and R8 cells, and in some cases multiple examples. In the central columns, there were 14 completely reconstructed neurons: Dm2, Dm8 (three cells), Dm9, Dm11, MeTu, Mi15, ML1, ML-VPN1, Tm5a, Tm5b, Tm5c (two cells). In the DRA, there were 10 completely reconstructed neurons: Dm-DRA1 (four cells), Dm-DRA2 (three cells), and MeTuDRA (three cells). We believe that a number of these cell types we reconstructed have not been previously described: aMe12, ML1, ML-VPN1, and ML-VPN2 in the central columns; MeMe-DRA, MeTu-DRA, DRA-VPN, and Mti-DRA in the DRA region. For one cell whose morphology was similar to the described Tm5a cell type, but whose connectivity was markedly different from the others, we annotated this cell as Tm5a-like. Likewise, for three cells whose morphology was similar to the described Tm5b cell type, but whose connectivity was markedly different from that reported by *Menon et al., 2019*, we annotated these cells as Tm5b-like. There were numerous tangential cells contacted in the central and DRA columns that do not match those characterized in previous anatomical studies (*Fischbach and Dittrich, 1989*). We have annotated these as Mti cells when the cells have processes intrinsic to the medulla, and as Mt-VPNs when they project to neuropils outside the medulla.

We additionally identified 970 incoming synapses to the seed column R7–8 cells from 56 cells, including the 14 R7–8 cells. All the 30 cells providing >2 synapses were reconstructed so that we could ascribe a cell type, and these cells provided 96% (936/970) of the total incoming synapses (*Table 3* and *Table 5*). In total, we identified the presynaptic cell type of 99% (963/970) of the synapses onto the R7–8 cells (including the 1-synapse contributing cells). All but 1 of the cells providing >2 synapses were themselves contacted by the seed column R7–8 cells, the exception being a DRA C2 cell that provided five synapses (*Supplementary file 1*). Including this C2 cell, there were 245 cells with >2 input or output synapses with the seed column photoreceptors (244 + 1 cells).

This report was focused on identifying the synaptic inputs and outputs of the R7 and R8 cells of the central and DRA columns, and it was not feasible to also detail the circuit connectivity for the cells targeted by these photoreceptors. For exceptional cell types, however, we traced more than the cells' connections to the seed column R7 and R8 cells. (1) We traced three aMe12 cells to near completion,

to determine the columns that contained the cell type's characteristic vertical processes that we used to determine whether a column was pale or not (see below). (2) We traced all the photoreceptor inputs to three Dm8 cells in the central columns (*Figure 2Cii–iii*). (3) For the Mt-VPN1 cell, we reconstructed the photoreceptor inputs in 20 medulla columns to discount the possibility that this was a pale-specific neuronal target in the central columns. (4) For three fully reconstructed Dm-DRA1 cells and three fully reconstructed Dm-DRA2 cells, we traced all their R7-DRA and R8-DRA inputs (*Figure 6E and F*). (5) For one Dm9 cell innervated by R7-DRA and R8-DRA, we traced in non-DRA columns to confirm that Dm9 cells in the DRA also receive non-DRA photoreceptor input (*Figure 6Cvi*).

## Defining pale, yellow, and DRA medulla columns

To identify medulla columns, we used a map of Mi1 neurons. This Mi1 map, which we have generated for a separate, ongoing study, includes nearly all Mi1 neurons in the medulla of the right hemisphere of the *Zheng et al., 2018*, data set. The Mi1 cell type is columnar, with one cell per column and unambiguous arborizations in medulla layers 1, 5, 9, and 10 (*Fischbach and Dittrich, 1989*). We defined the location of a column in the medulla as the center-of-mass of the Mi1 dendrite in layer 5 (*Figures 1C and 2A*).

We traced the vertical dendritic branches of all three aMe12 cells in the data set. Their vertical branches innervate pale columns (*Figure 1D*), and so we labeled the columns they occupied (by proximity to the nearest Mi1-defined column) as pale columns (*Figure 1C*). The aMe12 cells have short vertical branches reaching from M6 to M3 and longer vertical branches reaching up to M1 (*Figure 1Dii*, Eiii). Both the long and short vertical processes were used to assign pale medulla columns. Nearly all the assigned pale columns were innervated by one aMe12 cell, but three columns were innervated by two cells. Using this system, the columns not innervated by aMe12 were yellow candidates (*Figure 1C*). Despite extreme care and multiple reviews of the aMe12 neurons, errors of omission are always possible in the manual reconstruction procedure, and so we are confident that the 'pale' identified columns are innervated by aMe12, however, it is possible that a small number of the 'yellow' columns are mis-classified. The aMe12 cells innervated just 1 of the 42 identified DRA columns (*Figure 1C*). While most medulla columns innervated by R7–8 are pale or yellow, there are a small number of ommatidial cartridges with other identities, for example, rare (~1–2 per fly) ommatidial pairs of R7 and R8 cells expressing Rh3 and Rh6, respectively (*Chou et al., 1996*). The congruence of the pale and yellow specificity we observed for Tm5a, aMe12, and ML-VPN1 cell types indicated that our allocation of pale columns in the central columns was robust, and consistent with the previous report of the yellow specificity of Tm5a cells (*Karuppudurai et al., 2014*).

To identify the DRA columns, we reconstructed the R7 and R8 cells in 27 columns along the dorsal-rim-corresponding margin of the medulla and ascertained whether the R8 cell terminated in the same layer as the R7 cell (*Figure 1E*, arrow; *Figure 1—figure supplement 1A*). The R8-DRA cells terminate before the R7-DRA cells in this layer, allowing the two cell types to be distinguished. By careful visual inspection of the M6 layer of all medulla columns located close to the dorsal edge of the medulla, we identified 15 additional DRA columns resulting in a total of 42 DRA columns which is in good agreement with the previously reported average number of 39 DRA columns (*Weir et al., 2016*; *Figure 1C*).

## Analysis of synapse locations

To define the medulla and lobula layers, we first used the R package collection 'natverse' (*Bates et al., 2020*) to transform pre-defined neuropil meshes (*Jenett et al., 2012*; *Bogovic et al., 2020*) into the *Zheng et al., 2018*, data set space. Our meshes for the medulla and lobula computed in this way defined the top and bottom layers of these neuropils. We then interpolated the initial internal layer boundaries using demarcations established in prior studies (*Takemura et al., 2015*; *Wu et al., 2016*). Finally, we refined the layer boundaries using the characteristic arborizations patterns of the Mi1, C2, Tm5, and Tm20 cell types (*Gao et al., 2008*; *Fischbach and Dittrich, 1989*). For the medulla, the demarcations were: 0%, 8.2%, 26.2%, 36.1%, 45.9%, 54.1%, 62.3%, 67.2%, 76.2%, 91.0%, 100%. For the lobula the layer demarcations were: 0%, 4.2%, 11.3%, 22.4%, 37.4%, 49.4%, 64.7%, 100%. The layer designations are therefore guides to aid interpretations and comparisons, and not measurements of the neuropils themselves.

To plot the histograms of the synaptic depths in the medulla (e.g. *Figure 1Eiii*), we took the projection of the synapse locations along the columnar axis, and smoothed this distribution of synaptic

depths with a zero-phase Gaussian filter with a standard deviation of 0.4 μm. This filter width was selected to be narrow enough to allow individual synapses to be observable, for example, synapses between R7 and R8 in *Figure 2A*, and wide enough for the layer pattern to be clear, for example, the synapses from R8 to Dm9 are preferentially located in medulla layers M1 and M3 in *Figure 2B*.

For the histograms of distances of R7 synapses to each Dm8 cell's home column (*Figure 2Ciii*), we took the projection of the synapse locations onto the plane perpendicular to the columnar axis and calculated the distance in that plane to the column center. The resulting distribution of distances was then smoothed with a zero-phase Gaussian filter with a standard deviation of 0.6 μm. This filter width was selected to allow the columnar organization of the synapses to be appreciated.

For the histograms of the distances along the DRA of photoreceptor synapses to Dm-DRA1 and Dm-DRA2 cells (*Figure 7B and D*), we first fitted an ellipse to the centers of DRA columns, using least squares regression (*Figure 7—figure supplement 3Ai*). We then calculated the location of the perpendicular projection of every synapse along the fitted ellipse (*Figure 7—figure supplement 3Aii-iv*). Finally, these distances were filtered with a zero-phase Gaussian filter with a standard deviation of 0.6 μm.

## Comparison with medulla-7-column connectome

To compare the connectivity of R7 and R8 of our central column reconstructions with the connectivity identified in the medulla-7-column connectome (*Takemura et al., 2013*; *Takemura et al., 2015*; *Takemura et al., 2017*), we took advantage of the identification of pale and yellow columns in that data set used by *Menon et al., 2019*. That study used the presence or absence of Tm5a to indicate yellow and pale columns, respectively, and excluded columns that contained Tm5 neurons that were ambiguous for being Tm5a or Tm5b neurons. Using this scheme, columns B, D, and H are pale, and columns A, E, and F are yellow (here using the column references used for the 7-column data set). The medulla-7-column data set is now publicly available and accessible with the release of NeuPrint (*Clements et al., 2020*), and we used NeuPrint to compile the connectivity of cells with pale and yellow R7 and R8 cells from the data set (*Figure 1—figure supplement 2*). The unidentified cells including cells annotated as output ('out') and fragments of putative tangential cells ('tan') in the six pale and yellow columns are 38.1% for R7 (518/1342 synapses over six columns) and 38.9% for R8 (726/1854 synapses over six columns).

We also searched the medulla-7-column connectome for neurons with shapes and patterns of connectivity similar to aMe12, ML1, and MeTu neurons and identified the following putative matches (NeuPrint medulla-7-column identifiers in parenthesis): aMe12 (54028), ML1(16666), MeTu (11770; annotated as an unknown Tm; 35,751, and other reconstructions annotated as 'Dm7'). The ML1 and MeTu matches are further supported by multiple inputs other than R-cells (e.g. L3, L4, and Dm9 to ML1 and Mi15 to MeTu) that are present in both data sets. These matches provide examples of how the combination of our complete but focused reconstructions and the dense but strongly volume-limited medulla-7-column connectome can together enable insights that go beyond each individual data set.

## Genetics and molecular biology
### Fly genotypes
Fly genotypes used throughout the manuscript are listed organized by figure panel in *Supplementary file 4*.

## Driver lines
Split-GAL4 lines were constructed and characterized as in previous work (*Wu et al., 2016*). Briefly, we tested candidate AD and DBD hemidriver pairs (*Tirian and Dickson, 2017*; *Dionne et al., 2018*) for expression in cell types of interest and assembled successful combinations into stable fly strains that were then used for subsequent experiments. Split-GAL4 lines generated in this study and images of their expression patterns are available online (https://www.janelia.org/split-GAL4). aMe12, ML1, and ML-VPN1 were initially identified by light microscopy, allowing us to generate split-GAL4 driver lines targeting these previously undescribed cell types. Images of cells labeled by these driver lines were subsequently matched to the EM reconstructions. Split-GAL4 lines labeling Dm11, Mi15, L2, and VPN-DRA were from prior work (*Davis et al., 2020*; *Tuthill et al., 2013*; *Wu et al., 2016*). The

candidate VPN-DRA driver (OL0007B) was originally described as a split-GAL4 driver for a different cell type (LC12; *Wu et al., 2016*) but also labels neurons highly similar or identical to VPN-DRA. We also used GAL4 driver lines from the Janelia and Vienna Tiles collections (*Jenett et al., 2012*; *Tirian and Dickson, 2017*).

## Rhodopsin-LexA reporter constructs

To construct Rh3-, Rh5-, and Rh6-LexA driver lines, we amplified previously characterized promoter regions (*Mollereau et al., 2000*; *Pichaud and Desplan, 2001*; *Tahayato et al., 2003*; *Chou et al., 1996*; *Papatsenko et al., 1997*; *Huber et al., 1997*) by PCR from genomic DNA. Primer sequences are listed in the Key resources table. The PCR products were TOPO-cloned into pENTR-D-TOPO (Invitrogen) and transferred to pBPnlsLexA::GADflUw (addgene #26232) using standard Gateway cloning. Transgenic flies were generated by phiC31-mediated integration into the attP40 landing site (injections were done by Genetic Services, Inc).

## Histology

To characterize the neurons labeled by split-GAL4 lines, we visualized both overall expression patterns and individual cells. For the former, we used pJFRC51-3XUAS-IVS-Syt::smHA in su(Hw)attP1 and pJFRC225-5XUAS-IVS-myr::smFLAG in VK00005 (*Nern et al., 2015*) or 20XUAS-CsChrimson-mVenus in attP18 (*Klapoetke et al., 2014*) as reporters; the latter was achieved by stochastic labeling of individual cells with MCFO (*Nern et al., 2015*). Specimens were processed and imaged by the Janelia FlyLight Project team following protocols that are available online (https://www.janelia.org/project-team/flylight/protocols under 'IHC – Anti-GFP', 'IHC – Polarity Sequential', 'IHC – MCFO', and 'DPX mounting''). Additional MCFO images of cells labeled by GAL4 (instead of split-GAL4) driver lines were generated in the same way. Images were acquired on Zeiss LSM 710 or 780 confocal microscope using 20 × 0.8 NA or 63 × 1.4 NA objectives.

The specimens shown in *Figure 7Aiii*, 7Ciii, 8Aii, and 8v were prepared and imaged as previously described (*Sancer et al., 2019*).

For the combined labeling of aMe12, ML-VPN1, and VPN-DRA with photoreceptor markers, fly brains were processed using standard immunolabeling protocols as previously described in *Davis et al., 2020*. Briefly, flies were dissected in insect cell culture medium (Schneider's Insect Medium, Sigma Aldrich, #S0146) followed by fixation with 2% PFA (w/v; prepared from a 20% stock solution, Electron Microscopy Sciences: 15713) in cell culture medium for 1 hr at room temperature. This fixation step and the subsequent primary and secondary antibody incubations were each followed by several washes with PBT (0.5% v/v TX-100, Sigma Aldrich: X100, in PBS). To block nonspecific antibody binding, brains were incubated in PBT-NGS (5% Goat Serum ThermoFisher: 16210–064 in PBT) for at least 30 min prior to addition of primary antibodies in PBT-NGS. Primary and secondary antibody incubations were at 4°C overnight, all other steps were at room temperature. Brains were mounted in SlowFadeGold (ThermoFisher: S36937).

Primary antibodies were anti-GFP rabbit polyclonal (ThermoFisher: A-11122, RRID:AB_221569; used at 1:1000 dilution), anti-GFP mouse monoclonal 3E6 (ThermoFisher: A-11120, RRID:AB_221568; dilution 1:100), anti-dsRed rabbit polyclonal (Clontech Laboratories, Inc: 632496, RRID:AB_10013483; dilution 1:1000), anti-chaoptin mouse monoclonal 24B10 (*Fujita et al., 1982*; DSHB: RRID:AB_528161, dilution 1:20), and anti-Brp mouse monoclonal nc82 (*Wagh et al., 2006*; DSHB:RRID:AB_2314866; dilution 1:30). Secondary antibodies were from Jackson ImmunoResearch Laboratories, Inc Images were acquired on a Zeiss LSM 880 confocal microscope using a 40 × 1.3 NA objective.

## Image processing

Most anatomy figures show reconstructed views of cells of interest generated from confocal stack using FluoRender (http://www.sci.utah.edu/software/fluorender.html) or the related VVDviewer (https://github.com/takashi310/VVD_Viewer). Some images were manually edited to only show the cell relevant for the comparison to the EM reconstructions or to exclude parts of the image stacks that would occlude the illustrated views. Some panels in *Figure 5* are overlays of registered images with either the template brain used for registration (*Bogovic et al., 2020*) or the pattern of a second registered expression pattern (L2 lamina neuron terminals in *Figure 5Cv*). Images with rhodopsin reporter (Rh3-, Rh5-, or Rh6-LexA) labeling show single sections or maximum intensity projections

through a small number of adjacent slices. For these images, other processing was limited to adjustments of brightness and contrast across the entire field of view for each channel. Manual cell counts were performed using Fiji (http://fiji.sc).

## Data availability statement

All data generated or analyzed during this study are included in the manuscript and supporting files. *Supplementary files 1 and 3* contain all connectivity data. *Supplementary file 2* provides images of all EM skeletons. All code and necessary data to perform the analysis and generate the figures of this manuscript is available from https://github.com/reiserlab/FAFB-photoreceptor-connectivity (*Reiser Lab, 2022*; copy archived at swh:1:rev:f8e6732868c4afa184a379abfbc5042eef920321). All reconstructed neurons described in the manuscript will be made available at https://fafb.catmaid.virtualflybrain.org/.

## Acknowledgements

The authors thank: Lou Scheffer, Shinya Takemura, and Kazunori Shinomiya for access to data, advice, and discussions; Ruchi Parekh for managing the Connectome Annotation Team and coordinating the tracing effort; the FAFB tracing community for supportive and open sharing of methods and data; Janelia Fly Core for fly care and Janelia FlyLight Project Team for help with preparation and imaging of light microscopy samples; the Janelia Visitor program for facilitating early stages of this collaboration; Yu-Chieh David Chen for comments on an earlier version of the manuscript and the reviewers for their helpful and constructive feedback. This work was supported by the Deutsche Forschungsgemeinschaft (DFG) through grants WE 5761/2–1 (MFW) and SPP2205 (MFW), AFOSR grant FA9550-19-1-7005 (MFW), with support from the Fachbereich Biologie, Chemie & Pharmazie of the Freie Universität Berlin (MFW) and by the Howard Hughes Medical Institute through its support of the Janelia Research Campus (DDB, MBR, and GMR).

## Additional information

### Funding

| Funder | Grant reference number | Author |
| --- | --- | --- |
| Howard Hughes Medical Institute | | Kit D Longden<br>Aljoscha Nern<br>Arthur Zhao<br>Miriam A Flynn<br>Connor W Laughland<br>Bruck Gezahegn<br>Henrique DF Ludwig<br>Alex G Thomson<br>Heather Dionne<br>Davi D Bock<br>Gerald M Rubin<br>Michael B Reiser |
| Freie Universität Berlin | | Emil Kind<br>Gizem Sancer<br>Tessa Obrusnik<br>Paula G Alarcón<br>Mathias F Wernet |
| Deutsche Forschungsgemeinschaft | WE 5761/2-1 | Mathias F Wernet |
| Deutsche Forschungsgemeinschaft | WE 5761/4-1 | Mathias F Wernet |
| Air Force Office of Scientific Research | FA9550-19-1-7005 | Mathias F Wernet |

| Funder | Grant reference number | Author |
|---|---|---|
| Fachbereich Biologie, Chemie & Pharmazie of the Freie Universität Berlin | | Emil Kind Gizem Sancer Paula G Alarcón Tessa Obrusnik Mathias F Wernet |
| Deutsche Forschungsgemeinschaft | SPP 2205 | Mathias F Wernet |

The funders had no role in study design, data collection and interpretation, or the decision to submit the work for publication.

## Author contributions

Emil Kind, Conceptualization, Data curation, Formal analysis, Investigation, Methodology, Software, Validation, Visualization, Writing – original draft, Writing – review and editing; Kit D Longden, Formal analysis, Methodology, Validation, Visualization, Writing – original draft, Writing – review and editing; Aljoscha Nern, Conceptualization, Formal analysis, Investigation, Methodology, Resources, Validation, Visualization, Writing – original draft, Writing – review and editing; Arthur Zhao, Data curation, Formal analysis, Methodology, Software, Validation, Visualization, Writing – original draft, Writing – review and editing; Gizem Sancer, Miriam A Flynn, Connor W Laughland, Bruck Gezahegn, Henrique DF Ludwig, Alex G Thomson, Tessa Obrusnik, Paula G Alarcón, Investigation, Methodology; Heather Dionne, Investigation, Resources; Davi D Bock, Data curation, Funding acquisition, Methodology, Resources; Gerald M Rubin, Funding acquisition, Resources, Supervision, Writing – review and editing; Michael B Reiser, Conceptualization, Data curation, Funding acquisition, Methodology, Project administration, Resources, Supervision, Visualization, Writing – original draft, Writing – review and editing; Mathias F Wernet, Conceptualization, Funding acquisition, Methodology, Project administration, Resources, Supervision, Visualization, Writing – original draft, Writing – review and editing

## Author ORCIDs

Emil Kind http://orcid.org/0000-0001-5228-7638
Kit D Longden http://orcid.org/0000-0002-7686-6447
Aljoscha Nern http://orcid.org/0000-0002-3822-489X
Arthur Zhao http://orcid.org/0000-0003-2869-4393
Gizem Sancer http://orcid.org/0000-0002-0367-9421
Gerald M Rubin http://orcid.org/0000-0001-8762-8703
Michael B Reiser http://orcid.org/0000-0002-4108-4517
Mathias F Wernet http://orcid.org/0000-0001-5233-2654

## Decision letter and Author response

Decision letter https://doi.org/10.7554/eLife.71858.sa1
Author response https://doi.org/10.7554/eLife.71858.sa2

# Additional files

## Supplementary files

• Supplementary file 1. Tables of all seed column R7, R8, R7-DRA, and R8-DRA target cells by type.

• Supplementary file 2. Gallery plots of all seed column R7, R8, R7-DRA, and R8-DRA target cells by type.

• Supplementary file 3. Tables of seed column synapses outside the medulla.

• Supplementary file 4. Fly genotypes organized by figure panel.

• Transparent reporting form

## Data availability

All data generated or analyzed during this study are included in the manuscript and supporting files. Supplementary File 1 and 3 contain all connectivity data. Supplementary File 2 provides images of all EM skeletons. All code and necessary data to perform the analysis and generate the figures of this manuscript will be available from https://github.com/reiserlab/FAFB-photoreceptor-connectivity,

The following previously published datasets were used:

| Author(s) | Year | Dataset title | Dataset URL | Database and Identifier |
|---|---|---|---|---|
| Zheng Z | 2018 | Temca2 | https://temca2data.org/ | FAFB, temca2data |

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
