## [Editor Report]

This paper will be of interest to the large class of neuroscientists who perform network analyses and are interested in the processing of visual information. It sets a new standard in connectomic analysis because it combines EM data of a whole fly brain with fluorescent labeling of specific neurons. The key claims of the manuscript are well supported by the data, and the approaches used are thoughtful and rigorous.

---

## [Decision Letter]

**Decision letter after peer review:**

Thank you for submitting your article "Synaptic targets of photoreceptors specialized to detect color and skylight polarization in *Drosophila*" for consideration by *eLife*. Your article has been reviewed by 3 peer reviewers, and the evaluation has been overseen by Ronald Calabrese as the Senior and Reviewing Editor. The following individuals involved in review of your submission have agreed to reveal their identity: Simon G Sprecher (Reviewer #1); Charlotte Helfrich-Förster (Reviewer #2).

Essential revisions:

The authors have done a great job in the research and presentation. Some revisions will enhance the manuscript. These are detailed in the Recommendations to the Authors. As a high-level guide:

1) Describe more extensively how you have dealt with the edges of the reconstructed areas.

2) Some new cell types are described and any further information about these cells would enhance the usefulness of the manuscript.

3) Restrict abbreviations to a minimum and add an abbreviation list.

4) In print, many figures are too small to see the details and read the lettering. There are several places visibility can be improved by choosing different colors or increasing the brightness. For example, please always depict the reconstructed neurons in black instead of gray. The gray thin lines are extremely hard to see.

*Reviewer #1 (Recommendations for the authors):*

I feel this is an excellent manuscript clearly summarizing the findings of the EM-based connectomic study. I only have a few comments that I feel should be addressed.

– One of the most challenging parts of describing EM-based connectomes is to find a clear logic and structure in describing individual circuit elements. I feel the authors have done this in a very clear fashion using the same structure for all subheadings. I do not have any comment on this, just wanted to highlight that this makes reading indeed easier.

– Since many cells in the optic ganglia are interconnecting with neighboring columns and cartridges it is unclear how the delimited the extend to completely reconstruct all neurons. For all the cases they clearly state how many cells they reconstructed (e.g. line 233 "We reconstructed six Dm9 cells»), I would assume that they had to identify all input neurons in order to make a clear statistical statement. In other words how did the authors deal with the edges of the reconstructed areas? It may be beneficial to also explain this and also to some degree in the result section.

– Interestingly the authors also describe new types of neurons. Personally, I would have loved to see a bit more about these cells, since they are among the most striking novelties of the current manuscript.

– Similarly, the discussion appears a bit short. I do of course understand that the detailed description of connectivity is at the core of the manuscript. Also much of discussion is already in the Results section, nevertheless I feel in particular with such a rich data set one might want to take advantage of discussing how this information will help to go beyond current work on circuit dissection.

*Reviewer #2 (Recommendations for the authors):*

General comments:

1) I strongly recommend to restrict abbreviations to a minimum and add an abbreviation list.

2) In print, many figures are too small to see the details and read the lettering. I understand that most people will only read the paper online, but in several places visibility can be easily improved by choosing different colors or increasing the brightness. For example, I strongly recommend to depict the reconstructed neurons always in black instead of gray. The gray thin lines are extremely hard to see.

Specific comments:

Line 64: Please specify more precisely what you mean with the „main part of the retina"

Line 84: What exactly is the „central medulla"?

Lines 38, 110…: Is it really necessary to abbreviate the „Full Adult Fly Brain" (FAFB)?

Line 113: According to the systematic nomenclature of the insect brain by Ito et al., (2014), the accessory medulla has to be abbreviated as „AME"

Line 114: What about: „We then reconstructed and subsequently used this cell type…"

Line 293: Please define „medulla FIB-SEM data". Is this abbreviation really needed?

Line 303: The serpentine layer M7 is mentioned here for the first time. I wonder whether it is useful to highlight this important layer in faint gray in all figures showing the medulla layering. This would help orientation in the medulla a lot, especially those readers that are not specialists of the *Drosophila* optic lobe.

Related to the serpentine layer, I have another question. I noticed that many of the R7/R8 targets arborize in the serpentine layer. Why this is so? Do they have synapses with other neurons of the same or different types? If yes, what is the biological meaning?

Lines 351-352: "The cholinergic Tm5a cell type was the first cell type described to show pale versus yellow selectivity, being preferentially innervated by yellow R7."

I would write "show yellow versus pale selectivity".

Line 402: Has the Tm20 cell type a preference for pale or yellow?

Line 422 and Figure 4Dii, v, vi: The lobula layers are not visible in the figure. They are much too faint.

Line 458: How many ML-VPN1 cells are present?

Lines 704-707 and lines 818-822: I find it quite interesting that the inner photoreceptor cells make contact to L3 and L1. What about the other lamina monopolar cells? You don't describe any contact to these. Is it completely absent, and if yes, can you speculate about the biological meaning?

Lines 757-759: I am puzzled be the following sentence:

"Unlike in the DRA region, there were no cells projecting to the central brain that were substantial targets of R7."

Line 907: "the FIB-SEM/FAFB combination" These abbreviations are not useful.

Line 921: "pale or yellow" instead of "pale of yellow".

Figure 1Di: The fluorescent images are too dark. Please increase the brightness, especially of the magenta channel.

Figure 1Eii is too small. Please enlarge and indicate additional the medulla layer numbers.

Figure 1G and H: the not bold yellow lettering is hard to read. Is there a reason that L3 is depicted but not the other lamina monopolar cells?

Figure 1 – figure supplement 2 F: this image is too small.

Figure 3 – supplement 1B-D: The reconstructions of the cells have to be enlarged. The layering of the medulla cannot be seen.

Figure 4 – figure supplement 1 Aiii, Biii, Ciii: no chance to see the neuropil borders. The reconstructed neurons are also too faint.

---

## [Author Response]

Reviewer #1 (Recommendations for the authors):I feel this is an excellent manuscript clearly summarizing the findings of the EM-based connectomic study. I only have a few comments that I feel should be addressed.– One of the most challenging parts of describing EM-based connectomes is to find a clear logic and structure in describing individual circuit elements. I feel the authors have done this in a very clear fashion using the same structure for all subheadings. I do not have any comment on this, just wanted to highlight that this makes reading indeed easier.

We thank the reviewer for their comments and help, indeed making this rather complex manuscript now easier to read.

– Since many cells in the optic ganglia are interconnecting with neighboring columns and cartridges it is unclear how the delimited the extend to completely reconstruct all neurons. For all the cases they clearly state how many cells they reconstructed (e.g. line 233 "We reconstructed six Dm9 cells»), I would assume that they had to identify all input neurons in order to make a clear statistical statement. In other words how did the authors deal with the edges of the reconstructed areas? It may be beneficial to also explain this and also to some degree in the result section.

We now clarify this point in the opening of the Results.

Line 188: “This report is a comprehensive reconstruction of all inner photoreceptor synaptic outputs and inputs in our sample seed columns. The dataset comprises cells that were identified and reconstructed because they are connected, by identifiable synapses, to our seed column inner photoreceptors. To be clear what this dataset is not, it is not a reconstruction of all the cells, and the connections between them, in a volume around the seed columns.”

– Interestingly the authors also describe new types of neurons. Personally, I would have loved to see a bit more about these cells, since they are among the most striking novelties of the current manuscript.

We share the reviewer’s enthusiasm for learning more about these new cell types and have added some additional information to facilitate further exploration.

For the ML1, ML-VPN1 and Mti2 cell types, for which we have selective GAL4 lines, we have added and refined our data on the number of cells.

Line 331. “In our EM reconstructions, two cells shared these properties and overlapped around our seed columns (Figure 3Bv). From light microscopy, we estimate there are ~50 Mti2 cells. However, we note that the driver line used for this estimate may include related but distinct cell types with a similar cell body location or might not label all Mti2 cells.

Line 430. “We completely reconstructed one cell and used light microscopy data to explore the anatomy of the population (Figure 4D v-vi), and we estimate that there are ~45 cells per optic lobe (counts of 44, 44, 45 and 47 from four optic lobes).”

Line 477. “In our match using light-microscopic data, the dendrites of individual neurons overlap, with each cell spanning tens of columns, collectively covering the medulla (Figure 5Ci-vi), and we estimate ~65 cell bodies per optic lobe (counts of 64, 64, 65 and 68 from four optic lobes).”

– Similarly, the discussion appears a bit short. I do of course understand that the detailed description of connectivity is at the core of the manuscript. Also much of discussion is already in the Results section, nevertheless I feel in particular with such a rich data set one might want to take advantage of discussing how this information will help to go beyond current work on circuit dissection.

We really appreciate this supportive comment. The Discussion is 1650 words and given the length of paper, which already includes substantial background information throughout the Results, we prefer to retain a concise Discussion.

Reviewer #2 (Recommendations for the authors):General comments:1) I strongly recommend to restrict abbreviations to a minimum and add an abbreviation list.

Thank you for this suggestion. We have eliminated the use of the acronym ‘FAFB’ from the manuscript, and instead refer to the ‘Zheng et al., (2018) data set’ in seven instances. The medulla data set of Takemura et al., (2015, 2017) has been standardized to the ‘medulla-7-column connectome’. We have, as suggested, included a list of abbreviations used in Table 5, which is introduced at the end of the opening Results section, where we clarify the organization of the further results in the manuscript. The FAFB acronym is included in the table, as it is the standard name used by many groups for this data set, and it is mentioned indirectly in the methods via the software used for analysis.

The medulla-7-column connectome was released over a series of papers (Takemura et al., 2015, 2017) and updated and fully released in Clements et al., (2020). Unlike the Zheng et al., (2018) data set, it is not possible to refer to it by a single reference.

Line 217: “A key prior EM data set for comparison with our results is the medulla connectome of Takemura et al., (2015, 2017), fully released in Clements et al., (2020). We refer to this connectome as the ‘medulla-7-column connectome’. For ease of reference, all abbreviations used, including anatomical abbreviations, are listed in Table 5.”

2) In print, many figures are too small to see the details and read the lettering. I understand that most people will only read the paper online, but in several places visibility can be easily improved by choosing different colors or increasing the brightness. For example, I strongly recommend to depict the reconstructed neurons always in black instead of gray. The gray thin lines are extremely hard to see.

Thank you for this very helpful feedback. We have updated all relevant figures by making the neuron lines black and increasing the resolution of the rendering by 50%. We have also checked to ensure that all lettering is 6 point or larger, following common publishing guidelines. Where possible, we have increased the sizes of reconstructed neurons and increased the contrast of lines and lettering.

Specific comments:Line 64: Please specify more precisely what you mean with the „main part of the retina"

Thank you for pointing this out. The text has been revised (line 64):” The organization of the inner photoreceptors along the dorsal rim area (DRA) of the eye characteristically differs from that of the rest of the retina. In the nonDRA part of the retina,…”

Line 84: What exactly is the „central medulla"?

We have replaced “central medulla” with “non-DRA medulla”.

Lines 38, 110…: Is it really necessary to abbreviate the „Full Adult Fly Brain" (FAFB)?

This has been removed from the text, except where it is used in the software names etc. and is mentioned in Table 5 for this reason.

Line 113: According to the systematic nomenclature of the insect brain by Ito et al., (2014), the accessory medulla has to be abbreviated as „AME"

Thanks for pointing this out. We have changed all non-standard abbreviations of brain region names to be consistent with Ito et al., 2014 (e.g. “AME’ instead of ‘Acc Med.’ In Figure1 Ei). We kept ‘aMe12’, which refers to a cell type, not the accessory medulla brain region as such. The ‘aMe12’ name has by now also been used elsewhere (for example, in the neuprint release of the hemibrain connectome). Upon consideration, we think it is more helpful to maintain the link with other studies than to create different names for the same cell in the literature.

Line 114: What about: „We then reconstructed and subsequently used this cell type…"

Quite so, thank you. We have updated the text.

Line 293: Please define „medulla FIB-SEM data". Is this abbreviation really needed?

Apologies again for the handling of the acronyms. This point has been addressed above in the response to the use of abbreviations throughout the manuscript.

Line 303: The serpentine layer M7 is mentioned here for the first time. I wonder whether it is useful to highlight this important layer in faint gray in all figures showing the medulla layering. This would help orientation in the medulla a lot, especially those readers that are not specialists of the *Drosophila* optic lobe.

This is an excellent suggestion, and we tested how it looked. It reduced the contrast of the reconstructed neurons, and this being a raised concern, we elected to not implement it. Nevertheless, we are very grateful for the suggestion.

Related to the serpentine layer, I have another question. I noticed that many of the R7/R8 targets arborize in the serpentine layer. Why this is so? Do they have synapses with other neurons of the same or different types? If yes, what is the biological meaning?

Many of the cells arborizing in the serpentine layer extend laterally over many columns, e.g., aMe12, the Mti, and Mt-VPN cells. There are several cell types targeted by R7 and R8 that would appear to have large receptive fields, and it is possible that they are involved in chromatic processing over relatively large visual receptive fields. We could confirm this for aMe12, however, it was beyond the scope of the paper to verify the spatial distribution of R7 and R8 inputs for these other, wide field cell types, or the connectivity of these cells with each other. These are questions we would very much hope to address in future studies, both ones focused on particular cell classes, such as the aMe cells, and in connectomic analyses of the optic lobes as a whole.

Lines 351-352: "The cholinergic Tm5a cell type was the first cell type described to show pale versus yellow selectivity, being preferentially innervated by yellow R7."I would write "show yellow versus pale selectivity".

Thank you for this suggestion, now implemented.

Line 402: Has the Tm20 cell type a preference for pale or yellow?

The Tm20 cell type does not have a preference for pale or yellow. The numbers of synapses from R8 to Tm20 are quite consistent (range 30-35) across the pale and yellow columns (Figure 4—figure supplement 1 and Supplementary File 1, page 4). Because the inner photoreceptor input is restricted to the column, the direct photoreceptor input is either pale or yellow, and so the cell is potentially able to convey pale- or yellow-specific spectral information. This lack of pale or yellow selectivity is represented in the summary diagrams in Figure 11Bii where Tm20 appears as a major target of both pale and yellow R8s.

Line 422 and Figure 4Dii, v, vi: The lobula layers are not visible in the figure. They are much too faint.

Thank you for pointing this out. We have labelled the lobula layers (L1 – L6), used black rather than gray lines, and reoriented the lobula plots to make them easier to read.

Line 458: How many ML-VPN1 cells are present?

From light microscopy, we estimate ~65. The text now includes this estimate.

Line 477. “In our match using light-microscopic data, the dendrites of individual neurons overlap, with each cell spanning tens of columns, collectively covering the medulla (Figure 5Ci-vi), and we estimate ~65 cell bodies per optic lobe (counts of 64, 64, 65 and 68 from four optic lobes).”

Lines 704-707 and lines 818-822: I find it quite interesting that the inner photoreceptor cells make contact to L3 and L1. What about the other lamina monopolar cells? You don't describe any contact to these. Is it completely absent, and if yes, can you speculate about the biological meaning?

Thank you for your interest on this point. The connectivity of R7 and R8 with lamina cell types was a major finding of the previous work (Takemura et al., 2008, 2013, 2015, 2017), and so we emphasized the additional synapses in the optic chiasm that had not been previously reported. But the summary of our new findings is correct – we find that the inner photoreceptors make contacts with L1 and L3, but do not make contacts with L2, L4 or L5 with ≥ 3 synapses per cell; they do make contact with C2 and C3 with ≥ 3 synapses per cell. There are also some contacts with C2 and Lawf2 cells with <3 synapses in the central columns, and with L2 and C3 in DRA columns, which are listed in Supplementary File 1 in the ‘Identified <3 synapses’ tables.

With respect to the absence of connections with L2, L4 and L5, it is difficult to claim an absence from our sample, but these connections, if present, are likely to be weak. In the data set of Takemura et al., (2015, 2017), there are a small number of weak connections (mean < 2) between R7 and R8 and L5, which we show in Figure 1 Supplement 2 D.

We agree that R7 and R8 input into monopolar cells is interesting. Wardill et al., (2012) reported that LMCs receive spectral input from R7 and R8 but they proposed that gap junctions might be responsible. We show that there are many more direct synaptic inputs to L1 and L3 than known before, and that these inputs have been missed because they are in the optic chiasm. The likely significance of these connections is that neurons downstream of L1 and L3 may convey spectrally broader signals than would be expected if they only received R1-6 inputs.

Lines 757-759: I am puzzled be the following sentence:"Unlike in the DRA region, there were no cells projecting to the central brain that were substantial targets of R7."

Apologies for the confusion. We have removed this sentence.

Line 907: "the FIB-SEM/FAFB combination" These abbreviations are not useful.

Thank you. The sentence now reads:

Line 925: “While full exploration and follow-up analyses of these combined data is beyond the scope of this work, the combined analysis of the Zheng et al., (2018) and medulla-7-column connectome revealed several intriguing connectivity patterns, including new candidate paths for the integration of output from different photoreceptor types.”

Line 921: "pale or yellow" instead of "pale of yellow".

Thank you! Updated.

Figure 1Di: The fluorescent images are too dark. Please increase the brightness, especially of the magenta channel.

The brightness of the images, particularly the magenta channel, has been increased in Figure 1Di.

Figure 1Eii is too small. Please enlarge and indicate additional the medulla layer numbers.

We have enlarged Figure 1Eiil by 50%, added labels to indicate medulla layers, and used black lines for the neurons.

Figure 1G and H: the not bold yellow lettering is hard to read.

Thank you. The yellow lettering has been adjusted to be more saturated and darker.

Is there a reason that L3 is depicted but not the other lamina monopolar cells?

The legend text for Figure 1G has been revised to clarify this point.

Line 1272. “G. Illustrations of examples of cell types from reconstructed neuron classes, including lamina monopolar (L3), distal medulla (Dm9), medulla intrinsic (Mi15), transmedulla (Tm20), medulla tangential intrinsic (Mti), visual projection neurons targeting the central brain (example: aMe12), and medulla-to-tubercle (MeTu) cells projecting to the anterior optic tubercle (AOTU).”

Figure 1 —figure supplement 2 F: this image is too small.

Figure 1 supplement 2F has been enlarged.

Figure 3 – supplement 1B-D: The reconstructions of the cells have to be enlarged. The layering of the medulla cannot be seen.

The reconstructions of the cells in Figure 3 supplement 1B-D have been enlarged, and the medulla layers labelled.

Figure 4 —figure supplement 1 Aiii, Biii, Ciii: no chance to see the neuropil borders. The reconstructed neurons are also too faint.

Thank you again. All panels are now larger, with neurons in black and layers labelled.